# On the Efficacy of Differentially Private Few-shot Image Classification

**Marlon Tobaben**[*]                                          *marlon.tobaben@helsinki.fi*
*University of Helsinki*

**Aliaksandra Shysheya**[*]                                    *as2975@cam.ac.uk*
*University of Cambridge*

**John Bronskill**                                             *jfb54@cam.ac.uk*
*University of Cambridge*

**Andrew Paverd**                                             *andrew.paverd@microsoft.com*
*Microsoft*

**Shruti Tople**                                              *shruti.tople@microsoft.com*
*Microsoft*

**Santiago Zanella-Béguelin**                                 *santiago@microsoft.com*
*Microsoft*

**Richard E. Turner**                                         *ret26@cam.ac.uk*
*University of Cambridge*

**Antti Honkela**                                             *antti.honkela@helsinki.fi*
*University of Helsinki*

**Reviewed on OpenReview:** *https://openreview.net/forum?id=hFsr59Imzm*

## Abstract

There has been significant recent progress in training differentially private (DP) models which achieve accuracy that approaches the best non-private models. These DP models are typically pretrained on large public datasets and then fine-tuned on private downstream datasets that are relatively large and similar in distribution to the pretraining data. However, in many applications including personalization and federated learning, it is crucial to perform well (i) in the few-shot setting, as obtaining large amounts of labeled data may be problematic; and (ii) on datasets from a wide variety of domains for use in various specialist settings. To understand under which conditions few-shot DP can be effective, we perform an exhaustive set of experiments that reveals how the accuracy and vulnerability to attack of few-shot DP image classification models are affected as the number of shots per class, privacy level, model architecture, downstream dataset, and subset of learnable parameters in the model vary. We show that to achieve DP accuracy on par with non-private models, the shots per class must be increased as the privacy level increases. We also show that learning parameter-efficient FiLM adapters under DP is competitive with learning just the final classifier layer or learning all of the network parameters. Finally, we evaluate DP federated learning systems and establish state-of-the-art performance on the challenging FLAIR benchmark.

---

[*]These authors contributed equally

## 1 Introduction

It is well known that neural networks trained without formal privacy guarantees can be attacked to expose a subset of the training data (Carlini et al., 2021; Balle et al., 2022). For applications where training data are sensitive (Abowd, 2018; Cormode et al., 2018), it has become increasingly common to train under Differential Privacy (DP) (Dwork et al., 2006) which is considered to be the gold standard for protecting the privacy of individual training examples. Training with DP stochastic gradient descent (DP-SGD) (Rajkumar & Agarwal, 2012; Song et al., 2013; Abadi et al., 2016), which adapts SGD to guarantee DP, typically impairs model performance due to gradient clipping and the addition of noise during training in order to mask the contribution of individual examples to model updates. However, there has been significant recent progress in training DP models which achieve accuracy that approaches the best non-private models in both NLP (Li et al., 2022b; Yu et al., 2022) and computer vision (Kurakin et al., 2022; De et al., 2022; Mehta et al., 2022; Cattan et al., 2022).

The majority of these approaches are based on transfer learning where the models have been pretrained on large public datasets and then fine-tuned (Yosinski et al., 2014) on a private downstream dataset with DP-SGD, as transfer learning has been shown to be highly effective on non-private data (Kolesnikov et al., 2020; Shysheya et al., 2022). In the non-private setting, the subset of model parameters to fine-tune ranges from all model parameters (Kolesnikov et al., 2020) to only the final layer, with the tuning of parameter-efficient adapters (Perez et al., 2018; Houlsby et al., 2019; Mahabadi et al., 2021) becoming increasingly prevalent. Transfer learning has also proven successful in the DP setting with (Yu et al., 2022) and without (Mehta et al., 2022) adapters.

However, strong DP results have only been demonstrated with relatively large datasets, with no extensive DP few-shot studies performed. The few-shot setting is crucial to any application where obtaining large amounts of labeled data is problematic. It is particularly significant in federated learning, where a global model is trained using data from multiple distributed users, and personalized federated learning, which involves customizing a federated learning model with a specific user's data. In such scenarios, each user's data may be sensitive and of limited size, such as medical images (Sheller et al., 2020), personal photos (Massiceti et al., 2021), or confidential personal data or actions entered on a mobile device (Differential Privacy Team, 2017; Ding et al., 2017).

In addition, the strong DP transfer learning results that have recently been reported have largely considered the case where the data distribution of the downstream dataset is similar to the pretraining data distribution (Tramèr et al., 2022). A more demanding test is out-of-domain transfer where more information needs to be extracted from the downstream dataset, making private learning more challenging. Support for differing data distributions is essential for frequently encountered specialist settings such as medical imaging, Earth imaging, or personalized object recognition.

In this work, we answer the question: *Under what conditions is differentially private few-shot image classification effective?* Our contributions are:

- We provide the first comprehensive study on the efficacy of DP few-shot image classification. In particular, in the centralized setting we perform an exhaustive set of experiments that reveals how the accuracy of DP and non-private models are affected as the number of shots per class, privacy level, downstream dataset, model architecture, and the subset of learnable parameters in the model vary. We also investigate whether the trends observed in the centralized setting apply to federated learning. Novel insights include:

  *Amount of data required*: It is known that classification accuracy under DP decreases as the level of privacy increases and the amount of data decreases, however: (i) we quantify how much more data is required under various levels of DP to match non-private accuracy. In particular, we found that the number of shots per class must be increased significantly to match non-private performance, depending on the subset of learnable parameters; and (ii) we show that accuracy under DP is strongly related to the difficultly of the transfer learning task.

  *Model parameterization*: We show that fine-tuning parameter-efficient FiLM adapters in addition to the final linear classifier layer performs close to or better than fine-tuning all parameters in the model or

fine-tuning only the final layer under few-shot DP. This is demonstrated by superior accuracy for the FiLM configuration on the challenging VTAB-1k benchmark and establishing state-of-the-art in terms of accuracy (macro average precision increased from 44.3% to 51.9%) and communication efficiency (cost reduced from 11.9M to 0.017M parameters per round) on the large-scale FLAIR federated learning benchmark.

*Characterization of few-shot DP learning dynamics*: We show that non-private few-shot transfer learners are generally in the interpolating regime where they achieve 100% training accuracy. Under strong DP, trained networks are generally in the regularization regime where test and train accuracies are comparable.

- We assess the vulnerability of DP few-shot models with a strong membership inference attack (MIA) and find that non-private models are highly susceptible and the privacy level must be increased to a high level to mitigate them.

- Finally, we establish recommended practice guidelines for training DP few-shot models.

## 2 Background

In this section, we provide background information, definitions, and nomenclature required for subsequent sections. We focus our analysis on few-shot transfer learning based image classifiers that rely on large backbones pretrained on non-private data.

**Preliminaries** We denote input images $\boldsymbol{x}$ and image labels $y \in \{1, \ldots, C\}$ where $C$ is the number of image classes indexed by $c$. Assume that we have access to a model $f(\boldsymbol{x}) = h_{\boldsymbol{\phi}}(b_{\boldsymbol{\theta}}(\boldsymbol{x}))$ that outputs class-probabilities for an image $p(y|\boldsymbol{x}, \boldsymbol{\theta}, \boldsymbol{\phi}) = f(\boldsymbol{x}, \boldsymbol{\theta}, \boldsymbol{\phi})$ and comprises a feature extractor backbone $b_{\boldsymbol{\theta}} : \mathbb{R}^d \to \mathbb{R}^{d_b}$ with parameters $\boldsymbol{\theta}$ pretrained on a large upstream public dataset such as Imagenet-21K (Russakovsky et al., 2015) where $d$ is the input image dimension and $d_b$ is the output feature dimension, and a linear layer classifier or head $h_{\boldsymbol{\phi}} : \mathbb{R}^{d_b} \to \mathbb{R}^C$ with weights $\boldsymbol{\phi}$. Let $\mathcal{D} = \{(\boldsymbol{x}_n, y_n)\}_{n=1}^N$ be the private downstream dataset that we wish to fine-tune the model $f$ to. We denote the number of training examples per class or *shot* as $S$.

**Learnable Parameters** In all experiments, the head parameters $\boldsymbol{\phi}$ are initialized to zero and are always learned when fine-tuning on $\mathcal{D}$. For the backbone weights $\boldsymbol{\theta}$, we consider three options: (i) *Head*: $\boldsymbol{\theta}$ are fixed at their pretrained values and do not change during fine-tuning, only the head parameters $\boldsymbol{\phi}$ are updated; (ii) *All*: $\boldsymbol{\theta}$ are initialized with pretrained values, but can be updated during fine-tuning in addition to the head; and (iii) *FiLM*: using FiLM (Perez et al., 2018) layers. There exists myriad of adaptors for both 2D convolutional and transformer networks including FiLM, Adapter (Houlsby et al., 2019), LoRA (Hu et al., 2022a), VPT (Jia et al., 2022), AdaptFormer (Chen et al., 2022c), NOAH (Zhang et al., 2022), Convpass (Jie & Deng, 2022), Model Patch (Mudrakarta et al., 2019), and CaSE (Patacchiola et al., 2022) that enable a pretrained network to adapt to a downstream dataset in a parameter-efficient manner. In this work, we use FiLM due to its simplicity, high performance, and low parameter count (Shysheya et al., 2022), though another adapter could be used. A FiLM layer scales and shifts the activations $\boldsymbol{a}_{ij}$ arising from the $j^{th}$ output of a layer in the $i^{th}$ block of the backbone as $\texttt{FiLM}(\boldsymbol{a}_{ij}, \gamma_{ij}, \beta_{ij}) = \gamma_{ij}\boldsymbol{a}_{ij} + \beta_{ij}$, where $\gamma_{ij}$ and $\beta_{ij}$ are scalars. We implement FiLM by fixing $\boldsymbol{\theta}$ at their pretrained values except for a subset of the scale and offset parameters utilized in the backbone normalization layers (e.g. GroupNorm, LayerNorm, etc., see Appendix A.3.1 for details), which can update during fine-tuning. For example, in a ResNet50, there are only 11 648 learnable FiLM parameters, which is fewer than 0.05% of $\boldsymbol{\theta}$.

**Transfer Difficulty (TD)** The overlap between the distributions of the pretraining data and the downstream dataset as well other factors such as the number of classes in the downstream dataset are key determinants of the ease and success of transfer learning. We measure the *transfer difficulty* (TD) as the relative difference between the accuracy of the *All* and *Head* learnable parameter configurations for a non-private model: $TD = 100\,(Acc_{All} - Acc_{Head})\,/Acc_{All}$. This simple metric captures how different the downstream dataset is from the pretraining data as well as other factors that complicate transfer learning such as the number of classes $C$ in the downstream dataset and its size $|\mathcal{D}|$. If transfer learning is easy (i.e. TD is low), then only adapting the head of the network is sufficient. If transfer learning is more difficult (i.e. TD is high), then the backbone must also be adapted. Table 1 provides the TD values for all of the datasets used in the paper.

**Differential Privacy (DP)** DP (Dwork et al., 2006) is the gold standard for protecting sensitive data against privacy attacks. A stochastic algorithm is differentially private if it produces similar output distributions

on similar datasets. More formally, $(\epsilon, \delta)$-DP with privacy budget $\epsilon \geq 0$ (lower means more private) and additive error $\delta \in [0, 1]$ bounds how much the output distribution can diverge on adjacent datasets. We use *add/remove* adjacency, where two datasets are adjacent if one can be obtained from the other by adding or removing one data record, which could be a single datapoint in case of example-level privacy or data belonging to a single user in case of user-level privacy. The additive error is typically chosen such that $\delta < 1/|\mathcal{D}|$. We refer to Dwork & Roth (2014) for a thorough introduction to DP.

DP-SGD (Rajkumar & Agarwal, 2012; Song et al., 2013; Abadi et al., 2016) adapts stochastic gradient descent (SGD) to guarantee DP. DP-SGD selects mini-batches using Poisson sampling, clips the $\ell_2$ norm of per-example gradients, and adds isotropic Gaussian noise to the sum of mini-batch gradients. The level of privacy $((\epsilon, \delta)$-DP) is controlled by the noise multiplier $\sigma^2$ which scales the variance of the added noise, the number of steps, and the sampling ratio (the Poisson sampling probability, i.e., expected batch size$/|\mathcal{D}|$).

**Membership Inference Attacks (MIAs)** MIAs aim to determine if a particular example was used in the training set of a model (Shokri et al., 2017). MIAs can be used to derive lower bounds to complement the theoretical upper bounds of $(\epsilon, \delta)$-DP for trained models. While there are many types of MIA (Hu et al., 2022b), in this work we consider attacks that operate in the black-box mode (i.e. only model outputs can be observed) and can evaluate the loss on particular training or test examples (Carlini et al., 2022; Ye et al., 2022). In addition, we assume that attacks have access to images from the training data distribution and know the training algorithm used and its hyperparameters. To evaluate the effectiveness of a MIA, we examine the Receiver Operating Characteristic (ROC) curve which plots the attack true positive rate (TPR) against its false positive rate (FPR). We focus on the TPR at low FPR regime since a MIA is harmful if it can infer membership of even a small number of training examples with high confidence (Carlini et al., 2022).

## 3 Related Work

**DP Transfer Learning** Section 1 describes various works where DP transfer learning using models pretrained on large public datasets achieve accuracy close to non-private approaches. However, to the best of our knowledge, there are no comprehensive studies on few-shot transfer learning under DP. The closest work to ours is Luo et al. (2021) where the authors evaluate DP fine-tuning of a sparse subset of the parameters of models pretrained on public data on a small number of few-shot downstream datasets. Their work employs a relatively small backbone (ResNet18), pretrained on a small public dataset (miniImageNet), with limited analysis. In contrast, our work utilizes large backbones, a large public pretraining set, a wider range of privacy levels and downstream datasets, in addition to assessing vulnerability to attacks and the federated learning setting. Tramèr et al. (2022) point out that current DP benchmarks rely excessively on downstream datasets with a high level of overlap with the pretraining data. Our work addresses this issue by evaluating on datasets with a wide range of TD.

**Federated Learning (FL) and Transfer Learning** There has been a recent surge of interest in using large pretrained models as initialization for training decentralized models in both NLP (Lin et al., 2022; Stremmel & Singh, 2021; Weller et al., 2022; Tian et al., 2022) and computer vision (Chen et al., 2022b; Tan et al., 2022; Qu et al., 2021; Chen et al., 2022a; Nguyen et al., 2022; Liu et al., 2022). Most of these works were able to improve upon state-of-the-art results under different tasks and settings within FL as well as showing that the client data heterogeneity problem often seen in FL can be partially mitigated with pretrained networks.

**FL and DP** Even though the server in FL does not have access to raw user data, the privacy of users may still be compromised if (i) the server is untrusted (Huang et al., 2021) or (ii) a third party has access to the model after training (Geiping et al., 2020; Carlini et al., 2022). Cryptographic techniques like secure aggregation Goryczka et al. (2013) can partially mitigate the former issue, while to fully tackle it as well as the latter, DP adaptations of the FL aggregation algorithms are needed McMahan et al. (2018). Similarly to DP-SGD, DP-FedAvg (McMahan et al., 2018) is an adaptation of the baseline FL algorithm FedAvg (McMahan et al., 2017), which provides user-level DP guarantees by applying the Gaussian mechanism to parameter updates sent to the server. Recently, a few studies have investigated the use of large pretrained models for FL under DP constraints in NLP Basu et al. (2021), representation learning Xu et al. (2022), and image classification Song et al. (2022). The closest work to ours is Song et al. (2022) who introduce FLAIR, a few-shot federated learning image classification dataset, which they use to perform a relatively small evaluation

of pretrained models (only ResNet18 was used) fine-tuned using FL under DP. However, to the best of our knowledge, there are no other studies on how large pretrained models fine-tuned via FL aggregation algorithms behave under DP constraints for transfer-learned image classification. In this work we aim to fill this gap and evaluate these methods on real-world datasets.

## 4 Centralized Learning Experiments

In our experiments, we endeavor to answer the question: "Under what conditions is differentially private few-shot image classification effective?" We focus on transfer learning approaches that utilize large backbones pretrained on public data. We do this empirically by varying the: (i) number of shots $S$; (ii) set of learnable parameters in $f$ (*All*, *Head*, *FiLM*); (iii) downstream dataset $\mathcal{D}$ (with varying TD); and (iv) network architecture: BiT-M-R50x1 (R-50) (Kolesnikov et al., 2020) with 23.5M parameters, Vision Transformer VIT-Base-16 (VIT-B) (Dosovitskiy et al., 2021) with 85.8M parameters, both pretrained on the ImageNet-21K dataset. In all experiments, we assume that the pretraining data is public and the downstream data is private. Source code for all experiments can be found at: `https://github.com/cambridge-mlg/dp-few-shot`.

**Datasets** For the experiments where $S$ is varied, we use the CIFAR-10 (low TD) and CIFAR-100 (medium TD) datasets (Krizhevsky, 2009) which are commonly used in DP transfer learning, and SVHN (Netzer et al., 2011) which has a high transfer difficulty and hence requires a greater degree of adaptation of the pretrained backbone. We also evaluate on the challenging VTAB-1k transfer learning benchmark (Zhai et al., 2019) that consists of 19 datasets grouped into three distinct categories (natural, specialized, and structured) with training set size fixed at $|\mathcal{D}| = 1000$ and widely varying TD.

**Training Protocol** For all centralized experiments, we first draw $\mathcal{D}$ of the required size ($|\mathcal{D}| = CS$ (i.e. the number of classes $C$ multiplied by shot $S$) for varying shot or $|\mathcal{D}| = 1000$ for VTAB-1k) from the entire training split of the current dataset under evaluation. For the purposes of hyperparameter tuning, we then split $\mathcal{D}$ into 70% train and 30% validation. We then perform 20 iterations of Bayesian optimization based hyperparameter tuning (Bergstra et al., 2011) with Optuna Akiba et al. (2019) to derive a set of hyperparameters that yield the highest accuracy on the validation data. This set of parameters is subsequently used to train a final model on all of $\mathcal{D}$. We evaluate the final, tuned model on the entire test split of the current dataset. Details on the set of hyperparameters that are tuned and their ranges can be found in Appendix A.3.2.

For DP fine-tuning on $\mathcal{D}$, we use Opacus (Yousefpour et al., 2021) and compute the required noise multiplier depending on the targeted $(\epsilon, \delta)$. We report the results over three runs. For all experiments, we set $\delta = 1/|\mathcal{D}|$ and report $(\epsilon, \delta)$-DP computed with the RDP accountant (Mironov, 2017). Note that because we often change the dataset size $|\mathcal{D}|$ in our experiments this may make certain comparisons difficult since $\delta$ will also vary. Similarly to previous work (De et al., 2022; Mehta et al., 2022; Sander et al., 2022) we do not account for privacy loss originating from the tuning of the hyperparameters. See Appendix A.3 for additional training details.

### 4.1 Few-shot DP Data Requirements

Fig. 1 depicts the performance of transfer learning under DP when varying $S$, $\epsilon$, and TD. Tabular results can be found in Tables 2 to 7. We see that accuracy decreases as $S$ and $\epsilon$ decrease and as TD increases. For $S \leq 10$, accuracy is poor under DP. However, if the TD is low or medium, a moderate number of shots ($S \approx 100$) is sufficient to approach the accuracy of the non-private setting. For example, at $S = 100$, the model achieves better than 90% accuracy on CIFAR-10 using only 2% of the full training split at $\epsilon = 1$. On the other hand, if TD is high, learning is more challenging and more shots are required to approach non-private accuracy. For example, for $S = 100$ and $\epsilon = 2$, SVHN achieves just over 20% accuracy and falls well short of non-private levels even at $S = 500$. Note that $\delta$ changes based on $|\mathcal{D}|$. Tables 23 and 24 provide $(\epsilon, \delta)$-DP guarantees computed for when $\delta$ is fixed and thus independent of $|\mathcal{D}|$.

Fig. 2 shows the multiplier on the number of DP shots to match non-private accuracy (see Appendix A.2.2 for additional figures and details). On the left we average over $S \in \{5, 10\}$, datasets, and network architectures. For all configurations, at $\epsilon = 8$, $S$ must be increased by approximately $4 - 8\times$ to meet non-private accuracy and $20 - 35\times$ at $\epsilon = 1$. In effect, as the privacy level increases, the required multiplier increases in an

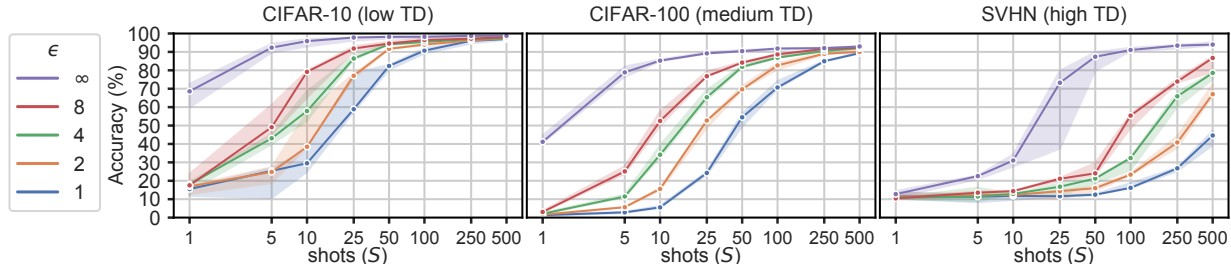

Figure 1: Classification accuracy as a function of shots and $\epsilon$ for CIFAR-10, CIFAR-100 and SVHN. Backbone is VIT-B and the best performing configuration out of *All*, *FiLM* and *Head* is used for each combination of $\epsilon$ and $S$, with $\delta = 1/|\mathcal{D}|$. The accuracy is reported over three seeds with the line showing the median and the band reporting the lowest and highest accuracy. Analysis: Classification accuracy decreases as $S$ and $\epsilon$ decrease and TD increases.

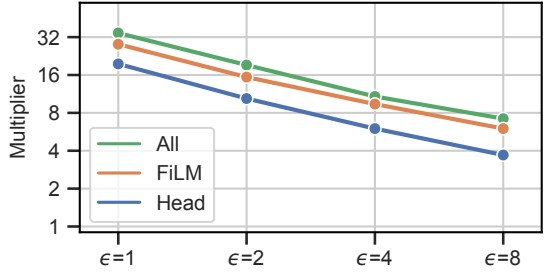 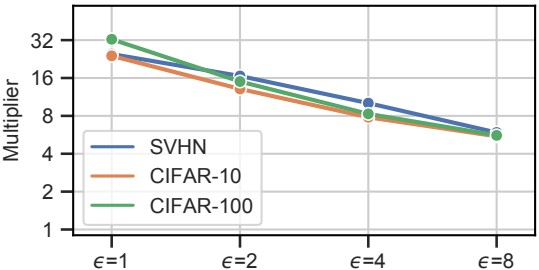

Figure 2: Multiplier of shots required to reach non-private accuracy. Left: Average over $S \in \{5, 10\}$, datasets, and network architectures. Right: Average over $S \in \{5, 10\}$, network architectures, and learnable parameters. The data is obtained using linear interpolation. See Appendix A.2.2. $\delta = 1/|\mathcal{D}|$. Analysis: $S$ must be increased by approximately $20 - 35\times$ to meet non-private accuracy at $\epsilon = 1$ and $4 - 8\times$ at $\epsilon = 8$

exponential manner. The multipliers are lower for simpler forms of adaptation (e.g. *Head* requires $20 \times S$ at $\epsilon = 1$) than for more complex forms (e.g. *All* requires $35 \times S$ at $\epsilon = 1$). On the right we average over $S \in \{5, 10\}$, network architectures, and learnable parameters. Even though high TD datasets require more data for good accuracy, the multiplier values are similar and independent of the TD of the dataset (around $30\times$ at $\epsilon = 1$ and $6\times$ at $\epsilon = 8$). Fig. 3 shows the classification accuracy as a function of TD at $S = 100$. The accuracy gap between non-private and private training increases as TD increases.

## 4.2 Characterization of Learning Under Few-Shot DP

In this section, we provide empirical evidence to highlight the different traits of private and non-private learning. Fig. 4 shows snapshots at $\epsilon \in \{1, 8, \infty\}$ of the train and test accuracies as a function of $S$ for CIFAR-100 (medium TD) and SVHN (high TD) (see Figs. 14 and 15 for versions with additional values of $\epsilon$). The three snapshots for each dataset can be viewed as discrete points on a continuum from low to high $\epsilon$. We see that learning under DP is fundamentally different from non-private. Non-private models with sufficient capacity operate in the interpolating regime and attain close to 100% training accuracy at all values of $S$, but have substantially lower test accuracy when $S$ is low. In contrast, models that are are trained with DP-SGD are learning under heavy regularization and thus the training and test accuracies are significantly lower, but similar in value. When $S$ is low, test accuracy is relatively poor and as $S$ increases, test accuracy steadily improves. The point at which accuracy begins to improve varies with TD (CIFAR-100 test accuracy improvement starts much earlier than for SVHN). Independent of $S$, for low $\epsilon$, the train-test gap is very small, with the train accuracy indicative of the test performance. As $\epsilon$ increases, the train accuracy grows as the amount of regularization pressure from DP is reduced, ultimately entering the interpolating regime. For SVHN with $\epsilon = \infty$, *Head* leaves the interpolating regime for $S > 100$, as there is not enough capacity to adapt to a high TD dataset. As $\epsilon$ increases and $S$ remains low, the test accuracy does not increase as quickly

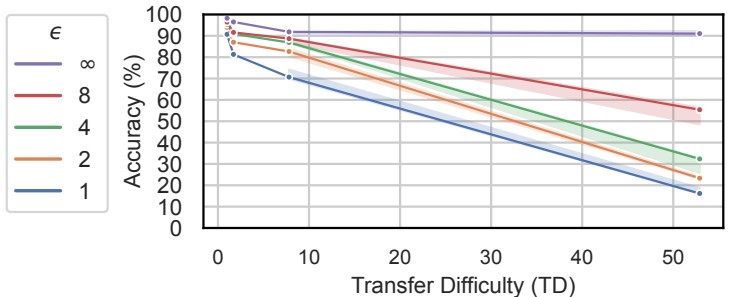

Figure 3: Classification accuracy as a function of transfer difficulty (TD) and $\epsilon$ for CIFAR-10 (TD= 1.0), EuroSAT (TD=1.7), CIFAR-100 (TD=7.8) and SVHN (TD= 52.9) at $S = 100$. EuroSAT has been chosen because the result for $S = 100$ can be easily taken from the VTAB results (Tables 8 to 13) due to $C = 10$. Backbone is VIT-B and the best performing configuration out of *All*, *FiLM* and *Head* is used for each $\epsilon$, with $\delta = 1/|\mathcal{D}|$. The accuracy is reported over three seeds with the line showing the median and the band reporting the lowest and highest accuracy. Analysis: The accuracy gap between non-private and private training increases as TD increases.

as the train accuracy and the accuracy gap grows. However, as $S$ increases, test accuracy starts to catch up with train accuracy, reducing the gap. Fig. 21 shows the train and test accuracies for all 19 VTAB datasets as a function of $\epsilon$, where the general trends noted in Fig. 4 are also evident. While the results of Section 4.1

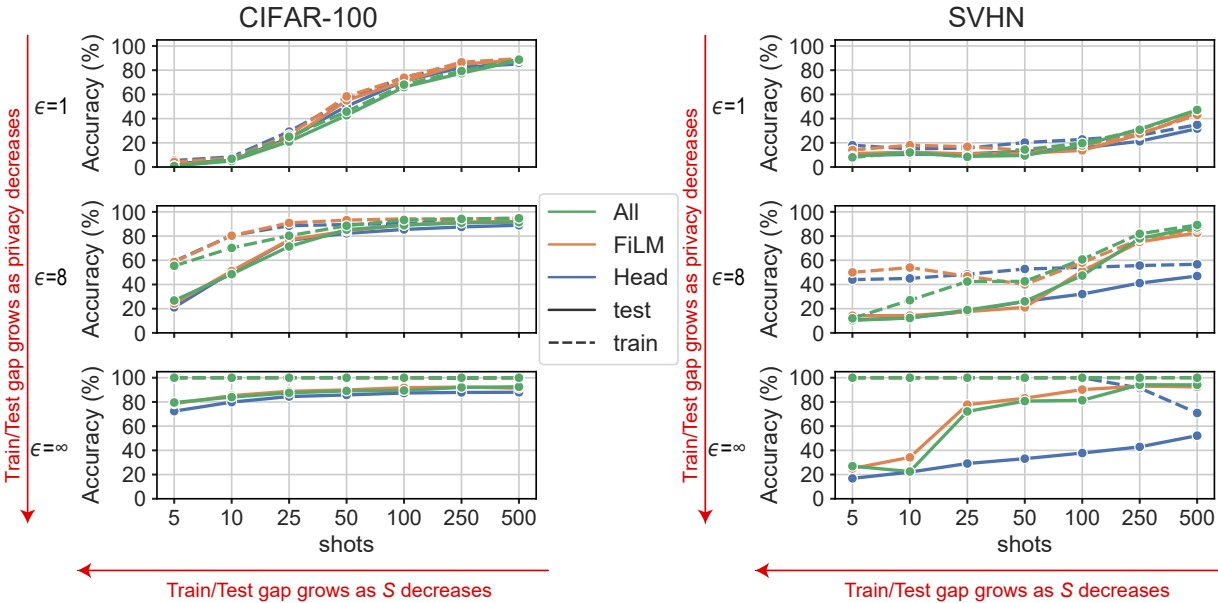

Figure 4: Snapshots at $\epsilon \in \{1, 8, \infty\}$, $\delta = 1/|\mathcal{D}|$ of the train/test accuracies as a function of $S$ for CIFAR-100 and SVHN. The trends in accuracy gap are shown with red arrows. At low $S$, the gap grows as $\epsilon$ increases, at high $S$ gap decreases to nearly zero, and the gap grows as $S$ decreases. Analysis: In the non-private setting ($\epsilon = \infty$), learning operates in the interpolation mode (i.e. train accuracy is 100%, yet accuracy continues to increase as $S$ increases). As the privacy level increases, learning operates learn under heavy regularization and the gap between train and test accuracy reduces.

indicate that both private and non-private test accuracy benefit from additional training data, it is evident that their learning behavior is significantly different.

## 4.3 Few-shot DP Model Parameterization

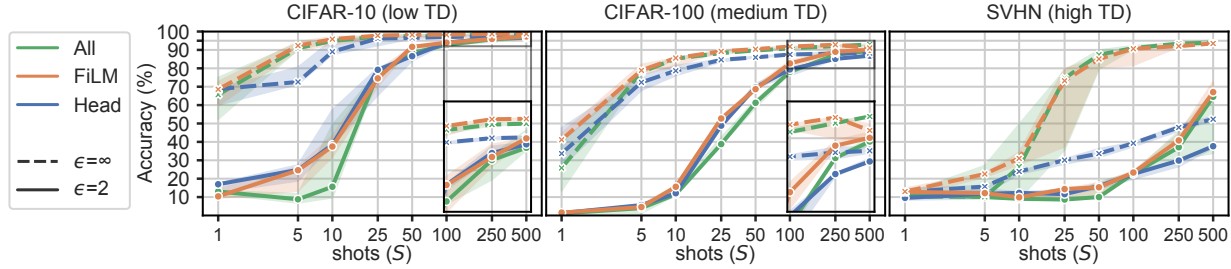

Figure 5: Classification accuracy as a function of shots and learnable parameters on VIT-B for CIFAR-10, CIFAR-100 and SVHN for $\epsilon \in \{2, \infty\}$ with $\delta = 1/|\mathcal{D}|$. The accuracy is reported over three seeds with the line showing the median and the band reporting the lowest and highest accuracy. Analysis: *FiLM* is comparable to or better than *All* and *Head* in terms of accuracy despite fine-tuning fewer than 0.05% of the parameters in the backbone.

Fig. 5 depicts classification accuracy as a function of $S$, two different values of $\epsilon$, and learnable parameters. *FiLM* is comparable to or better than *All* and *Head* in terms of accuracy despite fine-tuning fewer than 0.05% of the parameters in the backbone. When the TD is low, training only *Head* is competitive with *FiLM* and *All*, but when TD is medium or high, *Head* falls short as it cannot adapt the backbone to a dataset that has a different data distribution. These observations have two implications: (i) *FiLM* is able to adapt to differing downstream datasets under DP and serves as a computationally efficient alternative to *All*; (ii) The result provides empirical support for the observations of Li et al. (2022a) that the number of parameters has little effect on the privacy utility trade-off when fine-tuning large pretrained models. Prior theory (Chaudhuri et al., 2011; Bassily et al., 2014) suggested that *All* should perform worse under DP compared to configurations with fewer parameters.

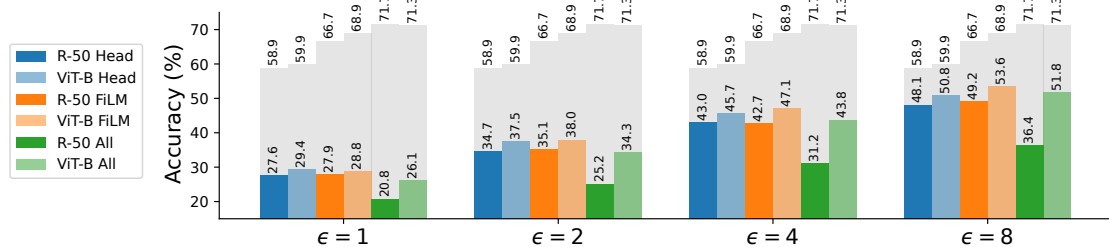

Figure 6: Average classification accuracy over all VTAB-1k datasets as a function of backbone, learnable parameters, and privacy level ($\epsilon$) at $\delta = 10^{-3}$. Colored columns indicate results under DP, light gray indicates non-private accuracy for the corresponding configuration. Analysis: DP classification accuracy (colored columns) decreases significantly as $\epsilon$ is decreased and always falls short of non-private accuracy (gray columns). For non-private settings, the *All* learnable parameters setting outperforms *FiLM* which outperforms *Head*. In contrast, for DP settings, *All* performs worst, *FiLM* and *Head* perform similarly, though *FiLM* is better in the majority of cases.

Fig. 6 shows average classification accuracy over all of the datasets in the VTAB-1k benchmark (tabular results are in Tables 8 to 13 and comprehensive graphical results are in Figs. 19 and 20). We see that DP classification accuracy decreases significantly as $\epsilon$ is decreased and always falls short of non-private accuracy. For non-private settings, the *All* learnable parameters setting outperforms *FiLM* which outperforms *Head*. In contrast, for DP settings, *All* performs worst, *FiLM* and *Head* perform similarly, though *FiLM* is better in the majority of cases. One explanation for this is that under DP at low $S$, *All* requires more data compared to *Head* and *FiLM* for accuracy to progress beyond random chance as can be seen in Fig. 5.

Fig. 7 shows the difference between the accuracy of *FiLM* and *Head* for VTAB-1k datasets as a function of $\epsilon$. The datasets are ordered from low to high TD (see Table 1). At $\epsilon = 1$, *Head* has an advantage over *FiLM* on several datasets. *FiLM* shows a significant advantage when the TD increases and as $\epsilon$ increases. Refer to Appendix A.2.5 for additional heat maps.

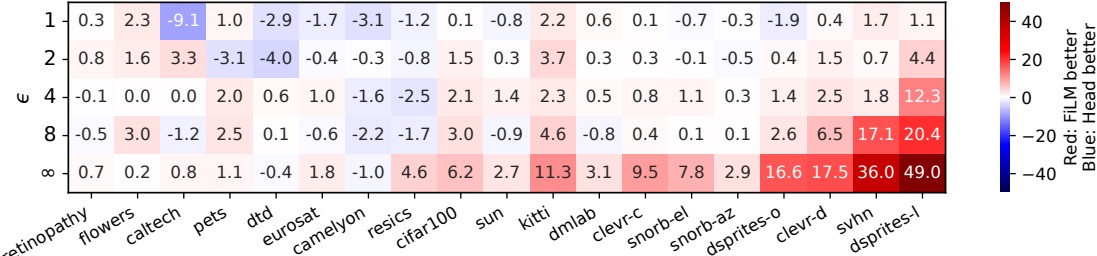

Figure 7: Heat map showing the accuracy difference between *FiLM* and *Head* for the VTAB-1k datasets as a function of $\epsilon$. Backbone is VIT-B. Darker red indicates *FiLM* is better. Darker blue indicates *Head* is better. Datasets ordered from low to high TD. $\delta = 10^{-3}$. Analysis: At $\epsilon = 1$, *Head* has an advantage over *FiLM* on several datasets. *FiLM* shows a significant advantage when the TD increases and as $\epsilon$ increases.

### 4.4 Membership Inference Attacks

We use the state-of-the-art Likelihood Ratio Attack (LiRA) (Carlini et al., 2022) to attack models trained on CIFAR-100 with varying $S$ and privacy level $\epsilon$ using 256 shadow models. Refer to Appendix A.3.5 for additional detail. Excerpts from attack results are shown in Fig. 8. The complete set of attack ROC curves are shown in Figs. 22 and 23, while Table 14 reports TPR at several low FPR values, AUC score, and the maximum membership inference advantage (defined as TPR - FPR by Yeom et al. (2018)) achieved over the curve. Key observations are:

- Non-private ($\epsilon = \infty$) models are extremely vulnerable to MIAs (see Fig. 8, middle). For example, in the case of $\epsilon = \infty$, $S = 10$, *Head* configuration, 82.2% of the examples can be successfully identified with a false positive rate of only 0.1%.

- Vulnerability of non-private ($\epsilon = \infty$) models decreases as $S$ increases. Also, the *FiLM* configuration is consistently less vulnerable than *Head* (see Fig. 8, middle). We hypothesize that *FiLM* generalizes better, so training examples do not stand out as much as in the *Head* configuration.

- When $S$ is fixed, vulnerability to MIAs greatly decreases with decreasing $\epsilon$ (see Fig. 8, right). Already with $\epsilon = 2$, when $S = 10$ and *FiLM* the vulnerability to MIA is substantially reduced, 2.5% of the examples can be successfully identified with an FPR of 1% and 0.3% of the examples with 0.1% FPR (see Table 14).

- Under DP, there appears to be little or no difference between the vulnerability of the *FiLM* and *Head* configurations at the same $\epsilon$ (see Fig. 8, right).

## 5 Federated Learning Experiments

In this section, we investigate how imposing user-level DP influences the performance of large pretrained models fine-tuned via federated aggregation. In our evaluation, we use FLAIR Song et al. (2022), which is a recently proposed real-world dataset for multi-label image classification. It has around 50k users (overall around 400k images) with heterogeneous data as well as a long-tailed label distribution, making it particularly appealing for benchmarking federated learning both in non-private and private settings. Comprising mainly natural image data, FLAIR is a low to medium TD dataset. As in (Song et al., 2022), $\delta$ is set to $N^{-1.1}$, where $N = 41131$ is the number of training clients, and $\epsilon = 2$. We also perform experiments on CIFAR-100 and Federated EMNIST, which have many fewer training users, but are widely used for benchmarking federated learning. Those results are in Appendix A.2.8.

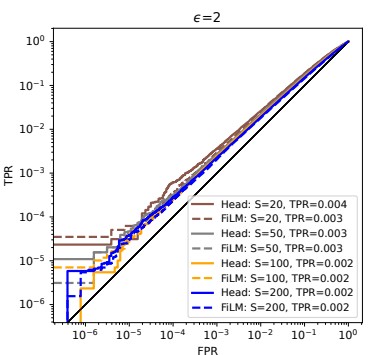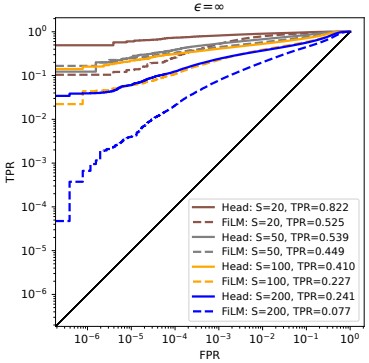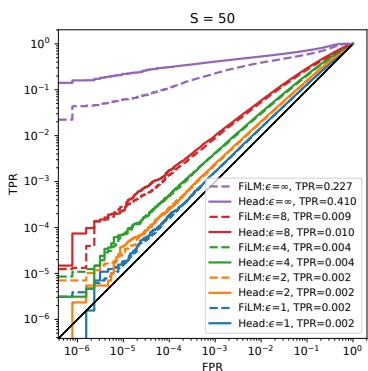

Figure 8: ROC curves for LiRA (Carlini et al., 2022) on CIFAR-100 with R-50 backbone for two values of $\epsilon$ (2 and $\infty$) where $S$ varies and for $S = 50$ where $\epsilon$ varies. TPR values in legends are measured at FPR=0.001. Complete results in Table 14 and Figs. 22 and 23. $\delta = 1/(100S)$. Analysis: Middle - Non-private models are extremely vulnerable to MIAs. For $\epsilon = \infty$, $S = 10$, *Head* configuration, 82.2% of the examples can be successfully identified with FPR = 0.1%. Also, vulnerability decreases as $S$ increases. Right: Increasing the privacy level reduces the vulnerability of the model as expected, when $S = 10$ with $\epsilon = 2$ and *FiLM*, only 2.5% of the examples can be successfully identified with a FPR of 1%.

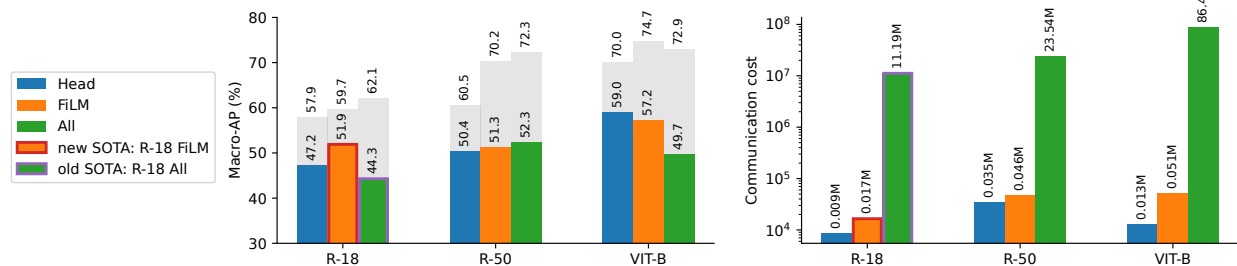

Figure 9: Left: Private (colored) and non-private (gray) FL performance on FLAIR as a function of backbone and learnable parameters. $\epsilon = 2, \delta = 41131^{-1.1}$. We use Macro-AP as the primary metric to report accuracy for FLAIR. The R-18 *All* result on FLAIR is taken from Song et al. (2022). Our FLAIR results set a new state-of-the-art. Right: FLAIR communication cost – the number of parameters sent at every user-server communication round. Analysis: We set a new state-of-the-art result on the FLAIR benchmark by using *FiLM* on R-18 by increasing Macro-AP from 47.2% to 51.9% and drastically reducing the communication cost from 11.9M parameters per round to 17K parameters. We further improve those results by using bigger backbones (R-50, VIT-B) with corresponding decreases in communication cost.

As in the centralized experiments, we use R-50 and VIT-B, both pretrained on ImageNet-21K. We also perform experiments on a smaller architecture, ResNet18 (R-18) (He et al., 2016) pretrained on ImageNet-1K with 11.2M parameters, as it was initially used to achieve SOTA results on FLAIR.

For FL experiments, user-level DP is considered. We use FedADAM (Reddi et al., 2021) aggregation, which was shown to have better empirical performance than standard FedAvg (McMahan et al., 2017). We do not use Bayesian optimization for hyperparameter tuning, as each FL run is prohibitively expensive. Instead, we perform a small grid search over the server and client learning rates. Refer to Appendix A.3.6 for the hyperparameter ranges searched. For fair comparison on FLAIR, we fixed the other training hyperparameters to the values in the original paper Song et al. (2022).

Fig. 9 (left) shows the performance of different model configurations on FLAIR with (color) and without (grey) DP. We report macro average precision (Macro-AP) results here, while additional metrics are shown in Tables 15 and 16. As communication cost is important in FL, in Fig. 9 (right) we report the number of parameters required to be transmitted for each model configuration in one user-server interaction. Summarizing Fig. 9, key observations are:

- With R-18 as used in the original paper, we achieve state-of-the-art performance under DP with *FiLM*, improving Macro-AP from 44.3% to 51.9%. This improvement comes with a reduction in communication cost from 11.2M parameters per each user-server interaction to only 17k.

- With VIT-B we further improve the state-of-the-art result on FLAIR in both DP and non-private settings. Under DP, the Macro-AP increases to 59%, while for non-private, the Macro-AP increases from 62.1% to 74.7%.

- *Head* is more robust under DP than *All* or *FiLM*. *Head* has the smallest relative drop in performance of around 10% for any model configuration.

- Although *FiLM* is outperforming *Head* and *All* on R-18, it is not always the case for other backbones. However, taking into account both test performance and communication cost, we can clearly see that either *FiLM* or *Head* is preferred. *FiLM* performs better for smaller backbones (R-18 and R-50), while *Head* is slightly better for VIT-B.

## 6 Discussion and Recommendations

Our work shows that DP few-shot learning works surprisingly well in the low TD setting, while the high TD setting is more difficult. Alternative strategies may include side-stepping privacy costs by leveraging the zero-shot capabilities of large pretrained models such as CLIP (Radford et al., 2021) or utilizing public data in addition to private data in the fine-tuning process as well (Golatkar et al., 2022) in order to improve utility. In summary, our experiments show that:

- How much additional data is required under few-shot DP? Image classification accuracy decreases as $\epsilon$ and $S$ decrease, and as TD increases. As a result, one should expect to use roughly $4-8\times$ larger $S$ for $\epsilon = 8$ and $20-35\times$ larger $S$ for $\epsilon = 1$ under DP to achieve accuracy comparable to non-private. (Note that $\delta = 1/|D|$ for these multipliers.) The multipliers are surprisingly similar across different TD levels.

- **Transfer learning dynamics under DP are fundamentally different from non-private** Non-private models with sufficient capacity operate in the interpolating regime and attain close to 100% training accuracy at all values of $S$, but have substantially lower test accuracy when $S$ is low. In contrast, models that are trained with DP-SGD are learning under heavy regularization and thus the training and test accuracies are significantly lower, but similar in value.

- **Parameter-efficient *FiLM* adapters perform well under DP** *FiLM* is comparable to or better than *All* and *Head* in terms of accuracy, demonstrating its ability to adapt to differing downstream datasets despite fine-tuning fewer than 0.05% of the parameters in the backbone. When the TD is easy, *Head* is competitive with *FiLM* and *All*, but when TD is difficult, *Head* falls short as it cannot adapt the backbone to a downstream dataset that has a different data distribution. *FiLM* is also effective in the DP FL setting, achieving state-of-the-art accuracy on the FLAIR benchmark while reducing communication cost by orders of magnitude.

- **Non-private Few-Shot Models Are Particularly Vulnerable to MIAs** The vulnerability of non-private few-shot models increases as $S$ decreases. DP significantly mitigates the effectiveness of MIAs, e.g., we found that DP few-shot models can expose 2.5% of the examples with a 1% FPR when $\epsilon = 2$ (on CIFAR-100 with $S$=10, *FiLM* on R-50) which is substantially less vulnerable than the non-private models.

**Limitations** We identify the following limitations in this work: (i) We focused exclusively on few-shot transfer learning from relatively large pretrained models and did not consider meta-learning approaches or training from scratch. (ii) We used FiLM adapters exclusively and did not consider other parameter-efficient adapters. Based on our experience, adapters do not have a large effect on the overall trends that we observed (in comparison to the items that we did vary), and making a fair comparison on a reasonable set of adapters would have exceeded our computational resources. (iii) The Transfer Difficulty (TD) metric is not ideal as it depends on the network architecture and training hyperparameters, but in practice it aligns extremely well with the empirical difficulty of adapting to a downstream dataset. (iv) We always set $\delta = 1/\mathcal{D}$. While this is standard practice, in some experiments in the paper where $\mathcal{D}$ varies, fair comparisons of results can be difficult and in that case an alternative would have been to choose a small constant value of $\delta$. (v) For each

experiment, we used hyperparameter tuning to set a constant learning rate, but this learning rate was not annealed as is often done in non-private training.

**Broader Impact Statement** Few-shot learning systems hold much positive potential – from personalizing object recognizers for people who are blind (Massiceti et al., 2021) to rendering personalized avatars (Zakharov et al., 2019) (see (Hospedales et al., 2020) for a full review). These systems, however, also have the potential to be used in adverse ways – for example, in few-shot recognition in military/surveillance applications. We demonstrate how to execute highly successful membership inference attacks against few-shot learning models which could be employed in a harmful manner. However, we also show how to effectively mitigate such attacks by training with DP.

## Acknowledgments

Marlon Tobaben and Antti Honkela are supported by the Research Council of Finland (Flagship programme: Finnish Center for Artificial Intelligence, FCAI; and grant 356499), the Strategic Research Council at the Research Council of Finland (Grant 358247) as well as the European Union (Project 101070617). Views and opinions expressed are however those of the author(s) only and do not necessarily reflect those of the European Union or the European Commission. Neither the European Union nor the granting authority can be held responsible for them. Aliaksandra Shysheya, John Bronskill, and Richard E. Turner are supported by an EPSRC Prosperity Partnership EP/T005386/1 between the EPSRC, Microsoft Research and the University of Cambridge. This work has been performed using resources provided by the CSC – IT Center for Science, Finland, and the Finnish Computing Competence Infrastructure (FCCI), as well as the Cambridge Tier-2 system operated by the University of Cambridge Research Computing Service `https://www.hpc.cam.ac.uk` funded by EPSRC Tier-2 capital grant EP/P020259/1. We thank Joonas Jälkö, Lukas Wutschitz, Stratis Markou, Massimiliano Patacchiola and Runa Eschenhagen for helpful comments and suggestions.

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

# A Appendix

## A.1 Transfer Difficulty

Table 1 shows the transfer difficulty for each of the datasets used in our experiments.

Table 1: Transfer difficulty for the 19 VTAB-1k datasets (plus CIFAR-10). The Score column is computed as the difference between the accuracy of the *All* learnable parameter configuration and the *Head* configuration, normalized by the *All* accuracy, and then scaled by 100. The lower the score, the lower the difficulty of the transfer. In the TD column, we map the score into three buckets: a score of 0-5 is low, 5-10 is medium, and greater than 10 is high. To compute the scores, we use the VIT-B backbone and use accuracies from the VTAB-1k experiments for the VTAB-1k datasets and the accuracies from the results of the effect of shots and DP experiments at $S = 100$ for CIFAR-10.

| Dataset | Score | TD |
|---|---|---|
| Caltech101 (Fei-Fei et al., 2006) | 0.4 | low |
| CIFAR10 (Krizhevsky, 2009) | 1.0 | low |
| CIFAR100 (Krizhevsky, 2009) | 7.8 | medium |
| Flowers102 (Nilsback & Zisserman, 2008) | 0.2 | low |
| Pets (Parkhi et al., 2012) | 1.1 | low |
| Sun397 (Xiao et al., 2010) | 8.8 | medium |
| SVHN (Netzer et al., 2011) | 52.9 | high |
| DTD (Cimpoi et al., 2014) | 1.3 | low |
| EuroSAT (Helber et al., 2019) | 1.7 | low |
| Resics45 (Cheng et al., 2017) | 6.7 | medium |
| Patch Camelyon (Veeling et al., 2018) | 3.8 | low |
| Retinopathy (Kaggle & EyePacs, 2015) | 0.2 | low |
| CLEVR-count (Johnson et al., 2017) | 26.2 | high |
| CLEVR-dist (Johnson et al., 2017) | 38.7 | high |
| dSprites-loc (Matthey et al., 2017) | 71.4 | high |
| dSprites-ori (Matthey et al., 2017) | 37.7 | high |
| SmallNORB-azi (LeCun et al., 2004) | 33.3 | high |
| SmallNORB-elev (LeCun et al., 2004) | 28.2 | high |
| DMLab (Beattie et al., 2016) | 21.9 | high |
| KITTI-dist (Geiger et al., 2013) | 13.6 | high |

## A.2 Additional Results

### A.2.1 Additional Effect of Shots per Class and $\epsilon$ Results

Tables 2 to 7 depict tabular results for different backbones (R-50, VIT-B), different learnable parameter sets (*Head*, *FiLM*, *All*), different numbers of shots per class ($S = 1, 5, 10, 25, 50, 100, 250, 500$) and various privacy levels ($\epsilon = 1, 2, 4, 8, \infty$), all at $\delta = 1/|\mathcal{D}|$).

Table 2: Classification accuracy as a function of $\epsilon$, $S$ and learnable parameters for CIFAR-10. Backbone is R-50 pretrained on ImageNet-21k. Accuracy figures are percentages and the $\pm$ sign indicates the 95% confidence interval over 3 runs with different seeds.

| | | $1S$ | $5S$ | $10S$ | $25S$ | $50S$ | $100S$ | $250S$ | $500S$ |
|---|---|---|---|---|---|---|---|---|---|
| | $\epsilon = 1$ | 10.0±0.0 | 9.8±0.3 | 9.9±0.2 | 29.3±4.3 | 54.1±7.4 | 74.3±1.2 | 86.1±1.7 | 90.5±0.7 |
| | $\epsilon = 2$ | 9.8±0.3 | 14.0±7.9 | 9.8±0.2 | 50.2±3.5 | 71.2±2.3 | 83.3±1.9 | 90.4±0.6 | 91.6±0.1 |
| All | $\epsilon = 4$ | 10.0±0.1 | 9.8±0.3 | 9.8±0.3 | 69.2±4.1 | 81.9±3.1 | 88.2±1.0 | 91.5±0.3 | 92.0±0.3 |
| | $\epsilon = 8$ | 10.0±0.0 | 20.1±19.7 | 28.0±35.9 | 53.0±26.2 | 85.9±2.0 | 90.1±0.8 | 91.8±0.5 | 92.8±0.8 |
| | $\epsilon = \infty$ | 51.3±6.0 | 78.7±3.0 | 86.3±0.7 | 89.3±1.1 | 91.9±0.6 | 93.5±0.3 | 94.6±0.4 | 95.5±0.3 |
| | $\epsilon = 1$ | 11.1±0.6 | 18.8±3.5 | 21.4±4.4 | 45.6±3.4 | 68.3±3.9 | 78.3±4.6 | 89.6±0.9 | 92.4±0.8 |
| | $\epsilon = 2$ | 13.3±3.3 | 21.0±5.6 | 31.7±6.6 | 63.4±3.0 | 80.9±2.4 | 86.9±0.9 | 92.3±0.4 | 93.6±0.7 |
| FiLM | $\epsilon = 4$ | 14.9±4.3 | 30.3±8.5 | 52.4±3.4 | 76.1±1.5 | 86.2±1.7 | 89.1±1.2 | 93.4±0.0 | 94.4±0.3 |
| | $\epsilon = 8$ | 17.1±5.3 | 40.1±5.0 | 67.0±5.8 | 80.9±1.2 | 89.2±1.7 | 92.4±0.7 | 94.0±0.3 | 94.7±0.3 |
| | $\epsilon = \infty$ | 48.1±3.6 | 79.4±6.3 | 86.5±2.6 | 91.7±0.4 | 94.1±0.3 | 94.9±0.2 | 95.5±0.2 | 95.8±0.2 |
| | $\epsilon = 1$ | 11.7±5.0 | 18.7±1.6 | 20.5±8.3 | 44.7±4.7 | 68.7±3.3 | 82.0±1.6 | 87.3±0.2 | 88.9±0.8 |
| | $\epsilon = 2$ | 11.9±3.8 | 23.7±1.5 | 31.1±8.4 | 62.2±8.8 | 80.1±1.2 | 86.2±0.6 | 89.3±0.4 | 90.0±0.7 |
| Head | $\epsilon = 4$ | 13.1±2.6 | 30.0±5.2 | 45.5±12.1 | 79.4±1.8 | 84.7±0.7 | 87.7±0.5 | 89.8±0.2 | 90.8±0.4 |
| | $\epsilon = 8$ | 16.8±4.4 | 41.5±6.5 | 65.5±3.1 | 83.2±1.7 | 86.2±0.7 | 89.1±0.3 | 90.1±0.3 | 91.2±0.1 |
| | $\epsilon = \infty$ | 49.2±4.7 | 75.2±6.0 | 83.5±0.4 | 87.2±0.6 | 89.0±0.2 | 90.3±0.2 | 91.4±0.2 | 92.1±0.3 |

Table 3: Classification accuracy as a function of $\epsilon$, $S$ and learnable parameters for CIFAR-100. Backbone is R-50 pretrained on ImageNet-21k. Accuracy figures are percentages and the $\pm$ sign indicates the 95% confidence interval over 3 runs with different seeds.

| | | $1S$ | $5S$ | $10S$ | $25S$ | $50S$ | $100S$ | $250S$ | $500S$ |
|---|---|---|---|---|---|---|---|---|---|
| | $\epsilon = 1$ | 1.0±0.0 | 1.7±0.8 | 1.7±1.4 | 7.9±1.5 | 23.9±0.7 | 48.5±2.1 | 50.6±0.3 | 62.4±5.2 |
| | $\epsilon = 2$ | 1.3±0.6 | 1.2±0.6 | 6.8±0.2 | 23.1±0.6 | 46.0±2.1 | 59.4±2.9 | 61.9±6.0 | 65.7±5.2 |
| All | $\epsilon = 4$ | 1.0±0.0 | 1.6±1.3 | 15.3±1.1 | 43.7±3.9 | 57.6±2.8 | 64.8±0.4 | 63.0±0.9 | 68.1±0.8 |
| | $\epsilon = 8$ | 1.0±0.0 | 5.1±7.9 | 31.1±2.0 | 55.9±2.6 | 59.6±3.7 | 67.3±0.3 | 67.6±0.8 | 74.1±2.3 |
| | $\epsilon = \infty$ | 28.3±2.4 | 51.9±5.6 | 59.7±9.9 | 71.9±0.5 | 76.0±0.8 | 79.9±0.1 | 82.2±3.0 | 85.8±2.1 |
| | $\epsilon = 1$ | 1.0±0.4 | 1.8±0.5 | 3.7±0.2 | 14.2±0.2 | 34.2±1.8 | 59.2±1.0 | 75.3±0.7 | 80.1±0.8 |
| | $\epsilon = 2$ | 1.4±0.5 | 2.9±0.3 | 9.3±0.6 | 33.4±0.8 | 55.0±3.2 | 72.2±0.3 | 80.2±0.4 | 82.8±0.5 |
| FiLM | $\epsilon = 4$ | 1.7±0.4 | 7.2±1.3 | 22.4±1.2 | 53.9±0.9 | 70.1±0.6 | 77.9±0.6 | 82.3±0.4 | 84.2±0.5 |
| | $\epsilon = 8$ | 2.2±0.3 | 18.6±0.3 | 38.7±3.7 | 65.3±1.6 | 75.6±0.5 | 80.5±0.8 | 83.5±0.5 | 85.2±0.4 |
| | $\epsilon = \infty$ | 25.8±6.1 | 64.8±3.2 | 71.8±1.1 | 77.6±0.1 | 81.4±0.4 | 82.9±0.4 | 83.8±0.8 | 83.4±1.2 |
| | $\epsilon = 1$ | 1.3±0.6 | 2.3±0.9 | 4.4±1.2 | 13.5±1.5 | 32.3±0.9 | 52.1±0.7 | 66.9±0.8 | 71.9±0.7 |
| | $\epsilon = 2$ | 1.4±0.7 | 3.9±0.8 | 8.7±1.4 | 31.6±1.2 | 51.0±1.2 | 63.1±1.0 | 71.6±0.4 | 75.2±0.3 |
| Head | $\epsilon = 4$ | 1.8±0.6 | 7.5±1.8 | 22.1±1.3 | 46.3±0.7 | 61.7±1.6 | 68.4±0.8 | 74.4±0.3 | 77.0±0.1 |
| | $\epsilon = 8$ | 2.5±0.7 | 17.2±1.7 | 38.9±1.9 | 58.9±0.3 | 67.5±0.4 | 72.1±0.5 | 76.3±0.1 | 78.4±0.3 |
| | $\epsilon = \infty$ | 25.6±5.6 | 56.4±0.3 | 62.8±1.4 | 69.6±0.2 | 72.9±0.3 | 75.8±0.3 | 78.5±0.2 | 78.8±1.5 |

Table 4: Classification accuracy as a function of $\epsilon$, $S$ and learnable parameters for SVHN. Backbone is R-50 pretrained on ImageNet-21k. Accuracy figures are percentages and the $\pm$ sign indicates the 95% confidence interval over 3 runs with different seeds.

| | | $1S$ | $5S$ | $10S$ | $25S$ | $50S$ | $100S$ | $250S$ | $500S$ |
|---|---|---|---|---|---|---|---|---|---|
| | $\epsilon = 1$ | 7.1±2.0 | 12.9±6.3 | 8.8±1.2 | 9.0±1.2 | 8.8±2.3 | 20.5±3.6 | 25.7±0.4 | 32.1±9.5 |
| | $\epsilon = 2$ | 7.2±2.0 | 9.0±1.6 | 9.4±1.9 | 8.6±0.7 | 12.3±6.3 | 23.1±4.9 | 35.5±2.9 | 46.1±5.0 |
| All | $\epsilon = 4$ | 7.2±2.0 | 7.3±2.3 | 7.6±1.7 | 7.8±1.8 | 9.5±1.4 | 27.8±3.7 | 42.9±6.7 | 65.1±6.9 |
| | $\epsilon = 8$ | 7.2±2.0 | 8.5±2.3 | 8.7±1.1 | 9.0±0.6 | 22.4±8.9 | 39.6±3.1 | 60.9±5.9 | 78.1±3.1 |
| | $\epsilon = \infty$ | 14.4±1.5 | 19.6±12.8 | 42.2±4.1 | 76.1±4.5 | 84.1±5.5 | 86.8±5.0 | 93.1±0.5 | 94.6±0.2 |
| | $\epsilon = 1$ | 12.6±3.6 | 9.5±0.6 | 11.4±2.8 | 12.3±1.4 | 13.9±1.6 | 18.6±2.1 | 25.1±1.1 | 33.8±2.5 |
| | $\epsilon = 2$ | 11.3±4.0 | 10.8±1.3 | 12.0±1.6 | 15.0±1.8 | 16.3±0.5 | 21.7±1.4 | 32.9±3.5 | 45.3±2.9 |
| FiLM | $\epsilon = 4$ | 9.3±0.4 | 11.4±2.0 | 13.5±1.3 | 17.0±0.8 | 20.7±1.8 | 27.6±2.1 | 39.8±3.4 | 61.1±2.6 |
| | $\epsilon = 8$ | 9.9±1.0 | 11.9±1.6 | 16.6±1.4 | 21.2±1.8 | 25.6±1.9 | 33.3±2.1 | 51.4±3.9 | 67.3±2.0 |
| | $\epsilon = \infty$ | 13.8±0.3 | 20.2±1.1 | 25.7±1.7 | 37.8±4.1 | 42.4±1.8 | 58.7±6.2 | 71.5±5.5 | 84.4±3.1 |
| | $\epsilon = 1$ | 10.0±0.9 | 8.7±1.0 | 10.8±0.1 | 11.1±1.2 | 13.3±0.9 | 18.1±1.4 | 24.7±1.0 | 31.2±0.7 |
| | $\epsilon = 2$ | 9.9±0.5 | 9.1±1.4 | 11.2±1.4 | 13.9±1.6 | 17.1±1.1 | 21.2±2.0 | 29.6±1.5 | 36.1±1.7 |
| Head | $\epsilon = 4$ | 10.3±0.8 | 10.2±0.8 | 13.5±2.1 | 17.4±0.2 | 19.3±0.7 | 24.9±1.1 | 35.3±1.2 | 40.9±1.0 |
| | $\epsilon = 8$ | 10.3±0.8 | 11.1±1.3 | 15.0±3.1 | 19.9±1.8 | 23.3±1.3 | 29.3±1.0 | 39.7±1.5 | 45.2±1.2 |
| | $\epsilon = \infty$ | 13.8±0.5 | 18.5±1.6 | 21.2±3.4 | 28.7±1.4 | 32.8±1.6 | 38.0±1.4 | 47.1±0.5 | 48.5±3.2 |

Table 5: Classification accuracy as a function of $\epsilon$, $S$ and learnable parameters for CIFAR-10. Backbone is VIT-B pretrained on ImageNet-21k. Accuracy figures are percentages and the $\pm$ sign indicates the 95% confidence interval over 3 runs with different seeds.

| | | $1S$ | $5S$ | $10S$ | $25S$ | $50S$ | $100S$ | $250S$ | $500S$ |
|---|---|---|---|---|---|---|---|---|---|
| | $\epsilon = 1$ | 12.5±1.3 | 20.9±10.2 | 18.8±7.4 | 64.9±13.2 | 70.3±15.1 | 84.2±10.9 | 93.0±2.0 | 95.3±1.0 |
| | $\epsilon = 2$ | 12.7±1.0 | 9.1±2.7 | 24.8±24.0 | 76.9±1.4 | 88.4±1.3 | 93.1±0.9 | 95.4±1.0 | 97.0±0.9 |
| All | $\epsilon = 4$ | 12.6±1.4 | 42.7±3.2 | 59.2±5.2 | 86.8±1.5 | 91.8±1.0 | 95.3±0.7 | 96.9±0.7 | 97.6±0.5 |
| | $\epsilon = 8$ | 12.7±1.7 | 51.7±9.6 | 54.9±10.7 | 86.2±10.5 | 90.9±4.9 | 96.6±0.8 | 97.3±0.3 | 98.1±0.3 |
| | $\epsilon = \infty$ | 64.3±12.8 | 91.4±1.6 | 95.1±0.8 | 97.2±0.3 | 97.6±0.4 | 97.9±0.3 | 98.3±0.1 | 98.4±0.1 |
| | $\epsilon = 1$ | 10.3±3.1 | 15.0±3.9 | 23.8±1.7 | 57.7±8.7 | 81.9±1.6 | 89.6±1.5 | 95.5±1.0 | 96.9±0.2 |
| | $\epsilon = 2$ | 11.4±2.9 | 21.5±8.0 | 37.5±6.4 | 74.5±4.8 | 91.7±0.2 | 93.5±1.5 | 96.1±0.9 | 97.3±0.1 |
| FiLM | $\epsilon = 4$ | 13.3±2.0 | 37.7±5.4 | 58.6±5.9 | 82.8±5.5 | 93.1±1.0 | 94.4±1.2 | 96.9±0.4 | 97.5±0.1 |
| | $\epsilon = 8$ | 16.4±1.1 | 51.4±10.9 | 71.5±2.0 | 89.4±4.4 | 94.6±1.1 | 96.0±1.0 | 97.1±0.1 | 97.6±0.6 |
| | $\epsilon = \infty$ | 67.0±7.4 | 92.1±2.7 | 95.3±2.4 | 97.2±1.1 | 97.9±0.6 | 98.0±0.4 | 98.6±0.1 | 98.7±0.1 |
| | $\epsilon = 1$ | 14.6±2.7 | 19.1±5.0 | 30.3±6.7 | 56.6±2.1 | 81.5±1.7 | 90.6±1.5 | 95.3±0.3 | 96.4±0.2 |
| | $\epsilon = 2$ | 15.7±2.9 | 23.7±4.9 | 44.7±12.3 | 74.9±9.3 | 86.2±2.1 | 93.9±0.2 | 96.3±0.4 | 96.9±0.1 |
| Head | $\epsilon = 4$ | 17.3±2.9 | 34.9±9.8 | 59.7±9.3 | 85.3±6.5 | 94.4±0.6 | 95.1±1.4 | 96.7±0.3 | 97.0±0.4 |
| | $\epsilon = 8$ | 19.5±4.1 | 42.2±2.8 | 74.7±9.3 | 91.7±1.6 | 92.9±5.5 | 95.4±0.7 | 97.0±0.2 | 97.1±0.3 |
| | $\epsilon = \infty$ | 66.0±5.7 | 74.8±5.6 | 90.7±4.7 | 95.1±2.6 | 96.5±0.4 | 97.0±0.1 | 97.3±0.1 | 97.4±0.1 |

Table 6: Classification accuracy as a function of $\epsilon$, $S$ and learnable parameters for CIFAR-100. Backbone is VIT-B pretrained on ImageNet-21k. Accuracy figures are percentages and the $\pm$ sign indicates the 95% confidence interval over 3 runs with different seeds.

|  |  | $1S$ | $5S$ | $10S$ | $25S$ | $50S$ | $100S$ | $250S$ | $500S$ |
|---|---|---|---|---|---|---|---|---|---|
| All | $\epsilon = 1$ | 1.1±0.3 | 1.0±0.3 | 3.4±2.1 | 18.7±3.3 | 41.0±2.0 | 62.7±2.0 | 80.2±4.9 | 85.7±2.9 |
|  | $\epsilon = 2$ | 1.1±0.1 | 3.2±1.9 | 11.9±1.4 | 39.9±2.5 | 60.8±2.2 | 78.0±0.8 | 87.3±0.1 | 89.5±0.6 |
|  | $\epsilon = 4$ | 0.9±0.2 | 10.5±2.1 | 24.3±2.2 | 56.4±3.5 | 68.5±15.8 | 82.9±5.3 | 89.0±1.3 | 90.2±1.2 |
|  | $\epsilon = 8$ | 1.3±0.4 | 17.3±7.7 | 18.8±2.9 | 60.8±18.8 | 84.2±0.3 | 86.9±2.4 | 90.3±0.3 | 91.2±0.2 |
|  | $\epsilon = \infty$ | 26.2±14.5 | 78.1±1.0 | 85.3±0.6 | 88.4±0.7 | 89.6±0.3 | 90.9±0.4 | 92.1±0.2 | 93.0±0.0 |
| FiLM | $\epsilon = 1$ | 1.3±0.1 | 2.1±1.2 | 5.1±1.3 | 22.4±0.4 | 53.5±4.1 | 71.6±2.5 | 84.9±0.2 | 89.4±0.3 |
|  | $\epsilon = 2$ | 1.6±0.2 | 4.5±1.5 | 15.5±0.7 | 51.4±2.7 | 69.1±1.3 | 82.4±2.0 | 89.1±0.6 | 90.3±0.5 |
|  | $\epsilon = 4$ | 1.6±0.1 | 11.2±1.9 | 35.3±4.0 | 66.2±3.2 | 82.0±0.8 | 87.0±0.4 | 90.6±0.4 | 92.0±0.2 |
|  | $\epsilon = 8$ | 2.7±0.7 | 25.6±2.4 | 53.3±4.6 | 77.6±1.8 | 83.6±3.5 | 88.1±1.2 | 91.6±0.1 | 92.3±0.5 |
|  | $\epsilon = \infty$ | 42.1±2.5 | 79.1±3.1 | 84.2±2.8 | 89.4±0.5 | 90.6±0.5 | 91.6±0.4 | 91.9±1.8 | 90.9±1.3 |
| Head | $\epsilon = 1$ | 1.3±0.4 | 2.8±0.3 | 5.7±0.7 | 24.2±0.9 | 49.0±3.2 | 70.9±0.3 | 82.1±0.5 | 85.1±0.3 |
|  | $\epsilon = 2$ | 1.4±0.2 | 5.7±0.1 | 12.7±2.4 | 48.7±1.1 | 70.2±1.3 | 78.5±2.4 | 85.1±0.2 | 87.0±0.3 |
|  | $\epsilon = 4$ | 2.2±0.4 | 11.7±0.9 | 29.5±5.0 | 65.7±3.9 | 76.0±5.0 | 83.7±0.6 | 86.9±0.4 | 88.1±0.3 |
|  | $\epsilon = 8$ | 3.2±0.5 | 21.7±2.0 | 51.9±4.6 | 74.5±4.2 | 82.0±0.3 | 85.4±0.5 | 87.2±1.3 | 88.5±0.9 |
|  | $\epsilon = \infty$ | 35.8±12.2 | 72.2±4.5 | 78.7±3.0 | 84.3±0.8 | 86.1±0.3 | 87.4±0.4 | 88.0±0.8 | 88.4±0.4 |

Table 7: Classification accuracy as a function of $\epsilon$, $S$ and learnable parameters for SVHN. Backbone is VIT-B pretrained on ImageNet-21k. Accuracy figures are percentages and the $\pm$ sign indicates the 95% confidence interval over 3 runs with different seeds.

|  |  | $1S$ | $5S$ | $10S$ | $25S$ | $50S$ | $100S$ | $250S$ | $500S$ |
|---|---|---|---|---|---|---|---|---|---|
| All | $\epsilon = 1$ | 11.9±1.9 | 9.5±2.1 | 9.7±2.4 | 9.2±1.5 | 10.3±1.4 | 14.1±4.6 | 22.4±2.8 | 33.5±14.9 |
|  | $\epsilon = 2$ | 11.7±1.7 | 10.1±0.3 | 9.9±1.9 | 8.6±0.5 | 12.6±5.5 | 22.8±0.4 | 37.9±8.4 | 55.2±20.9 |
|  | $\epsilon = 4$ | 10.7±0.5 | 10.9±2.1 | 10.8±1.7 | 9.7±2.0 | 15.6±6.2 | 28.6±5.3 | 45.8±17.7 | 66.1±22.0 |
|  | $\epsilon = 8$ | 10.5±0.5 | 10.5±0.6 | 9.1±0.5 | 14.3±5.1 | 25.5±4.0 | 36.6±18.3 | 64.6±31.4 | 84.4±5.3 |
|  | $\epsilon = \infty$ | 10.5±1.2 | 15.6±11.2 | 28.4±22.9 | 63.0±26.9 | 86.1±5.0 | 91.2±1.0 | 93.0±1.1 | 94.2±1.0 |
| FiLM | $\epsilon = 1$ | 11.7±3.0 | 10.7±2.4 | 11.1±1.4 | 10.0±1.2 | 11.4±2.4 | 17.0±1.7 | 26.4±1.1 | 43.7±4.5 |
|  | $\epsilon = 2$ | 11.6±2.8 | 12.2±0.6 | 10.3±0.9 | 13.1±2.8 | 14.4±4.2 | 23.5±0.6 | 41.0±3.6 | 68.6±4.0 |
|  | $\epsilon = 4$ | 10.3±2.2 | 11.1±2.2 | 12.5±2.2 | 15.8±4.0 | 20.5±1.9 | 30.3±4.4 | 64.8±4.6 | 77.5±2.3 |
|  | $\epsilon = 8$ | 9.1±1.6 | 13.1±1.4 | 14.4±2.1 | 20.9±0.9 | 23.4±2.1 | 53.4±4.8 | 74.3±1.2 | 83.7±0.5 |
|  | $\epsilon = \infty$ | 12.6±2.6 | 22.1±1.2 | 31.0±3.6 | 65.9±21.8 | 83.7±3.1 | 87.7±5.9 | 92.2±0.7 | 93.6±0.7 |
| Head | $\epsilon = 1$ | 9.1±0.5 | 10.0±2.9 | 10.9±1.5 | 11.5±0.3 | 12.5±0.4 | 15.5±0.9 | 21.5±1.6 | 31.1±1.5 |
|  | $\epsilon = 2$ | 9.2±0.7 | 11.7±2.9 | 12.1±0.7 | 12.4±1.6 | 15.8±1.4 | 22.0±2.1 | 28.8±2.1 | 37.4±0.6 |
|  | $\epsilon = 4$ | 9.7±0.5 | 12.2±4.1 | 12.9±0.1 | 15.5±2.2 | 19.5±0.5 | 24.8±2.1 | 35.5±1.2 | 42.9±1.0 |
|  | $\epsilon = 8$ | 9.3±0.8 | 10.7±0.8 | 14.5±0.2 | 18.4±0.5 | 23.2±2.1 | 29.6±2.3 | 40.5±0.9 | 46.3±1.1 |
|  | $\epsilon = \infty$ | 12.7±2.0 | 14.6±2.8 | 23.5±0.9 | 29.6±1.6 | 33.8±2.6 | 38.8±1.3 | 47.6±1.7 | 52.6±1.1 |

### A.2.2 Additional Shot Multiplier Variations

We compute the multiplier for a configuration and dataset at $\epsilon$ as follows: using the median accuracy obtained through the experiments depicted in Tables 2 to 7 ($S = 1, 5, 10, 25, 50, 100, 250, 500$) we linearly interpolate the median accuracy in the complete $S = [1, 500]$ grid. We determine the minimum $S$ required to reach at least the same accuracy as for non-private at $S \in \{5, 10\}$ using the $S = [1, 500]$ grid. The multiplier is then the minimum $S$ required for DP divided by the $S$ for non-private.

The Figs. 10 and 11 display the same analysis as Fig. 2 for all backbones (VIT-B, R-50) and non-private shots of $S \in \{5, 10\}$. The Figs. 12 and 13 display the same analysis grouped by datasets.

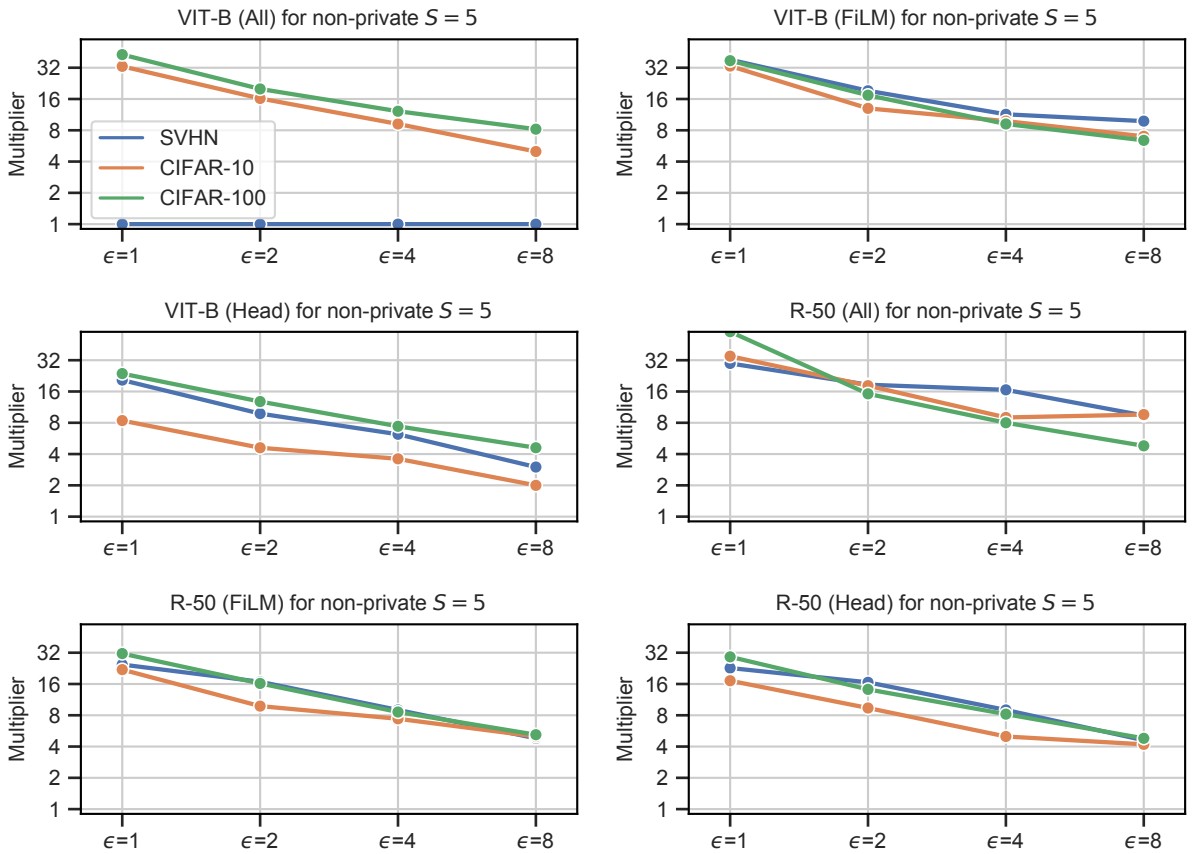

Figure 10: Multiplier of shots required to reach same accuracy as non-private with $S = 5$ for VIT-B and R-50 on CIFAR-10, CIFAR-100 and SVHN with $\delta = 1/|\mathcal{D}|$. The data is obtained using linear interpolation of the median results of the experiments of Appendix A.2.1. The multiplier is 1 for all $\epsilon$ for ViT-B with All parameters on SVHN at $S = 5$ (top left plot) because non-DP achieves a random accuracy and achieving random accuracy requires $S = 1$ for all configurations in the experiment.

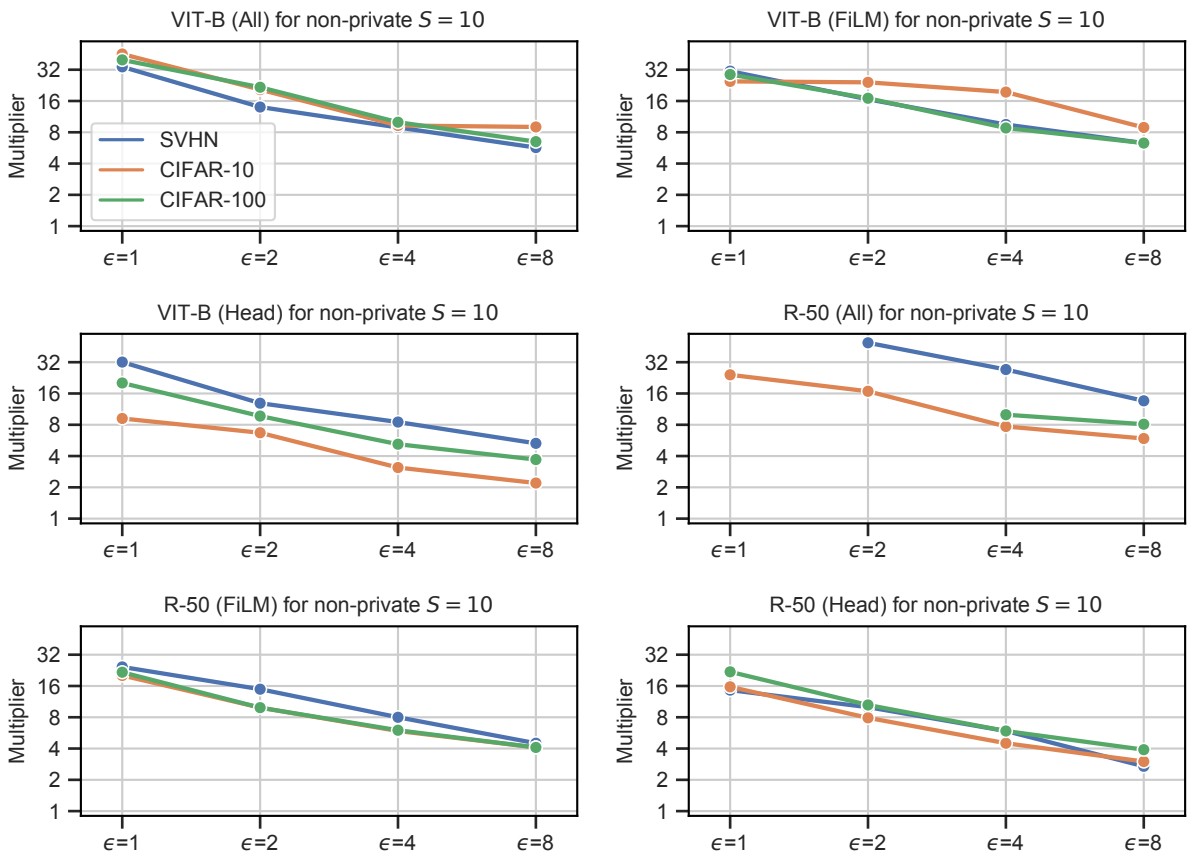

Figure 11: Multiplier of shots required to reach same accuracy as non-private with $S = 10$ for VIT-B and R-50 on CIFAR-10, CIFAR-100 and SVHN with $\delta = 1/|\mathcal{D}|$. The data is obtained using linear interpolation of the median results of the experiments of Appendix A.2.1.

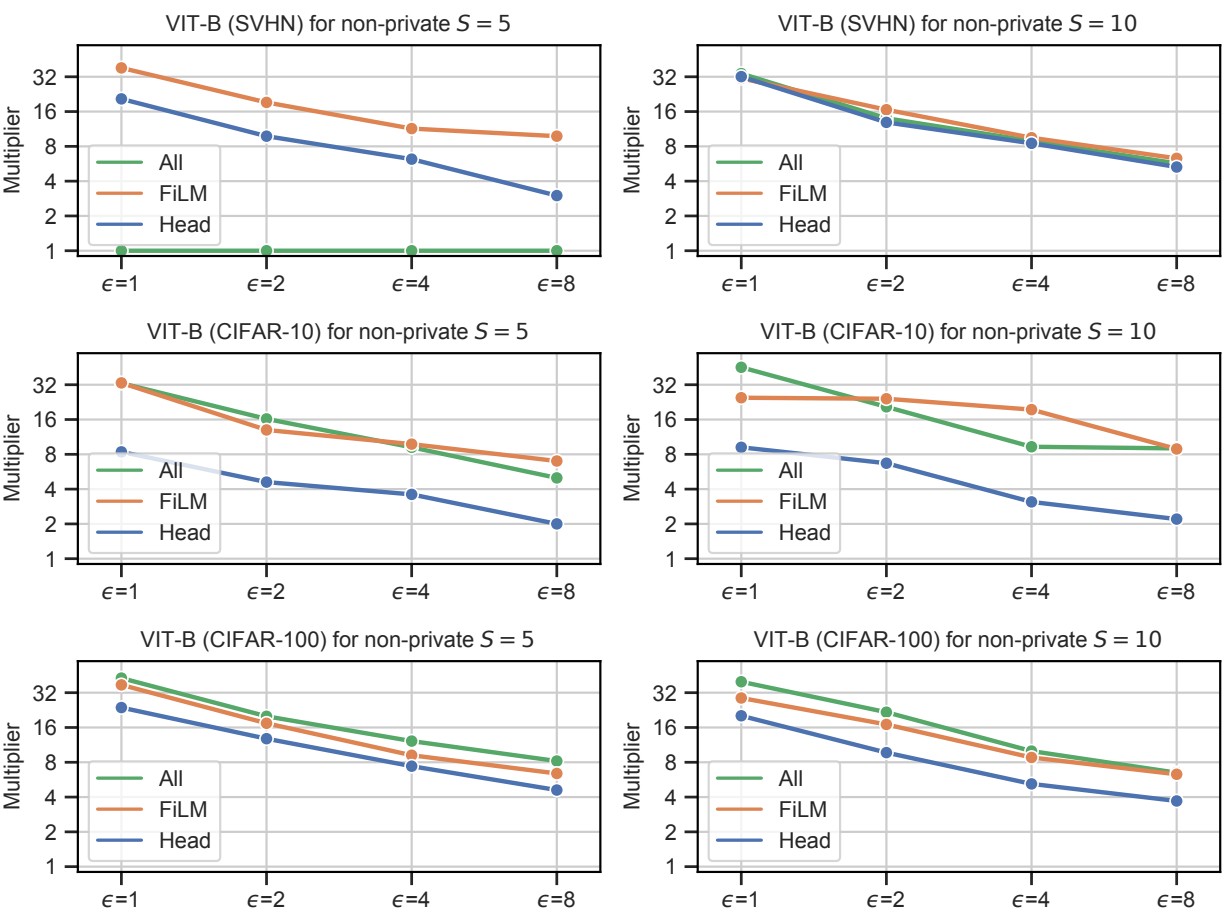

Figure 12: Multiplier of shots required to reach same accuracy as non-private with $S \in \{5, 10\}$ for VIT-B on CIFAR-10, CIFAR-100 and SVHN with $\delta = 1/|\mathcal{D}|$. The data is obtained using linear interpolation of the median results of the experiments of Appendix A.2.1. The multiplier is 1 for all $\epsilon$ for ViT-B with All parameters on SVHN at $S = 5$ (top left plot) because non-DP achieves a random accuracy and achieving random accuracy requires $S = 1$ for all configurations in the experiment.

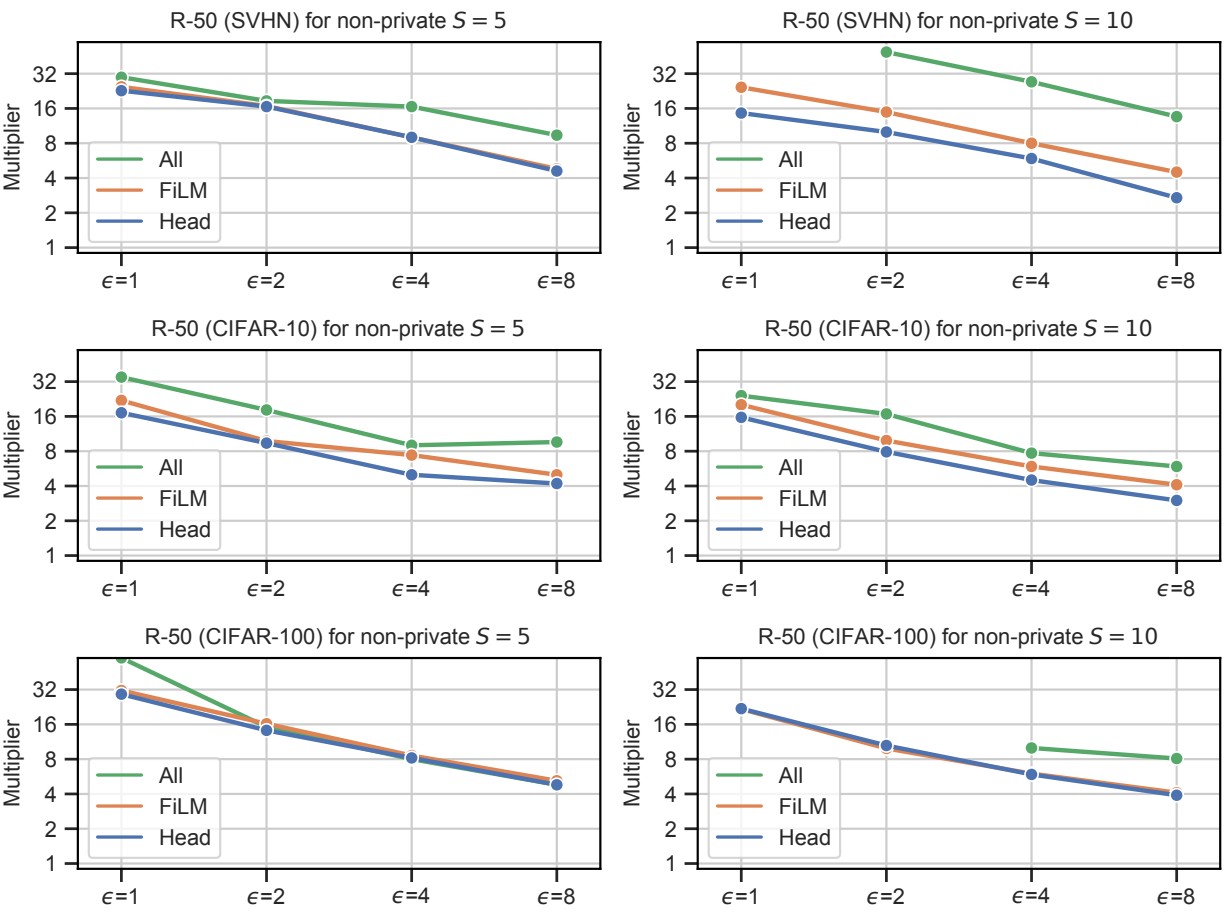

Figure 13: Multiplier of shots required to reach same accuracy as non-private with $S \in \{5, 10\}$ for R-50 on CIFAR-10, CIFAR-100 and SVHN with $\delta = 1/|\mathcal{D}|$. The data is obtained using linear interpolation of the median results of the experiments of Appendix A.2.1.

### A.2.3 Additional versions of Fig. 4

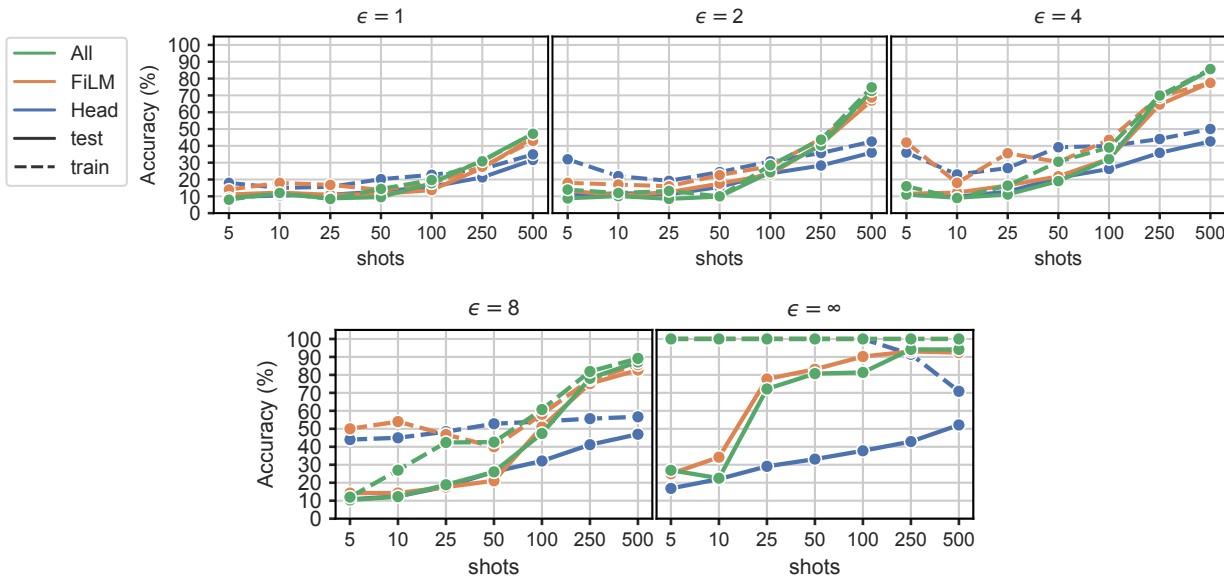

Figure 14: Test and train classification accuracy as a function of shots and learnable parameters (*All*, *FiLM* and *Head*) on VIT-B for SVHN for different $\epsilon$ with $\delta = 1/|\mathcal{D}|$. The accuracy is reported for the median runs of Table 7.

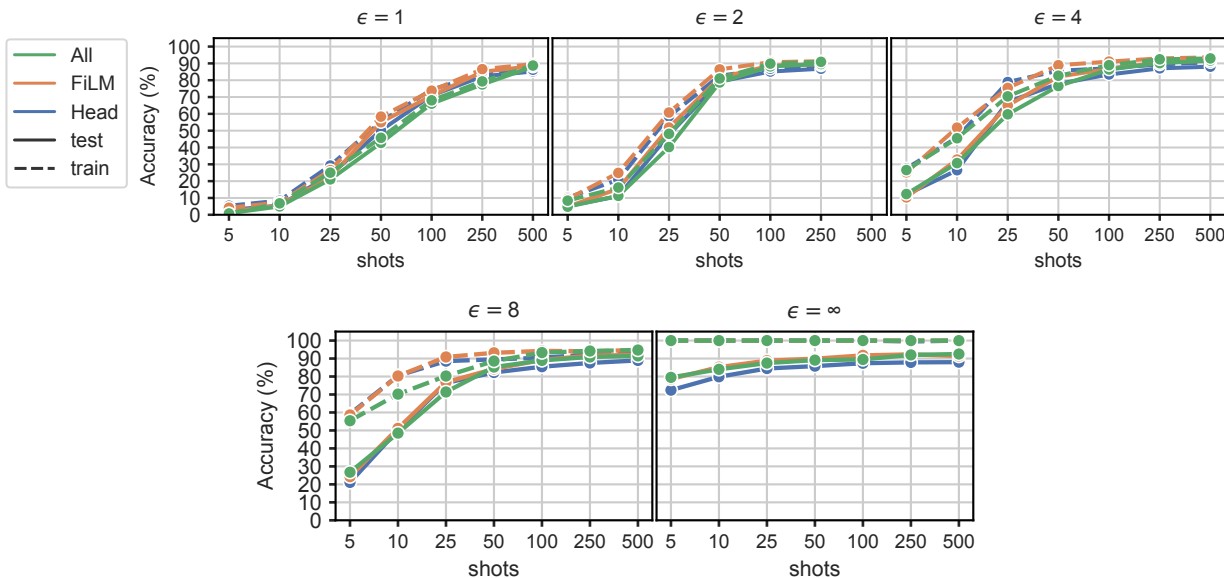

Figure 15: Test and train classification accuracy as a function of shots and learnable parameters (*All*, *FiLM* and *Head*) on VIT-B for CIFAR-100 for different $\epsilon$ with $\delta = 1/|\mathcal{D}|$. The accuracy is reported for the median runs of Table 6.

### A.2.4    Comparison of Backbones for Effect of Shots and $\epsilon$

Fig. 16 compares the backbones (VIT-B, R-50) using their best performing configuration. The VIT-B backbone achieves comparable or better performance.

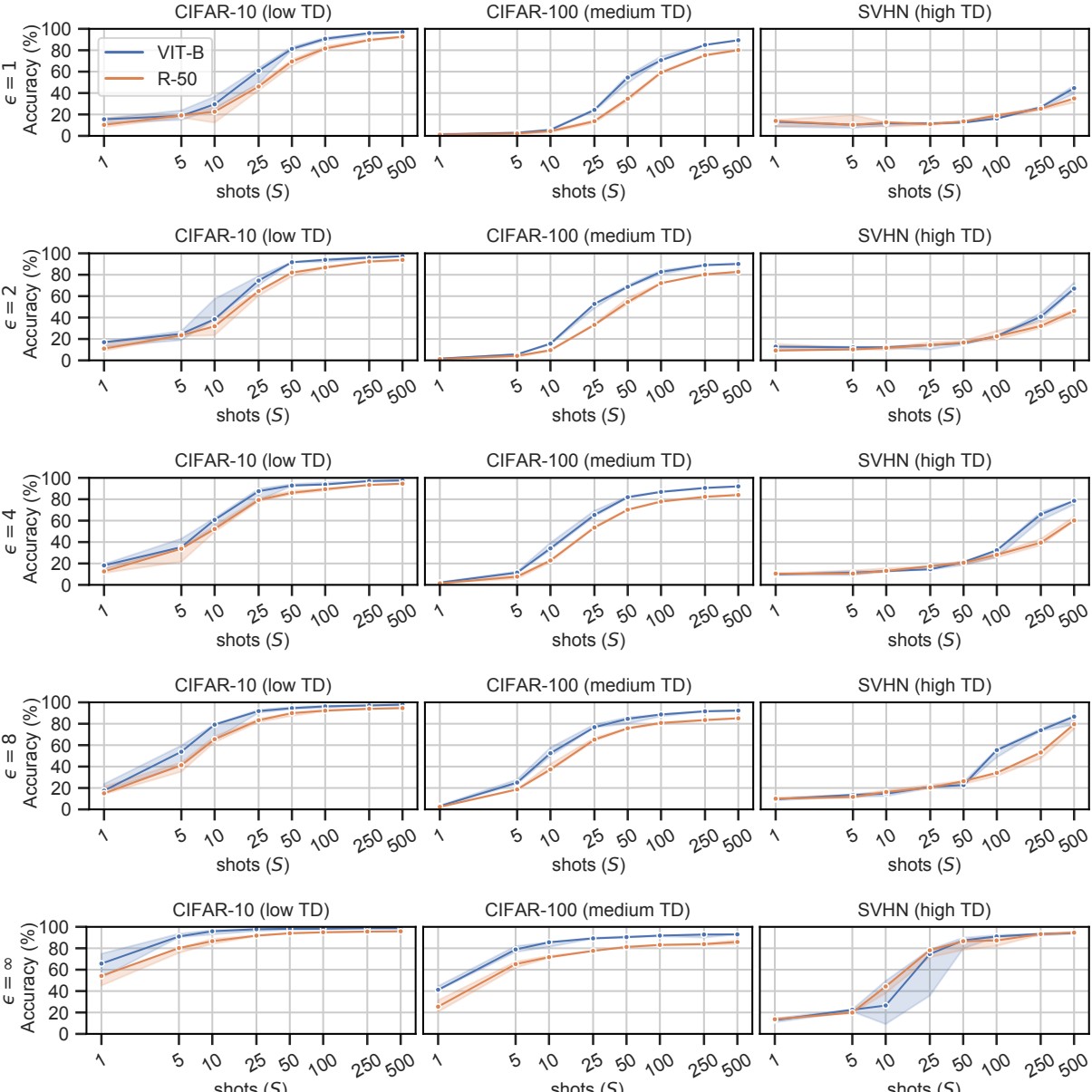

Figure 16: Classification accuracy for different $\epsilon$ as a function of $S$ and backbone (VIT-B, R-50) for CIFAR-10, CIFAR-100 and SVHN. TD (low, medium, high) refers to the transfer difficulty and is computed as in Appendix A.1. The best performing configuration out of *All*, *FiLM* and *Head* for each combination of $\epsilon$, $S$ and backbone is used. The accuracy is reported over three seeds with the line showing the median and the band reporting the lowest and highest accuracy.

### A.2.5    Advantage of *FiLM* as a Function of Shots

Figs. 17 and 18 show the difference between the mean classification accuracy of *FiLM* and *Head*. Darker red indicates *FiLM* is better. Darker blue indicates *Head* is better.

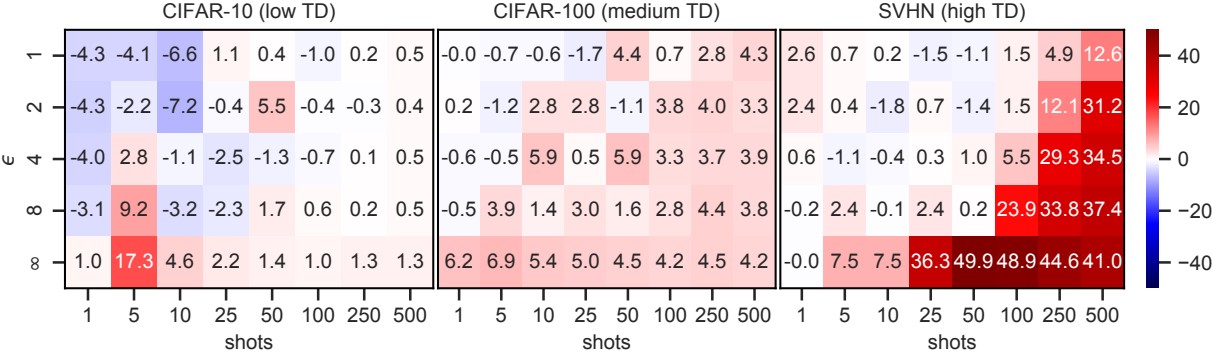

Figure 17: Heat map showing the accuracy advantage of *FiLM* over *Head* for CIFAR-10, CIFAR-100 and SVHN as a function of $\epsilon$. Backbone is VIT-B. Darker red indicates *FiLM* is better. Darker blue indicates *Head* is better. Datasets ordered from low to high TD.

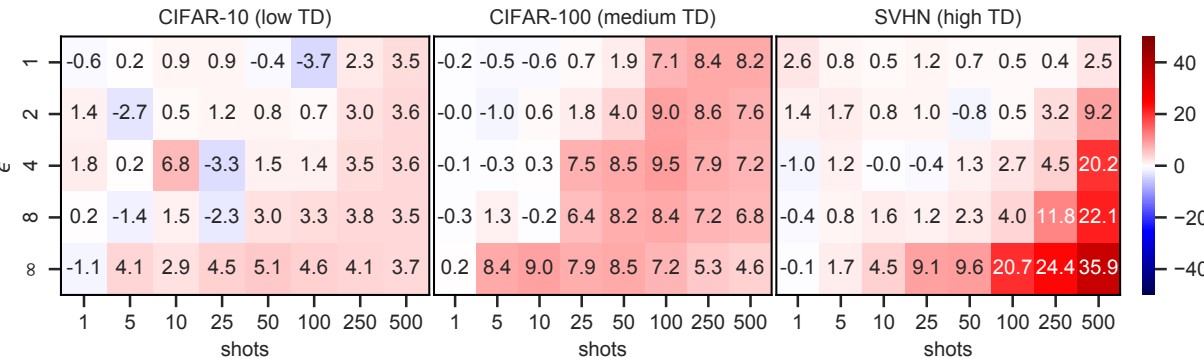

Figure 18: Heat map showing the accuracy advantage of *FiLM* over *Head* for CIFAR-10, CIFAR-100 and SVHN as a function of $\epsilon$. Backbone is R-50. Darker red indicates *FiLM* is better. Darker blue indicates *Head* is better. Datasets ordered from low to high TD.

### A.2.6   Additional VTAB-1k Results

Tables 8 to 13 depict tabular results for different backbones (R-50, ViT-B), different learnable parameter sets (*Head*, *FiLM*, *All*), and various privacy levels ($\epsilon = 1, 2, 4, 8, \infty$), all at $\delta = 10^{-3}$.

Table 8: Classification accuracy as a function of $\epsilon$ for each of the datasets in the VTAB-1k benchmark. Backbone is R-50 pretrained on ImageNet-21k. Learnable parameters are Head. Accuracy figures are percentages and the $\pm$ sign indicates the 95% confidence interval over 3 runs with different seeds.

| dataset | classes | $\epsilon = 1$ | $\epsilon = 2$ | $\epsilon = 4$ | $\epsilon = 8$ | $\epsilon = \infty$ |
|---|---|---|---|---|---|---|
| Caltech101 (Fei-Fei et al., 2006) | 102 | 11.8±6.9 | 30.0±4.7 | 57.1±3.6 | 69.3±1.4 | 87.9±0.2 |
| CIFAR100 (Krizhevsky, 2009) | 100 | 4.2±1.2 | 10.6±1.0 | 20.8±1.6 | 34.7±2.2 | 61.5±0.6 |
| Flowers102 (Nilsback & Zisserman, 2008) | 102 | 11.3±2.7 | 33±6.4 | 73.1±0.7 | 89.8±2.1 | 98.4±0.1 |
| Pets (Parkhi et al., 2012) | 37 | 28.6±5.4 | 50±2.4 | 65.6±1.5 | 73.4±1.0 | 84.4±0.3 |
| Sun397 (Xiao et al., 2010) | 397 | 4.7±0.2 | 8.4±0.2 | 13.4±1.2 | 21.5±0.7 | 46.2±0.4 |
| SVHN (Netzer et al., 2011) | 10 | 23.0±1.0 | 26.8±2.5 | 30.6±1.9 | 34.9±2.0 | 41.9±2.2 |
| DTD (Cimpoi et al., 2014) | 47 | 19.6±4.0 | 36.2±1.3 | 51.2±1.2 | 61.3±0.9 | 72.0±0.4 |
| EuroSAT (Helber et al., 2019) | 10 | 77.2±1.9 | 85.3±1.4 | 88.4±1 | 91.1±0.0 | 94.3±0.2 |
| Resics45 (Cheng et al., 2017) | 45 | 19.3±3.5 | 33.4±2.8 | 48.6±2.5 | 60.9±0.8 | 78.5±0.2 |
| Patch Camelyon (Veeling et al., 2018) | 2 | 77.6±2.3 | 79.2±1.0 | 80.8±1.6 | 80.6±0.3 | 81.2±0.2 |
| Retinopathy (Kaggle & EyePacs, 2015) | 5 | 73.2±0.6 | 73.7±0.3 | 73.4±0.6 | 74.0±0.4 | 75.2±0.1 |
| CLEVR-count (Johnson et al., 2017) | 8 | 27.5±1.5 | 30.1±1.9 | 33.6±1.5 | 36.8±2.1 | 51.2±1.2 |
| CLEVR-dist (Johnson et al., 2017) | 6 | 26.4±2.2 | 28.7±0.7 | 29.8±0.9 | 30.9±1.6 | 36.2±0.9 |
| dSprites-loc (Matthey et al., 2017) | 16 | 6.6±0.1 | 6.8±0.7 | 7.6±0.5 | 7.5±1.2 | 18.9±6.5 |
| dSprites-ori (Matthey et al., 2017) | 16 | 9.3±0.9 | 10.8±0.5 | 13.2±0.7 | 16.2±0.2 | 45.9±2.5 |
| SmallNORB-azi (LeCun et al., 2004) | 18 | 6.1±0.5 | 7.5±0.5 | 8.1±0.4 | 8.7±0.7 | 11.7±0.1 |
| SmallNORB-elev (LeCun et al., 2004) | 9 | 17±3.3 | 19.8±1.1 | 22.5±0.9 | 24.5±0.7 | 31±0.5 |
| DMLab (Beattie et al., 2016) | 6 | 26.9±0.8 | 28.8±0.6 | 31±0.2 | 32.4±0.1 | 34.6±3.7 |
| KITTI-dist (Geiger et al., 2013) | 4 | 54.5±2.5 | 59.6±3.5 | 66.9±1.6 | 65.4±1.3 | 69.2±0.7 |
| All | | 27.6 | 34.7 | 43.0 | 48.1 | 58.9 |
| Natural | | 14.7 | 27.9 | 44.5 | 55.0 | 70.4 |
| Specialized | | 61.8 | 67.9 | 72.8 | 76.7 | 82.3 |
| Structured | | 21.8 | 24.0 | 26.6 | 27.8 | 37.3 |

Table 9: Classification accuracy as a function of $\epsilon$ for each of the datasets in the VTAB-1k benchmark. Backbone is R-50 pretrained on ImageNet-21k. Learnable backbone parameters are FiLM. Accuracy figures are percentages and the $\pm$ sign indicates the 95% confidence interval over 3 runs with different seeds.

| dataset | classes | $\epsilon = 1$ | $\epsilon = 2$ | $\epsilon = 4$ | $\epsilon = 8$ | $\epsilon = \infty$ |
|---|---|---|---|---|---|---|
| Caltech101 (Fei-Fei et al., 2006) | 102 | 11.3±1.3 | 35.8±4.1 | 55.7±0.5 | 72.2±1.9 | 88.8±0.5 |
| CIFAR100 (Krizhevsky, 2009) | 100 | 3.4±0.8 | 10.2±0.5 | 23.2±0.8 | 38.5±1.3 | 71.7±1.3 |
| Flowers102 (Nilsback & Zisserman, 2008) | 102 | 10.4±0.6 | 34.2±5.4 | 70.4±1.0 | 89.3±0.3 | 98.7±0.1 |
| Pets (Parkhi et al., 2012) | 37 | 28.4±1.4 | 48.8±3.0 | 64.1±1.1 | 75±1.1 | 88±0.4 |
| Sun397 (Xiao et al., 2010) | 397 | 4.2±0.5 | 8.0±0.1 | 14.1±0.8 | 21.7±0.8 | 46.8±0.7 |
| SVHN (Netzer et al., 2011) | 10 | 23.3±1.4 | 28.1±0.9 | 32.9±0.7 | 38.6±3.2 | 56.6±2.4 |
| DTD (Cimpoi et al., 2014) | 47 | 20.8±2.7 | 36.7±1.5 | 50.3±4.5 | 61.4±1.3 | 72.4±0.4 |
| EuroSAT (Helber et al., 2019) | 10 | 79.2±0.8 | 85.1±1.5 | 88.8±2.2 | 92.1±0.7 | 95±0.1 |
| Resics45 (Cheng et al., 2017) | 45 | 21.1±1.7 | 35.2±0.6 | 49.5±1.6 | 61.3±0.8 | 81.9±0.1 |
| Patch Camelyon (Veeling et al., 2018) | 2 | 76.8±0.8 | 77.3±2.7 | 79.1±1.2 | 79.4±0.4 | 81.3±0.1 |
| Retinopathy (Kaggle & EyePacs, 2015) | 5 | 73.4±0.3 | 73.5±0.1 | 73.9±0.5 | 74.4±0.2 | 74.0±3.2 |
| CLEVR-count (Johnson et al., 2017) | 8 | 29.1±1.5 | 31.0±0.4 | 34.6±1.3 | 38±1.4 | 73±1.3 |
| CLEVR-dist (Johnson et al., 2017) | 6 | 26.7±1.3 | 28.7±0.7 | 30.5±0.6 | 31.8±0.6 | 49.3±1.6 |
| dSprites-loc (Matthey et al., 2017) | 16 | 6.6±0.3 | 6.4±0.3 | 6.7±0.5 | 8.5±1.4 | 64.0±8.7 |
| dSprites-ori (Matthey et al., 2017) | 16 | 9.1±2.0 | 11.2±1.7 | 12.3±1.0 | 16.7±0.8 | 56.8±3.8 |
| SmallNORB-azi (LeCun et al., 2004) | 18 | 6.4±0.4 | 7.3±0.8 | 8.2±0.8 | 9.4±0.5 | 14.6±0.2 |
| SmallNORB-elev (LeCun et al., 2004) | 9 | 17.6±1.0 | 20.8±0.3 | 22.7±1.1 | 25.6±0.3 | 32.0±4.0 |
| DMLab (Beattie et al., 2016) | 6 | 25.8±0.7 | 28.7±0.7 | 30.5±0.9 | 32.1±0.8 | 41.8±0.4 |
| KITTI-dist (Geiger et al., 2013) | 4 | 56.3±1.6 | 60.7±3.9 | 63.4±2.5 | 68.5±1.6 | 80.4±0.5 |
| All | | 27.9 | 35.1 | 42.7 | 49.2 | 66.7 |
| Natural | | 14.6 | 28.8 | 44.4 | 56.7 | 74.7 |
| Specialized | | 62.6 | 67.8 | 72.8 | 76.8 | 83.1 |
| Structured | | 22.2 | 24.3 | 26.2 | 28.8 | 51.5 |

Table 10: Classification accuracy as a function of $\epsilon$ for each of the datasets in the VTAB-1k benchmark. Backbone is R-50 pretrained on ImageNet-21k. All parameters are learnable. Accuracy figures are percentages and the $\pm$ sign indicates the 95% confidence interval over 3 runs with different seeds.

| dataset | classes | $\epsilon = 1$ | $\epsilon = 2$ | $\epsilon = 4$ | $\epsilon = 8$ | $\epsilon = \infty$ |
|---|---|---|---|---|---|---|
| Caltech101 (Fei-Fei et al., 2006) | 102 | 8.2±8.5 | 17.6±7.0 | 26.0±3.9 | 33.0±3.9 | 86.8±2.0 |
| CIFAR100 (Krizhevsky, 2009) | 100 | 1.0±0.1 | 2.3±1.5 | 6.9±2.9 | 13.3±2.7 | 59.3±7.0 |
| Flowers102 (Nilsback & Zisserman, 2008) | 102 | 6.2±2.8 | 6.9±7.1 | 33.7±11.6 | 69.7±7.3 | 95.8±1.8 |
| Pets (Parkhi et al., 2012) | 37 | 12.8±0.5 | 22.7±2.2 | 32.4±4.1 | 24.7±9.0 | 83.0±0.2 |
| Sun397 (Xiao et al., 2010) | 397 | 3.3±0.4 | 3.0±0.5 | 2.7±0.4 | 3.4±1.2 | 38.3±0.8 |
| SVHN (Netzer et al., 2011) | 10 | 19.2±0.8 | 23.6±5.5 | 26.8±7.0 | 37.2±2.3 | 88.5±2.5 |
| DTD (Cimpoi et al., 2014) | 47 | 13.7±3.6 | 21.6±2.3 | 27.7±3.8 | 34.0±1.9 | 72.4±0.1 |
| EuroSAT (Helber et al., 2019) | 10 | 49.8±9.5 | 69.2±5.5 | 72.7±1.2 | 82.4±3.4 | 96.0±1.0 |
| Resics45 (Cheng et al., 2017) | 45 | 11.9±1.4 | 13.8±6.4 | 24.3±1.0 | 26.8±6.8 | 84.1±1.1 |
| Patch Camelyon (Veeling et al., 2018) | 2 | 65.9±15.7 | 70.5±20.1 | 80.5±1.2 | 79.5±2.9 | 85.0±0.8 |
| Retinopathy (Kaggle & EyePacs, 2015) | 5 | 73.6±0.0 | 73.6±0.0 | 73.6±0.0 | 73.6±0.0 | 76.0±1.3 |
| CLEVR-count (Johnson et al., 2017) | 8 | 18.1±4.6 | 26.3±2.0 | 36.2±2.7 | 41.4±7.0 | 93.2±0.2 |
| CLEVR-dist (Johnson et al., 2017) | 6 | 23.8±1.4 | 22.7±2.2 | 25.2±1.7 | 36.9±3.3 | 62.1±1.7 |
| dSprites-loc (Matthey et al., 2017) | 16 | 6.2±0.1 | 6.4±0.3 | 6.3±0.1 | 6.2±0.1 | 89.1±3.7 |
| dSprites-ori (Matthey et al., 2017) | 16 | 7.5±0.0 | 6.6±1.8 | 7.2±0.1 | 8.8±2.9 | 61.0±5.2 |
| SmallNORB-azi (LeCun et al., 2004) | 18 | 5.4±0.2 | 5.7±0.1 | 5.7±0.3 | 6.2±0.8 | 21.9±3.3 |
| SmallNORB-elev (LeCun et al., 2004) | 9 | 12.3±1.2 | 13.8±2.4 | 21.2±2.7 | 22.7±5.5 | 39.5±6.3 |
| DMLab (Beattie et al., 2016) | 6 | 22.5±0.3 | 24.9±3.1 | 28.5±1.4 | 30.3±5.2 | 48.4±0.8 |
| KITTI-dist (Geiger et al., 2013) | 4 | 34.4±6.7 | 46.9±1.8 | 55.2±0.5 | 60.6±2.5 | 81.1±0.2 |
| All | | 20.8 | 25.2 | 31.2 | 36.4 | 71.7 |
| Natural | | 9.2 | 13.9 | 21.1 | 28.7 | 73.3 |
| Specialized | | 50.3 | 56.8 | 61.1 | 65.6 | 85.3 |
| Structured | | 16.3 | 19.2 | 23.2 | 26.6 | 62.0 |

Table 11: Classification accuracy as a function of $\epsilon$ for each of the datasets in the VTAB-1k benchmark. Backbone is VIT-B pretrained on ImageNet-21k. Learnable backbone parameters are Head. Accuracy figures are percentages and the $\pm$ sign indicates the 95% confidence interval over 3 runs with different seeds.

| dataset | classes | $\epsilon = 1$ | $\epsilon = 2$ | $\epsilon = 4$ | $\epsilon = 8$ | $\epsilon = \infty$ |
|---|---|---|---|---|---|---|
| Caltech101 (Fei-Fei et al., 2006) | 102 | 20.8±2.1 | 39.7±5.7 | 65.6±1.1 | 79.9±0.3 | 93.3±0.3 |
| CIFAR100 (Krizhevsky, 2009) | 100 | 7.0±1.4 | 15.9±2.7 | 33.3±1.5 | 49.9±2.3 | 77.6±2.4 |
| Flowers102 (Nilsback & Zisserman, 2008) | 102 | 13.7±3.0 | 47.2±1.5 | 85.4±1.8 | 93.5±2.6 | 99.3±0.3 |
| Pets (Parkhi et al., 2012) | 37 | 38.3±2.8 | 65.6±0.2 | 76.0±3.9 | 81.1±2.4 | 90.7±0.1 |
| Sun397 (Xiao et al., 2010) | 397 | 3.5±0.5 | 6.9±0.8 | 13.2±1.5 | 24.0±0.3 | 51.0±3.4 |
| SVHN (Netzer et al., 2011) | 10 | 23.3±1.4 | 27.2±1.5 | 31.6±1.2 | 35.3±0.3 | 43.1±0.4 |
| DTD (Cimpoi et al., 2014) | 47 | 20.4±2.6 | 37.0±3.1 | 49.9±4.0 | 61.6±3.2 | 75.7±0.3 |
| EuroSAT (Helber et al., 2019) | 10 | 81.3±1.3 | 87.0±1.0 | 89.9±0.9 | 91.6±1.1 | 94.6±0.4 |
| Resics45 (Cheng et al., 2017) | 45 | 23.2±2.8 | 41.4±2.1 | 58.0±2.7 | 67.9±2.1 | 82.5±0.5 |
| Patch Camelyon (Veeling et al., 2018) | 2 | 79.8±2.9 | 78.5±2.1 | 81.6±1.8 | 82.8±0.4 | 83.8±0.7 |
| Retinopathy (Kaggle & EyePacs, 2015) | 5 | 73.3±0.6 | 72.6±1.3 | 73.6±0.6 | 74.0±0.2 | 73.8±2.3 |
| CLEVR-count (Johnson et al., 2017) | 8 | 25.5±0.9 | 27.7±1.3 | 30.8±0.4 | 33.3±0.5 | 42.5±0.5 |
| CLEVR-dist (Johnson et al., 2017) | 6 | 26.1±0.7 | 27.5±0.5 | 30.1±0.3 | 31.5±0.5 | 35.1±0.3 |
| dSprites-loc (Matthey et al., 2017) | 16 | 6.9±0.5 | 7.8±0.7 | 8.7±0.1 | 9.4±0.6 | 19.1±2.7 |
| dSprites-ori (Matthey et al., 2017) | 16 | 11.2±0.9 | 13.3±1.2 | 15.6±0.9 | 18.9±1.5 | 31.2±0.6 |
| SmallNORB-azi (LeCun et al., 2004) | 18 | 6.9±0.6 | 7.8±0.5 | 8.1±1.3 | 9.0±0.8 | 12.2±0.1 |
| SmallNORB-elev (LeCun et al., 2004) | 9 | 16.9±1.4 | 19.3±0.4 | 20.4±1.1 | 22.7±1.8 | 27.5±0.4 |
| DMLab (Beattie et al., 2016) | 6 | 29.2±1.7 | 33.0±1.6 | 35.0±1.0 | 37.3±1 | 40.2±0.6 |
| KITTI-dist (Geiger et al., 2013) | 4 | 51.3±8.4 | 57.1±5.6 | 61.2±0.6 | 61.4±3.0 | 65.7±3.2 |
| All | | 29.4 | 37.5 | 45.7 | 50.8 | 59.9 |
| Natural | | 18.1 | 34.2 | 50.7 | 60.8 | 75.8 |
| Specialized | | 64.4 | 69.9 | 75.8 | 79.1 | 83.7 |
| Structured | | 21.7 | 24.2 | 26.2 | 27.9 | 34.2 |

Table 12: Classification accuracy as a function of $\epsilon$ for each of the datasets in the VTAB-1k benchmark. Backbone is VIT-B pretrained on ImageNet-21k. Learnable backbone parameters are FiLM. Accuracy figures are percentages and the $\pm$ sign indicates the 95% confidence interval over 3 runs with different seeds.

| dataset | classes | $\epsilon = 1$ | $\epsilon = 2$ | $\epsilon = 4$ | $\epsilon = 8$ | $\epsilon = \infty$ |
|---|---|---|---|---|---|---|
| Caltech101 (Fei-Fei et al., 2006) | 102 | 11.7±6.0 | 42.9±4.9 | 65.7±3.0 | 78.7±2.6 | 94.1±0.8 |
| CIFAR100 (Krizhevsky, 2009) | 100 | 7.1±0.2 | 17.4±1.2 | 35.4±2.0 | 52.9±2.0 | 83.8±0.6 |
| Flowers102 (Nilsback & Zisserman, 2008) | 102 | 16.0±2.8 | 48.8±5.2 | 85.3±1.7 | 96.4±0.7 | 99.5±0.0 |
| Pets (Parkhi et al., 2012) | 37 | 39.3±2.2 | 62.5±3.6 | 78.0±0.9 | 83.6±2.4 | 91.8±0.4 |
| Sun397 (Xiao et al., 2010) | 397 | 2.7±0.4 | 7.1±1.2 | 14.6±1.0 | 23.1±0.5 | 53.7±2.0 |
| SVHN (Netzer et al., 2011) | 10 | 25.1±1.5 | 28.0±1.1 | 33.4±0.7 | 52.4±7.4 | 79.1±2.6 |
| DTD (Cimpoi et al., 2014) | 47 | 17.5±2.0 | 33.0±3.5 | 50.5±1.4 | 61.7±1.1 | 75.3±3.9 |
| EuroSAT (Helber et al., 2019) | 10 | 79.6±1.8 | 86.6±2.2 | 90.9±0.2 | 91.0±0.5 | 96.5±0.2 |
| Resics45 (Cheng et al., 2017) | 45 | 22.0±3.1 | 40.6±2.3 | 55.5±3.9 | 66.1±1.7 | 87.0±0.5 |
| Patch Camelyon (Veeling et al., 2018) | 2 | 76.6±2.7 | 78.1±2.2 | 80.1±1.6 | 80.6±0.4 | 82.8±1.0 |
| Retinopathy (Kaggle & EyePacs, 2015) | 5 | 73.5±0.1 | 73.4±0.4 | 73.5±0.5 | 73.5±0.9 | 74.5±0.6 |
| CLEVR-count (Johnson et al., 2017) | 8 | 25.6±1.5 | 28.0±1.3 | 31.6±0.5 | 33.7±0.6 | 52.0±3.3 |
| CLEVR-dist (Johnson et al., 2017) | 6 | 26.5±0.9 | 29.0±0.7 | 32.6±1.3 | 38.0±3.2 | 52.6±6.4 |
| dSprites-loc (Matthey et al., 2017) | 16 | 8.0±1.6 | 12.2±1.1 | 20.9±6.1 | 29.8±5.4 | 68.1±10 |
| dSprites-ori (Matthey et al., 2017) | 16 | 9.3±0.7 | 13.7±1.5 | 17.0±1.4 | 21.5±1.1 | 47.8±3.9 |
| SmallNORB-azi (LeCun et al., 2004) | 18 | 6.6±0.3 | 7.3±0.5 | 8.4±0.3 | 9.1±0.4 | 15.1±1.5 |
| SmallNORB-elev (LeCun et al., 2004) | 9 | 16.2±1.6 | 19.2±1.3 | 21.5±1.2 | 22.9±1.4 | 35.3±4.6 |
| DMLab (Beattie et al., 2016) | 6 | 29.8±1.7 | 33.3±0.5 | 35.5±0.8 | 36.5±0.9 | 43.3±3.6 |
| KITTI-dist (Geiger et al., 2013) | 4 | 53.6±2.6 | 60.8±2.1 | 63.5±0.8 | 66±2.9 | 76.9±4.4 |
| All | | 28.8 | 38.0 | 47.1 | 53.6 | 68.9 |
| Natural | | 17.0 | 34.3 | 51.9 | 64.1 | 82.4 |
| Specialized | | 62.9 | 69.7 | 75.0 | 77.8 | 85.2 |
| Structured | | 21.9 | 25.4 | 28.9 | 32.2 | 48.6 |

Table 13: Classification accuracy as a function of $\epsilon$ for each of the datasets in the VTAB-1k benchmark. Backbone is VIT-B pretrained on ImageNet-21k. All parameters are learnable. Accuracy figures are percentages and the $\pm$ sign indicates the 95% confidence interval over 3 runs with different seeds.

| dataset | classes | $\epsilon = 1$ | $\epsilon = 2$ | $\epsilon = 4$ | $\epsilon = 8$ | $\epsilon = \infty$ |
|---|---|---|---|---|---|---|
| Caltech101 (Fei-Fei et al., 2006) | 102 | 16.1±5.2 | 34.9±2.1 | 55.3±1.0 | 69.9±2.7 | 93.7±0.4 |
| CIFAR100 (Krizhevsky, 2009) | 100 | 7.1±0.4 | 14.3±0.7 | 24.2±1.5 | 36.2±4.0 | 84.2±0.3 |
| Flowers102 (Nilsback & Zisserman, 2008) | 102 | 10.6±2.9 | 33±4.9 | 77.3±6.9 | 96±1.2 | 99.5±0.0 |
| Pets (Parkhi et al., 2012) | 37 | 26.7±6.0 | 56.9±7.0 | 76.0±3.7 | 84.2±0.7 | 91.7±0.2 |
| Sun397 (Xiao et al., 2010) | 397 | 2.4±2.1 | 5.7±1.5 | 7.7±0.4 | 11.6±3.1 | 55.9±0.2 |
| SVHN (Netzer et al., 2011) | 10 | 22.9±1.5 | 28.8±0.7 | 34±5.6 | 44.3±9.0 | 91.6±0.8 |
| DTD (Cimpoi et al., 2014) | 47 | 17.3±1.1 | 29.3±2.4 | 41.1±1.2 | 51.7±5.0 | 76.7±0.5 |
| EuroSAT (Helber et al., 2019) | 10 | 74.3±1.4 | 78.9±2.2 | 86±1.4 | 91.6±1.6 | 96.3±0.5 |
| Resics45 (Cheng et al., 2017) | 45 | 16±2.4 | 28±1.6 | 45.7±3.3 | 60.8±2.1 | 88.4±0.4 |
| Patch Camelyon (Veeling et al., 2018) | 2 | 74.1±1.5 | 76.6±1.4 | 78.9±2.1 | 76.2±5.3 | 87.1±0.7 |
| Retinopathy (Kaggle & EyePacs, 2015) | 5 | 73.4±0.5 | 73.1±0.5 | 73.6±0.1 | 73.6±0.1 | 74.0±1.3 |
| CLEVR-count (Johnson et al., 2017) | 8 | 21.5±5.6 | 28.8±1.5 | 33.6±2.4 | 38.2±0.7 | 57.6±8.7 |
| CLEVR-dist (Johnson et al., 2017) | 6 | 27.0±1.8 | 36.4±3.5 | 42.2±3.2 | 45.8±1.3 | 57.2±2.5 |
| dSprites-loc (Matthey et al., 2017) | 16 | 6.4±0.5 | 7.9±3.4 | 22.7±2.6 | 37.6±5.0 | 66.8±5.2 |
| dSprites-ori (Matthey et al., 2017) | 16 | 7.9±2.1 | 11.1±3.7 | 13.5±6.5 | 19.9±2.9 | 50.1±1.1 |
| SmallNORB-azi (LeCun et al., 2004) | 18 | 5.9±0.7 | 7.9±0.8 | 8.5±0.4 | 11.4±2.3 | 18.3±0.7 |
| SmallNORB-elev (LeCun et al., 2004) | 9 | 14.5±1.0 | 17.0±3.8 | 18.7±4.4 | 26.7±0.1 | 38.3±2.9 |
| DMLab (Beattie et al., 2016) | 6 | 29.2±1.3 | 32.7±1.4 | 35.4±2.0 | 39.3±1.1 | 51.5±1.9 |
| KITTI-dist (Geiger et al., 2013) | 4 | 41.7±5.8 | 51.2±3.4 | 57.9±8.6 | 68.9±0.3 | 76.0±0.7 |
| All | | 26.1 | 34.3 | 43.8 | 51.8 | 71.3 |
| Natural | | 14.7 | 29.0 | 45.1 | 56.3 | 84.8 |
| Specialized | | 59.4 | 64.2 | 71.0 | 73.6 | 86.4 |
| Structured | | 19.3 | 24.1 | 29.1 | 36.0 | 52.0 |

Figs. 19 and 20 depict the complete set of VTAB-1k accuracy results as a function of dataset, privacy level ($\epsilon$), backbone, and learnable parameters. The datasets are ordered increasingly by transfer difficulty (TD). Although classifiers for the Retinopathy dataset appear to perform equally well independently of $\epsilon$, a closer inspection reveals that this dataset is unbalanced and learned classifiers predict the most common class in all settings.

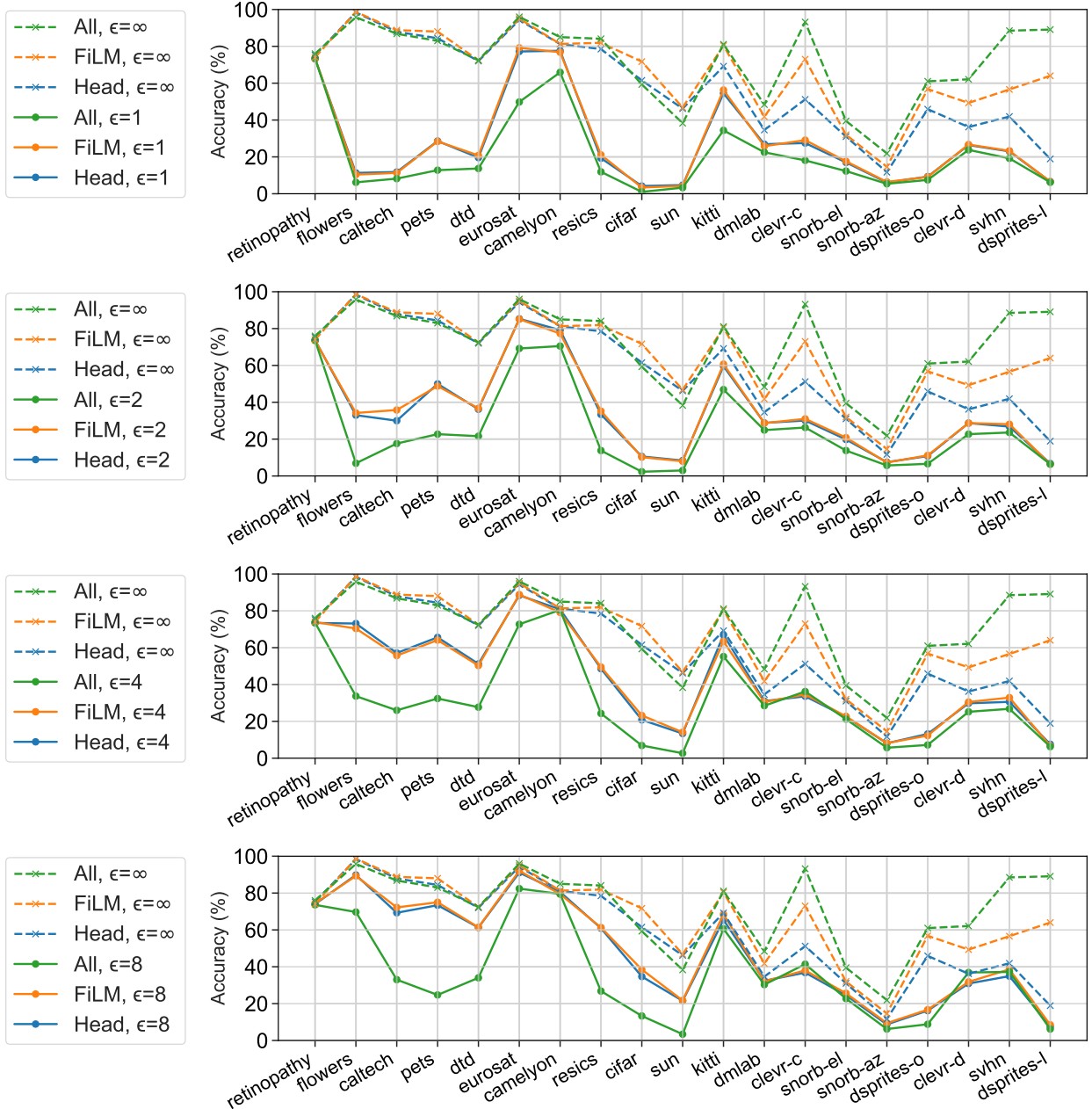

Figure 19: Classification accuracy for VTAB-1k datasets as a function of privacy level ($\epsilon$) and learnable parameters. Backbone is R-50. Dashed lines in all plots indicate non-private accuracy as a reference. The datasets are ordered increasingly by transfer difficulty (TD).

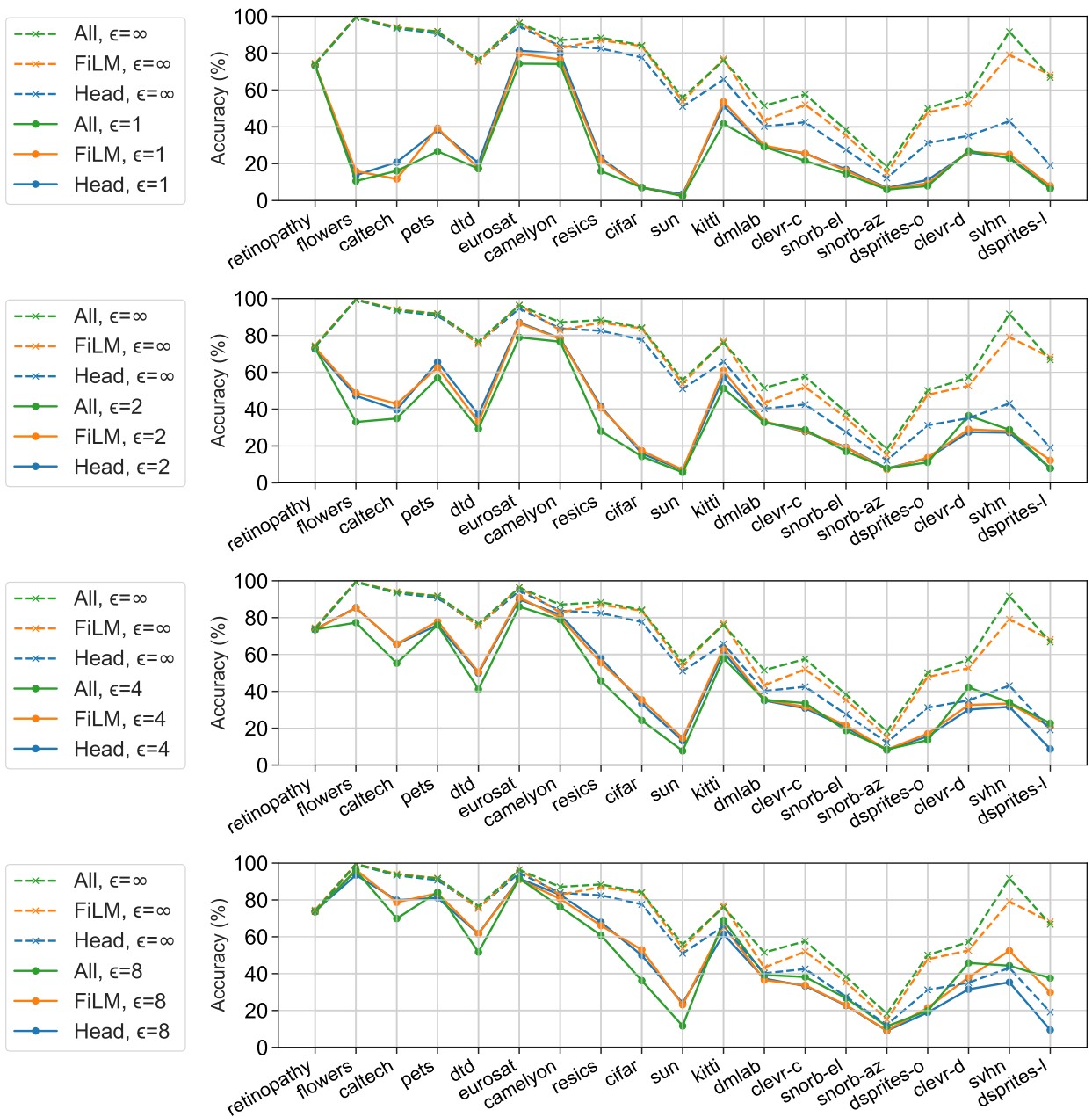

Figure 20: Classification accuracy for VTAB-1k datasets as a function of privacy level ($\epsilon$) and learnable parameters. Backbone is ViT-B. Dashed lines in all plots indicate non-private accuracy as a reference. The datasets are ordered increasingly by transfer difficulty (TD).

Fig. 21 depicts the final training and test accuracy as a function of $\epsilon$ and learnable parameters for all 19 VTAB-1k datasets with the ViT-B backbone. Although classifiers for the Retinopathy dataset appear to perform equally well independently of $\epsilon$, a closer inspection reveals that this dataset is unbalanced and learned classifiers predict the most common class in all settings.

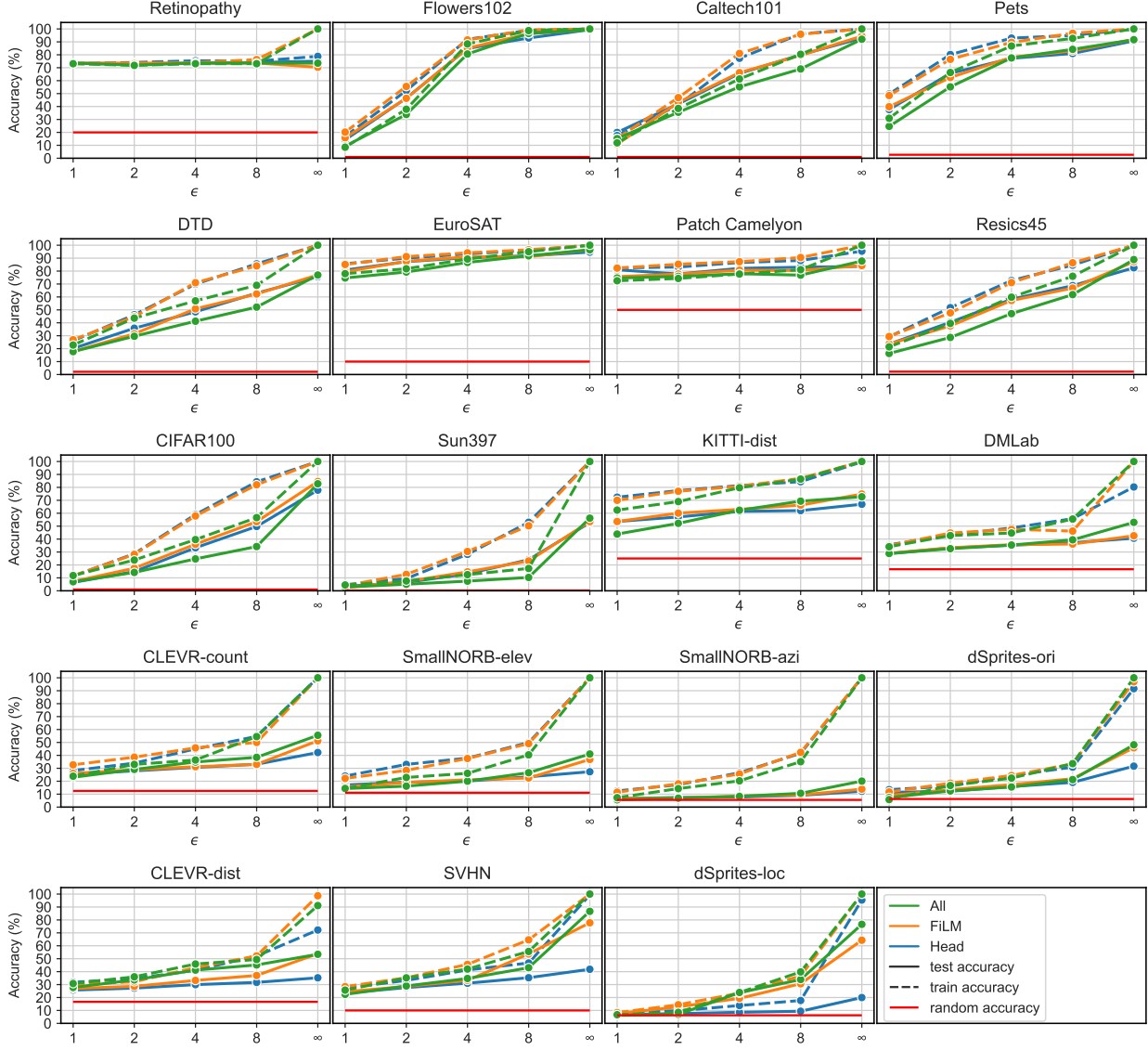

Figure 21: Test and train classification accuracy as a function of $\epsilon$ and learnable parameters (*All*, *FiLM* and *Head*) on VIT-B for all VTAB datasets with $\delta = 1/|\mathcal{D}|$. The accuracy is reported for the median run of Tables 11 to 13. The datasets are in order of increasing transfer difficulty (TD) from left-to-right and top-to-bottom.

### A.2.7 Additional Membership Inference Attack Results

Fig. 22 depicts the complete set of ROC curves for LiRA on CIFAR-100 with the R-50 backbone for various privacy levels ($\epsilon$) and learnable parameters *Head* and *FiLM* at a fixed $S$.

Fig. 23 depicts the complete set of ROC curves for LiRA on CIFAR-100 with the R-50 backbone for various shots $S$ at fixed privacy levels ($\epsilon$) and learnable parameters *Head* and *FiLM*.

Table 14 presents the True Positive Rates (TPR) at various False Positive Rates (FPR), Area Under Receiver Operating Curve (AUC), and Attack Advantage (Attack Adv) (Yeom et al., 2018) for various privacy levels ($\epsilon$) and shots per class (S) corresponding to the plots in Figs. 22 and 23.

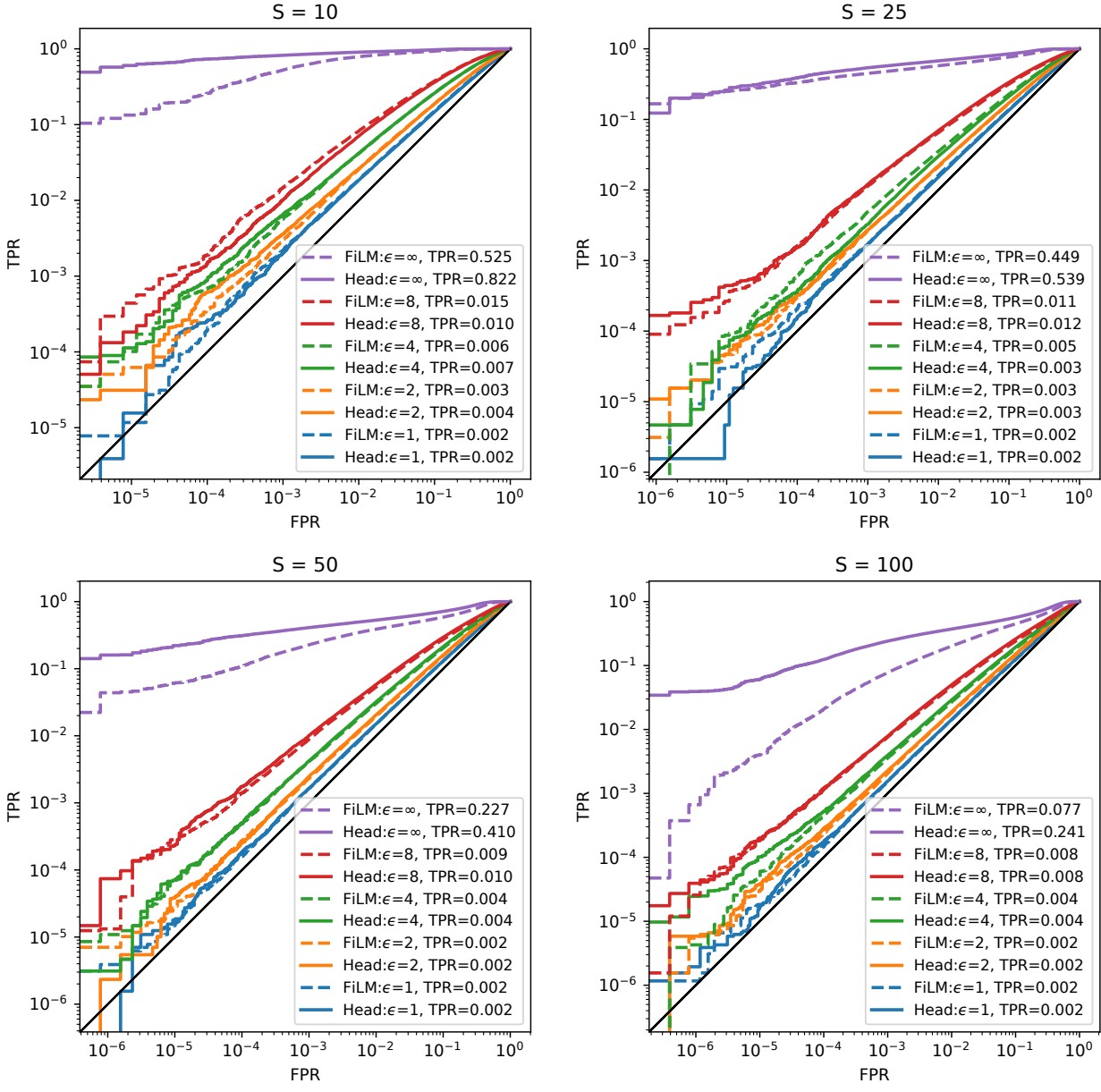

Figure 22: ROC curves for LiRA (Carlini et al., 2022) on CIFAR-100 with R-50 backbone for various privacy levels ($\epsilon$) and backbone configurations *Head* and *FiLM* at a fixed $S$. TPR values in legends are measured at FPR=0.001.

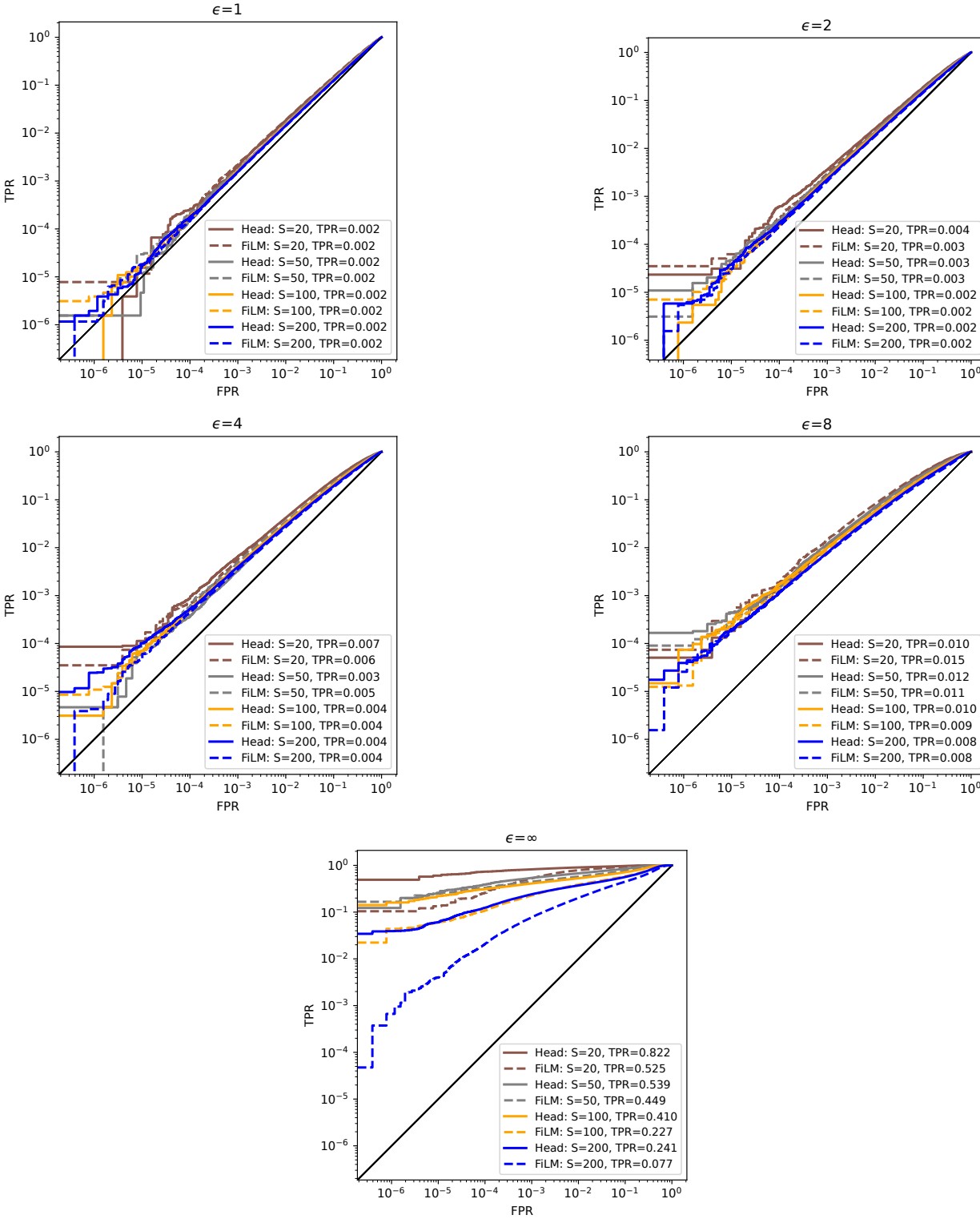

Figure 23: ROC curves for LiRA (Carlini et al., 2022) on CIFAR-100 with R-50 backbone for various $S$ at fixed privacy levels ($\epsilon$) and backbone configurations *Head* and *FiLM*. TPR values in legends are measured at FPR=0.001.

Table 14: True Postive Rates (TPR) at various False Positive Rates (FPR), Area Under Receiver Operating Curve (AUC), and Attack Advantage (Yeom et al., 2018) for various privacy levels ($\epsilon$) and shots per class (S) corresponding to the plots in Figs. 22 and 23. Dataset ($\mathcal{D}$) is CIFAR-100. Backbone is R-50 pretrained on ImageNet-21k.

| $\epsilon$ | $S$ | TPR (%) @ 0.1% FPR | | TPR (%) @ 1% FPR | | TPR (%) @ 10% FPR | | AUC | | Attack Adv | |
|---|---|---|---|---|---|---|---|---|---|---|---|
| | | Head | FiLM | Head | FiLM | Head | FiLM | Head | FiLM | Head | FiLM |
| | 10 | 0.20 | 0.22 | 1.85 | 1.88 | 14.71 | 15.10 | 0.564 | 0.572 | 0.092 | 0.106 |
| 1 | 25 | 0.17 | 0.17 | 1.52 | 1.61 | 13.51 | 13.77 | 0.550 | 0.552 | 0.070 | 0.074 |
| | 50 | 0.16 | 0.16 | 1.50 | 1.50 | 13.04 | 12.99 | 0.541 | 0.541 | 0.058 | 0.058 |
| | 100 | 0.16 | 0.15 | 1.43 | 1.41 | 12.54 | 12.26 | 0.535 | 0.531 | 0.049 | 0.042 |
| | 10 | 0.36 | 0.30 | 2.62 | 2.56 | 18.76 | 18.83 | 0.610 | 0.613 | 0.164 | 0.166 |
| 2 | 25 | 0.27 | 0.27 | 2.19 | 2.12 | 17.12 | 16.83 | 0.593 | 0.593 | 0.138 | 0.138 |
| | 50 | 0.25 | 0.23 | 2.05 | 1.96 | 15.95 | 15.46 | 0.579 | 0.573 | 0.115 | 0.106 |
| | 100 | 0.23 | 0.21 | 1.89 | 1.81 | 15.11 | 14.47 | 0.566 | 0.557 | 0.094 | 0.080 |
| | 10 | 0.65 | 0.56 | 4.20 | 4.14 | 26.06 | 26.19 | 0.678 | 0.677 | 0.262 | 0.260 |
| 4 | 25 | 0.33 | 0.49 | 3.00 | 3.53 | 21.07 | 22.77 | 0.637 | 0.646 | 0.203 | 0.214 |
| | 50 | 0.42 | 0.41 | 3.21 | 3.02 | 21.27 | 20.28 | 0.629 | 0.617 | 0.187 | 0.167 |
| | 100 | 0.39 | 0.36 | 2.90 | 2.68 | 19.33 | 18.30 | 0.606 | 0.595 | 0.148 | 0.131 |
| | 10 | 1.01 | 1.47 | 7.06 | 8.23 | 36.20 | 37.58 | 0.748 | 0.753 | 0.370 | 0.378 |
| 8 | 25 | 1.20 | 1.14 | 6.95 | 6.47 | 33.49 | 31.66 | 0.717 | 0.702 | 0.316 | 0.294 |
| | 50 | 1.00 | 0.88 | 5.81 | 5.33 | 29.40 | 27.31 | 0.688 | 0.667 | 0.267 | 0.234 |
| | 100 | 0.78 | 0.76 | 5.00 | 4.62 | 26.01 | 23.98 | 0.660 | 0.636 | 0.221 | 0.180 |
| | 10 | 82.22 | 52.50 | 90.37 | 78.78 | 97.17 | 93.35 | 0.992 | 0.981 | 0.905 | 0.846 |
| $\infty$ | 25 | 53.92 | 44.88 | 67.53 | 57.58 | 84.76 | 76.60 | 0.959 | 0.930 | 0.748 | 0.666 |
| | 50 | 41.00 | 22.72 | 52.96 | 38.84 | 71.46 | 58.63 | 0.913 | 0.854 | 0.616 | 0.491 |
| | 100 | 24.09 | 7.72 | 37.19 | 20.15 | 56.44 | 44.80 | 0.845 | 0.777 | 0.472 | 0.362 |

### A.2.8 Additional Federated Learning Results

Table 15 shows the non-private performance on the FLAIR dataset, while Table 16 shows the same performance under DP guaranties with $\epsilon = 2$.

Table 15: Non-private Federated Learning performance on FLAIR as a function of backbone $b_\theta$ and learnable parameters. C stands for averaged per-class metrics (Macro) and O denotes overall metrics (Micro). P, R and AP denote precision, recall, and average precision, respectively. The R-18 *All* result is taken from the original paper Song et al. (2022). Due to the significant computational requirements, only a single random seed was used in all experiments on FLAIR.

| $b_\theta$ | | $\epsilon$ | C-P | O-P | C-R | O-R | C-F1 | O-F1 | C-AP | O-AP |
|---|---|---|---|---|---|---|---|---|---|---|
| | All | $\infty$ | 71.8 | 83.5 | 48.6 | 76.0 | 58.0 | 79.5 | 62.1 | 88.8 |
| R-18 | FiLM | $\infty$ | 73.8 | 82.0 | 44.8 | 74.4 | 55.7 | 78.0 | 59.7 | 87.7 |
| | Head | $\infty$ | 71.0 | 79.9 | 43.8 | 72.9 | 54.1 | 76.2 | 57.9 | 85.8 |
| | All | $\infty$ | 76.9 | 85.2 | **62.0** | 82 | 68.6 | 83.6 | 72.3 | 91.9 |
| R-50 | FiLM | $\infty$ | 78.3 | 83.8 | 57.9 | 80.0 | 66.6 | 81.9 | 70.2 | 90.6 |
| | Head | $\infty$ | 76 | 82.3 | 42.7 | 71.3 | 54.6 | 76.4 | 60.5 | 86.7 |
| | All | $\infty$ | 79.6 | **86.8** | 57.4 | **82.9** | 66.7 | **84.8** | 72.9 | **93.1** |
| VIT-B | FiLM | $\infty$ | **81.9** | **86.8** | 59.3 | 81.6 | **68.8** | 84.1 | **74.7** | 92.7 |
| | Head | $\infty$ | 81.6 | 83.7 | 52 | 72.2 | 63.4 | 77.5 | 70.0 | 87.6 |

Table 16: Federated Learning performance on FLAIR under DP with $\epsilon = 2$ as a function of backbone $b_\theta$ and learnable parameters. C stands for averaged per-class metrics (Macro) and O denotes overall metrics (Micro). P, R and AP denote precision, recall, and average precision, respectively. The R-18 *All* result is taken from the original paper Song et al. (2022). Due to significant computational requirements, only single random seed was used in all experiments with FLAIR.

| $b_\theta$ | | $\epsilon$ | C-P | O-P | C-R | O-R | C-F1 | O-F1 | C-AP | O-AP |
|---|---|---|---|---|---|---|---|---|---|---|
| | All | 2 | 47.3 | 77.5 | 32.3 | 64.3 | 38.4 | 70.3 | 44.3 | 80.2 |
| R-18 | FiLM | 2 | 59.0 | 81.0 | 39.1 | 70.3 | 47.0 | 75.3 | 51.9 | 85.2 |
| | Head | 2 | 47.6 | 81.4 | 34.2 | 66.4 | 39.8 | 73.1 | 47.2 | 83.4 |
| | All | 2 | 56.2 | 83.1 | 38.1 | 70.9 | 45.4 | 76.6 | 52.3 | 86.6 |
| R-50 | FiLM | 2 | 59.7 | 79.3 | 39.4 | 69.9 | 47.5 | 74.3 | 51.3 | 84.2 |
| | Head | 2 | 57.0 | 79.8 | 38.0 | 68.5 | 45.6 | 73.7 | 50.4 | 83.8 |
| | All | 2 | 47.8 | 82.3 | 37.5 | 71.0 | 42.1 | 76.2 | 49.7 | 86.1 |
| VIT-B | FiLM | 2 | 58.1 | **84.2** | **42.5** | **76** | 49.1 | **79.9** | 57.2 | **89.2** |
| | Head | 2 | **67.1** | 83.4 | 39.8 | 68.9 | **50.0** | 75.5 | **59.0** | 85.9 |

**CIFAR-100 and Federated EMNIST** Additionally, we perform experiments on CIFAR-100 and Federated EMNIST, which are commonly used to benchmark FL methods. We opt for these datasets as they have different degree of TD: CIFAR-100 has medium TD, while Federated EMNIST had high TD. For CIFAR-100, we use 500 training clients and 100 test clients, with each client having 100 samples and no clients sharing any data. To introduce more client heterogeneity, the data are distributed using the Pachinko Allocation Method (Li & McCallum, 2006) as in Reddi et al. (2021). Federated EMNIST Caldas et al. (2018) is a dataset of black-and-white handwritten symbols from 62 classes grouped according to the writer. EMNIST is a highly out-of-distribution dataset (i.e. high TD) with respect to the ImageNet-21K pretraining data. As the number of training users in CIFAR-100 (500 users) and Federated EMNIST (3400 users) is relatively low, we need to increase $\epsilon$ from 2 to 8, such that the amount of added noise during aggregation is not excessive. $\delta$ is set to $N^{-1.1}$, where $N$ is the number of training clients. For CIFAR-100 and Federated EMNIST, we report standard test classification accuracy. All training details and hyperparameters are in Appendix A.3.6.

Fig. 24 shows the performance of different model configurations on CIFAR-100 and Federated EMNIST with and without DP. Table 17 illustrates private with $\epsilon = 8$ and non-private performance on CIFAR-100 and Federated EMNIST. These tables present a tabular version of the results in Fig. 9.

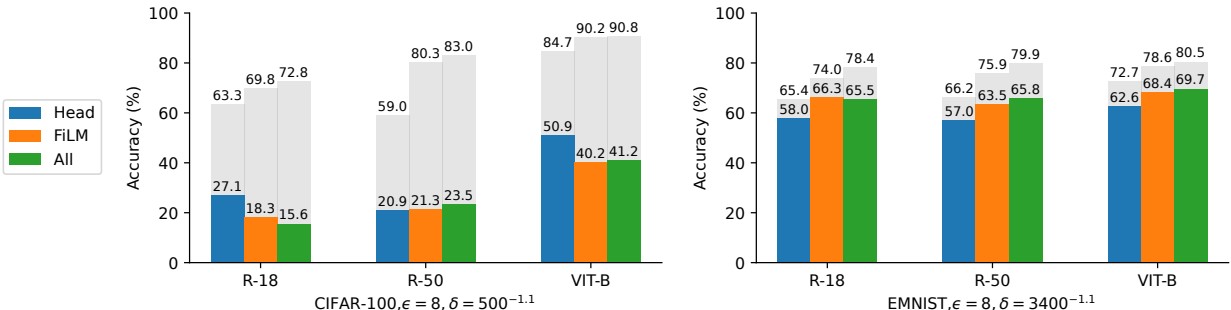

Figure 24: Private ($\epsilon = 8$, colored) and non-private ($\epsilon = \infty$, gray) FL performance on CIFAR-100 (left) and Federated EMNIST (right) as a function of backbone and learnable parameters. We report accuracy on test clients. R-18 backbone is pretrained on ImageNet-1k, VIT-B and R-50 are pretrained on ImageNet-21k.

Table 17: Federated Learning performance on CIFAR-100 and EMNIST with ($\epsilon = 8$) and without ($\epsilon = \infty$) DP as a function of backbone $b_\theta$ and learnable parameters. Accuracy (in %) is reported. R-18 backbone is pretrained on ImageNet-1k, VIT-B and R-50 are pretrained on ImageNet-21k. The $\pm$ sign indicates the 95% confidence interval over 3 runs with different seeds.

| Dataset | $\epsilon$ | R-18 | | | R-50 | | | VIT-B | | |
| | | Head | FiLM | All | Head | FiLM | All | Head | FiLM | All |
|---|---|---|---|---|---|---|---|---|---|---|
| CIFAR-100 | $\infty$ | 63.3±0.2 | 69.8±0.3 | 72.8±0.7 | 59.1±0.5 | 79.8±0.5 | 83.0±0.1 | 84.6±0.1 | 90.2±0.3 | **90.8±0.3** |
| | 8 | 27.1±1.4 | 18.3±0.9 | 15.6±1.0 | 20.9±0.6 | 21.3±1.0 | 23.5±1.3 | **50.8±0.1** | 40.2±2.3 | 41.2±3.4 |
| EMNIST | $\infty$ | 65.4±0.1 | 74.0±0.9 | 78.4±1.1 | 66.2±0.4 | 75.9±0.4 | 79.9±0.4 | 72.7±0.2 | 78.6±0.1 | **80.5±0.1** |
| | 8 | 58.0±0.4 | 66.3±0.5 | 65.5±0.2 | 57.0±0.3 | 63.5±0.1 | 65.8±0.3 | 62.6±0.1 | 68.4±0.2 | **69.7±0.3** |

### A.3 Training and Evaluation Details

### A.3.1 FiLM Layer Implementation

Table 18 details the locations and count of the parameters that are updateable for the *FiLM* configuration in each of the backbones used in the experiments.

Table 18: Backbone parameter count, FiLM parameter count, FiLM parameter count as a percentage of the backbone parameter count, and FiLM parameter locations within the backbone for each of the backbones used in the experiments.

| Backbone | Backbone Count | FiLM Count | FiLM (%) | Locations |
|---|---|---|---|---|
| R-18 | 11.2M | 7808 | 0.07 | GroupNorm Scale and Bias that follows each 3x3 Conv layer |
| R-50 | 23.5M | 11648 | 0.05 | GroupNorm Scale and Bias that follows each 3x3 Conv layer
Final GroupNorm Scale and Bias before Head |
| VIT-B | 85.8M | 38400 | 0.04 | All LayerNorm Scale and Bias |

### A.3.2 Hyperparameter Tuning

For all centralized experiments, we first draw $\mathcal{D}$ of the required size ($|\mathcal{D}| = CS$, or $|\mathcal{D}| = 1000$ in the case of VTAB-1k) from the entire training split of the current dataset under evaluation. For the purposes of hyperparameter tuning, we then split $\mathcal{D}$ into 70% train and 30% validation. We then perform 20 iterations of hyperparameter tuning using the tree-structured parzen estimator (Bergstra et al., 2011) strategy as implemented in Optuna (Akiba et al., 2019) to derive a set of hyperparameters that yield the highest accuracy on the validation split. This set of parameters are subsequently used to train a final model on all of $\mathcal{D}$. We the evaluate the final, tuned model on the entire test split of the current dataset. Details on the set of hyperparameters that are tuned and their ranges can be found in Table 19. For DP training, we compute the required noise multiplier depending on the target $(\epsilon, \delta)$-DP guarantee. The hyperparameter ranges are purposely broad and have been empirically derived. We fine-tune models for at most 200 epochs to limit the amount of compute necessary.

Table 19: Hyperparameter ranges used for the Bayesian optimization.

| | lower bound | upper bound |
|---|---|---|
| EPOCHS | 1 | 200 |
| LEARNING RATE | 1E-7 | 1E-2 |
| BATCH SIZE | 10 | $|\mathcal{D}|$ |
| CLIPPING NORM | 0.2 | 10 |
| NOISE MULTIPLIER | BASED ON TARGET $\epsilon$ | |

### A.3.3 Effect of Shots per Class and $\epsilon$ Experiments

For each evaluated configuration, we draw $|\mathcal{D}| = CS$ examples from the dataset training split, tune hyperparameters as described in Appendix A.3.2, and then test on the entire test split of the dataset. We use the DP-Adam optimizer as implemented in Opacus (Yousefpour et al., 2021) for all private experiments. For non-private experiments, we used the Adam (Kingma & Ba, 2015) optimizer for the *Head* and *FiLM* parameter configurations and the SGD optimizer for the *All* configuration. No data augmentation was used and images were scaled to 224×224 pixels.

All of the effect of $S$ and $\epsilon$ experiments were carried out on 1 (for *Head* and *FiLM*) and up to 3 (for *All*) NVIDIA V100 GPUs with 32GB of memory. The runtime for executing the whole experiment depends on the the size of the few-shot training set and the number of parameters resulting from the choice of the backbone

and the number of learnable parameters ($All > FiLM > Head$). For CIFAR-10 and SVHN the runtime for one configuration ranges from less than 5 GPU minutes ($S = 1 + Head$) to 60 GPU hours ($S = 500 + All$). For CIFAR-100, the range is from 15 GPU minutes ($S = 1 + Head$) to over 700 GPU hours ($S = 500 + All$).

### A.3.4 VTAB-1k Experiments

For each evaluated configuration of each of the 19 datasets in the VTAB-1k benchmark, we draw $|\mathcal{D}| = 1000$ examples from the dataset training split, tune hyperparameters as described in Appendix A.3.2, and then test on the entire test split of the dataset. We use the DP-Adam optimizer as implemented in Opacus (Yousefpour et al., 2021) for all private experiments. For non-private experiments, we used the Adam (Kingma & Ba, 2015) optimizer for the *Head* and *FiLM* parameter configurations and the SGD optimizer for the *All* configuration.

No data augmentation was used. For the R-50 backbone, images were scaled to 384×384 pixels unless the image size was 32×32 pixels or less, in which case the images were scaled to 224×224 pixels. For the VIT-B backbone, images were scaled to 224×224 pixels.

All of the VTAB-1k transfer learning experiments were carried out on a single NVIDIA A100 GPU with 80GB of memory. Processing times for each configuration of each dataset will vary with the selected hyperparameters and the size of the test split, but approximate times are listed in Table 20.

Table 20: Approximate time to tune, train, and test a single configuration of parameters on a single VTAB-1k dataset for various backbones and parameter configurations. Units are wall clock GPU hours.

| Backbone | Parameter Configuration | | |
|---|---|---|---|
| | **None** | **FiLM** | **All** |
| **R-50** | 0.6 | 0.9 | 2.7 |
| **VIT-B** | 1.3 | 2.4 | 6.5 |

### A.3.5 Membership Inference Attacks Experiments

For each setting of $S$ and $\epsilon$, we first sample $2|\mathcal{D}|$ examples (recall $|\mathcal{D}| = CS = 100S$) from the CIFAR-100 training set, and then train 257 different models (1 target model plus 256 shadow models) where each sample for the training set is randomly selected with 50% probability from the $2|\mathcal{D}|$ examples. This ensures that approximately half of the models are trained on each example and half are not so that we can create distributions over the losses for each example being in and out of the training set as described in the LiRA algorithm (Carlini et al., 2022). We use each of the trained models in turn as the target model and then accumulate the attack predictions over all 257 targets to produce the ROC curve for the attack. Due to the extreme computation demand in training a large number of shadow models for each setting of $S$ and $\epsilon$, we restrict the attacks to the R-50 backbone and the *Head* and *FiLM* parameter configurations.

Our implementation is based on code from the TensorFlow Privacy library (Google, 2019b). All of the VTAB-1k transfer learning experiments were carried out on a single NVIDIA A100 GPU with 80GB of memory. When training the 257 models for each attack configuration, we do not perform hyperparameter tuning, instead we used the hyperparameter set from the CIFAR-100 experiments in Table 3 that yielded the highest accuracy for the particular configuration. Approximate training times for all 257 models in each configuration are listed on Table 21. The value of $\epsilon$ did not alter the training times to a significant degree.

### A.3.6 Federated Learning Experiments

All experiments were performed in TensorFlow using tensorflow-federated Google (2019a) for federated aggregation and tensorflow-privacy Google (2019b) for privacy accounting and the adaptive clipping algorithm Andrew et al. (2021). CIFAR-100 and Federated EMNIST datasets were taken from tensorflow-federated.

**FLAIR** Each model configuration is trained for 5000 rounds with a cohort size of 200. Each sampled user trains the model locally with SGD for 2 epochs with local batch size set to 16. The maximum number of

Table 21: Approximate time to train 257 models for a single configuration of parameters for a LiRA attack on the CIFAR-100 dataset for various parameter and shot configurations. Units are wall clock GPU hours.

| | Shot (S) | | | |
|---|---|---|---|---|
| **Parameter Configuration** | **10** | **25** | **50** | **100** |
| Head | 6 | 12 | 16 | 46 |
| FiLM | 8 | 25 | 49 | 96 |

images for each user is set to 512. For DP, $\epsilon = 2, \delta = N^{-1.1}$, where $N$ is the number of training users. As in the original paper, we set L2 norm quantile to 0.1 for adaptive clipping and we use 200 users sampled uniformly per round to simulate the noise-level with a cohort size of 5000.

For the non-private setting we perform the grid search over:

- server learning rate $\in \{0.01, 0.05, 0.1\}$

- client learning rate $\in \{0.01, 0.05, 0.1\}$

For the private setting ($\epsilon = 2$) we fixed the client learning rate to the optimal value found for the non-private run and a perform grid search over the server learning rate in the set $\{a/2, a/10, a/50, a/100\}$, where $a$ is the optimal server learning rate found for the non-private setting.

Processing times for each configuration on FLAIR are given in Table 22.

Table 22: Approximate time to train and test a single configuration of parameters on FLAIR dataset for various backbones and parameter configurations. Units are wall clock GPU hours.

| | Parameter Configuration | | |
|---|---|---|---|
| **Backbone** | **Head** | **FiLM** | **All** |
| **R-18** | 18 | 30 | - |
| **R-50** | 30 | 43 | 60 |
| **VIT-B** | 40 | 60 | 75 |

**CIFAR-100 and Federated EMNIST**  Each model configuration is trained for 500 rounds with a cohort size of 20. Each sampled user trains the model locally with SGD for 5 epochs with local batch size set to 100. The maximum number of images for each user is set to 512. For DP, $\epsilon = 8, \delta = N^{-1.1}$, where $N$ is the number of training users ($N = 500$ for CIFAR-100, $N = 3400$ for Federated EMNIST). As in the original paper, we set L2 norm quantile to 0.1 for adaptive clipping and we use 20 users sampled uniformly per round to simulate the noise-level with a cohort size of 100.

For the non-private setting we perform the grid search over:

- server learning rate $\in \{0.05, 0.1, 0.5\}$

- client learning rate $\in \{0.01, 0.05, 0.1\}$

For the private setting ($\epsilon = 8$) we fixed the client learning rate to the optimal value found for the non-private run and perform a grid search over:

- server learning rate $\in \{a/2, a/10, a/50, a/100\}$, where $a$ is the optimal server learning rate found for the non-private setting.

- quantile for adaptive clipping bound $\in \{0.1, 0.5, 0.8\}$

### A.3.7 On the $(\epsilon, \delta)$-DP accounting

In the centralized experiments we compute the $(\epsilon, \delta)$-DP guarantees using the RDP accountant (Mironov, 2017) with $\delta = 1/|\mathcal{D}|$ where $\mathcal{D}$ where $|\mathcal{D}| = CS$ (i.e. the number of classes $C$ multiplied by shot $S$). Setting $\delta = 1/|\mathcal{D}|$ is a standard choice and simplifies comparisons with other papers. To allow for an easier comparison among different $|\mathcal{D}|$ we provide Table 23 which illustrates the change of $\epsilon$ computed using the RDP accountant for $\delta = 1e^{-5}$.

Additionally, we recompute the $(\epsilon, \delta)$-DP guarantees with the PRV accountant (Gopi et al., 2021), which is a accurate numerical accountant and results in slightly smaller $\epsilon$ than the RDP accountant given the same privacy parameters and $\delta$. Table 24 shows the results for that.

Table 23: Recomputed $\epsilon$ at $\delta = 1e^{-5}$ as a function of $S$ for the datasets CIFAR-10, CIFAR-100 and SVHN and original $\epsilon \in \{1, 2, 4, 8\}$ that was computed originally at $\delta = 1/|\mathcal{D}|$. The computation is done using the RDP accountant (Mironov, 2017) provided in opacus (Yousefpour et al., 2021). The ranges of $\epsilon$ result from the fact that there is not a direct mapping from the original $\epsilon$ to the recomputed $\epsilon$ but the recomputed $\epsilon$ depends on the used privacy parameters (noise multiplier, subsampling ratio and number of steps).

| original $\epsilon$ | | $1S$ | $5S$ | $10S$ | $25S$ | $50S$ | $100S$ | $250S$ | $500S$ |
|---|---|---|---|---|---|---|---|---|---|
| CIFAR-10 | 1 | 3.30-3.32 | 2.20-2.33 | 1.94-2.20 | 1.69-1.71 | 1.54-1.56 | 1.43-1.46 | 1.30-1.34 | 1.22-1.24 |
| | 2 | 5.41-5.43 | 3.95-4.49 | 3.56-3.60 | 3.18-3.22 | 2.95-2.97 | 2.76-2.92 | 2.54-2.66 | 2.41-2.50 |
| | 4 | 9.14-9.16 | 7.14-8.25 | 6.57-6.78 | 5.99-6.11 | 5.61-5.73 | 5.31-5.68 | 4.96-5.31 | 4.73-4.87 |
| | 8 | 15.80-15.82 | 13.02-13.84 | 12.19-13.41 | 11.35-11.71 | 10.72-11.57 | 10.25-10.83 | 9.67-10.50 | 9.28-9.51 |
| CIFAR-100 | 1 | 1.94-1.97 | 1.54-1.65 | 1.43-1.44 | 1.30-1.34 | 1.22-1.23 | 1.16-1.17 | 1.08-1.09 | 1.03-1.04 |
| | 2 | 3.56-3.64 | 2.95-2.98 | 2.75-2.77 | 2.54-2.55 | 2.41-2.42 | 2.29-2.30 | 2.16-2.17 | 2.07-2.08 |
| | 4 | 6.60-6.81 | 5.63-5.73 | 5.32-5.42 | 4.96-5.03 | 4.73-4.94 | 4.53-4.55 | 4.29-4.30 | 4.14-4.15 |
| | 8 | 12.26-12.74 | 10.76-10.86 | 10.25-10.48 | 9.66-9.85 | 9.28-9.39 | 8.94-9.03 | 8.52-8.55 | 8.25-8.29 |
| SVHN | 1 | 3.30-3.32 | 2.19-2.33 | 1.94-1.96 | 1.69-1.71 | 1.54-1.56 | 1.43-1.50 | 1.30-1.31 | 1.22-1.24 |
| | 2 | 5.41-5.43 | 3.95-4.03 | 3.56-3.62 | 3.17-3.23 | 2.94-2.97 | 2.75-2.91 | 2.54-2.56 | 2.41-2.43 |
| | 4 | 9.14-9.16 | 7.14-7.44 | 6.60-7.52 | 5.99-6.50 | 5.62-5.93 | 5.32-5.68 | 4.96-5.01 | 4.73-4.77 |
| | 8 | 15.80-15.82 | 13.03-13.73 | 12.19-12.72 | 11.34-11.85 | 10.76-11.02 | 10.25-10.45 | 9.67-9.82 | 9.28-9.40 |

Table 24: Recomputed $\epsilon$ at $\delta = 1e^{-5}$ as a function of $S$ for the datasets CIFAR-10, CIFAR-100 and SVHN and original $\epsilon \in \{1, 2, 4, 8\}$ that was computed originally at $\delta = 1/|\mathcal{D}|$. The computation is done using the PRV accountant (Gopi et al., 2021) provided in opacus (Yousefpour et al., 2021). The ranges of $\epsilon$ result from the fact that there is not a direct mapping from the original $\epsilon$ to the recomputed $\epsilon$ but the recomputed $\epsilon$ depends on the used privacy parameters (noise multiplier, subsampling ratio and number of steps).

| original $\epsilon$ | | $1S$ | $5S$ | $10S$ | $25S$ | $50S$ | $100S$ | $250S$ | $500S$ |
|---|---|---|---|---|---|---|---|---|---|
| CIFAR-10 | 1 | 3.05-3.07 | 2.04-2.13 | 1.79-1.97 | 1.56-1.57 | 1.43-1.44 | 1.32-1.34 | 1.20-1.22 | 1.13-1.14 |
| | 2 | 5.02-5.04 | 3.66-4.04 | 3.30-3.33 | 2.94-2.96 | 2.73-2.74 | 2.55-2.62 | 2.35-2.38 | 2.23-2.24 |
| | 4 | 8.52-8.54 | 6.64-7.45 | 6.11-6.24 | 5.56-5.64 | 5.21-5.28 | 4.93-5.11 | 4.60-4.68 | 4.39-4.41 |
| | 8 | 14.80-14.81 | 12.17-12.74 | 11.39-12.17 | 10.57-10.79 | 10.00-10.48 | 9.54-9.83 | 9.00-9.08 | 8.63-8.68 |
| CIFAR-100 | 1 | 1.79-1.81 | 1.43-1.48 | 1.32-1.33 | 1.20-1.22 | 1.13-1.14 | 1.07-1.08 | 1.00-1.01 | 0.96-0.96 |
| | 2 | 3.30-3.35 | 2.73-2.75 | 2.55-2.56 | 2.35-2.36 | 2.23-2.24 | 2.12-2.13 | 2.00-2.00 | 1.92-1.92 |
| | 4 | 6.12-6.27 | 5.22-5.28 | 4.93-4.98 | 4.60-4.62 | 4.39-4.46 | 4.20-4.20 | 3.98-3.99 | 3.83-3.84 |
| | 8 | 11.43-11.74 | 10.02-10.08 | 9.54-9.65 | 9.00-9.06 | 8.63-8.66 | 8.31-8.32 | 7.92-7.93 | 7.61-7.68 |
| SVHN | 1 | 3.05-3.07 | 2.03-2.13 | 1.79-1.81 | 1.56-1.58 | 1.43-1.44 | 1.32-1.35 | 1.20-1.21 | 1.13-1.14 |
| | 2 | 5.02-5.04 | 3.66-3.71 | 3.30-3.34 | 2.94-2.97 | 2.72-2.74 | 2.55-2.62 | 2.35-2.36 | 2.23-2.24 |
| | 4 | 8.52-8.54 | 6.64-6.85 | 6.12-6.71 | 5.56-5.83 | 5.22-5.37 | 4.93-5.11 | 4.60-4.62 | 4.38-4.40 |
| | 8 | 14.80-14.81 | 12.18-12.65 | 11.39-11.73 | 10.57-10.86 | 10.02-10.16 | 9.54-9.64 | 9.00-9.05 | 8.64-8.66 |

