# OpenReview forum: "On the Efficacy of Differentially Private Few-shot Image Classification"
_TMLR — Accepted by TMLR_

### Review · Reviewer_RPAK · 2023-08-23

**Summary Of Contributions:**

This paper empirically studies the effect of different factors (including the number of shots per class, privacy level, model architecture, difficulty of downstream dataset, and learnable parameters) in the performance of DP few shot classification tasks.

**Audience:**

Yes

**Claims And Evidence:**

Yes

**Requested Changes:**

Please kindly refer to the comments in the section Weaknesses and make corresponding changes.

**Strengths And Weaknesses:**

Strengths:

1. The paper conducted comprehensive experiments to study the effects of different factors.
2. The authors seem to achieve the new SOTA results in DP few shot classification tasks.

Weaknesses:

1. I cannot quickly obtain the conclusions for each ablation study. It would be better for the authors to make a conclusion about the effect of each factor before illustrating the experimental results.
2. Why not try to use different adaptors? Besides, It would be better to report the performance and the amount of extra tunable parameters of the adaptors of different adaptors.
3. The metric of “transfer difficulty (TD)” could be problematic, which leads to possibly wrong conclusions. TD is related to the performance of ALL fine-tuning and HEAD fine-tuning. Fine-tuning, especially HEAD fine-tuning is sensitive to hyperparameters. Therefore, the TD can be affected by different hyperparameters.

---

> ### Author Response · Authors · 2023-08-29
>
> Thank you for taking the time to review our paper and providing insightful comments and questions. Responses below:
>
> **1. I cannot quickly obtain the conclusions for each ablation study. It would be better for the authors to make a conclusion about the effect of each factor before illustrating the experimental results.**
>
> This is an excellent suggestion. We will revise the paper and post the changes once all of the reviews have been received.
>
> **2. Why not try to use different adaptors? Besides, It would be better to report the performance and the amount of extra tunable parameters of the adaptors of different adaptors.**
>
> We agree that the paper would be improved by evaluating different adapters. However, we varied the amount of training data, privacy level, learnable parameter set, downstream dataset, and network architecture and in each experiment we tuned the hyperparameters (gradient clipping norm, epochs, batch size, and learning rate) with 20 iterations of a hyperparameter tuner (Optuna) and averaged the results of 3 runs. This consumed more than 50,000 GPU hours of computation time. Based on our experience, the specific type of adapters     used does not have a large effect on the overall trends that we observed (in comparison to the items that we did vary) and making a fair comparison on a reasonable set of adapters would have exceeded our computational resources.
>
> **3. The metric of “transfer difficulty (TD)” could be problematic, which leads to possibly wrong conclusions. TD is related to the performance of ALL fine-tuning and HEAD fine-tuning. Fine-tuning, especially HEAD fine-tuning is sensitive to hyperparameters. Therefore, the TD can be affected by different hyperparameters.**
>
> We agree that the TD metric is not ideal and a limitation, but in practice it aligns extremely well with the empirical difficulty of adapting to a downstream dataset. Indeed, TD is sensitive to the hyperparameters used. However, in all of our experiments, including the TD computation, we carefully tuned the hyperparameters using 20 iterations the Optuna hyperparameter tuner to make the calculation of the metric as robust as possible. We will add TD as a limitation in the revision of the paper once all the reviews have been received.

---

### Review · Reviewer_drm9 · 2023-08-30

**Summary Of Contributions:**

This paper looks at DP few-shot learning for the context of image classification. The authors provide an empirical evaluation of this setting. They demonstrate that, while it is feasible to achieve reasonable accuracy via fine-tuning pretrained models in a DP-few-shot setting, a larger number of examples is required compared to non-private settings.

**Audience:**

Yes

**Claims And Evidence:**

Yes

**Requested Changes:**

The authors should instead fix delta to something small such as $\delta=1e-5$ for all experiments and rerun all the experiments with the $\varepsilon$ corresponding to that $\delta$.

The authors should provide more comparison to other PEFT methods.

The paper needs to clearly state its contributions because it's very unclear what is being claimed by this work and what is shared with prior work.

**Strengths And Weaknesses:**

**Strengths**

The paper's experiments are well-detailed, comprehensive, and straightforward to replicate.
The paper is well-written and all terminology is clearly defined before use, making it easy to follow.
The subject matter is timely and important, because privacy is important and few-shot learning is especially relevant.

**Weaknesses**

*Major*

First and foremost the authors need to address the issue of the choice of $\delta$ in the experiments. This work sets $\delta = 1/dataset size$ but when the dataset size is small, such as CS=10 (which is the case in the key few-shot setting the authors are interested in) this means $\delta = 1/10$. That is, the DP training has a $10$% chance to 'catastrophically fail', or provide no privacy guarantees.

I feel the paper really lacks novelty. Most of the analysis and findings, such as the privacy-utility tradeoff and the need for more data in DP, has been covered in prior works. What lesson do we as the readers take away from this work? I really wonder what information a reader who is, e.g., a researcher in privacy or a practitioner in industry, would take away from this paper given all the limitations (which to give credit to the authors are clearly stated and discussed) and limited new findings.

While the paper does mention the effectiveness of parameter-efficient methods like FiLM, it does not compare these with other PEFT methods, limiting the depth of the analysis.

I will go over some specific claims in the paper and why I am not satisfied with them. The following claims are not incorrect, but they are known from prior work, so it's not clear why the paper's analysis is stating these as conclusions rather than referencing them from prior work.

"Additional data is required under DP" This point has already been made in well-known work ([1])

"Transfer learning under DP is fundamentally different from non-private" This point has already been made in many prior works, see [2]

"The point at which accuracy begins to improve varies with TD" This is known even non-privately, see the well-cited work done with the Stanford WILDS dataset [3]

[1] https://arxiv.org/abs/2011.11660

[2] https://arxiv.org/abs/2205.02973

[3] https://arxiv.org/pdf/2202.10054.pdf

"We see that learning under DP is fundamentally different from non-private."

Then there are some claims that can be interesting but are under-supported.

"As ϵ increases, the train accuracy grows as the amount of DP regularization is reduced, ultimately entering the interpolating regime" There are two issues with this statement. First, without knowing the specific hyperparameters searched for, it's not necessarily the case that the amount of noise (what I assume the authors mean by DP regularization) is reduced, it could be that the number of iterations is increased. Second, I don't see any evidence on the plots that the models are in the interpolating regime.

"For SVHN with ϵ = ∞, Head leaves the interpolating regime for S > 100, as there is not enough capacity to adapt to a high TD dataset. " It's not at all clear that this is an issue of capacity, because there is prior work in non-private (and even private) fine-tuning for linear probing (what the community would typically call the 'head' setting) that shows good performance on SVHN.

*Minor*

I didn't see an explicit definition for CS in the text. The term appears at the top of page 5. My understanding is that it's the number of classes times the number of shots.

---

> ### Author Response · Authors · 2023-09-14
>
> Thank you for taking the time to review our paper and providing insightful feedback. We will upload a new revision of the paper to address some of your points. The responses to your review are below:
>
> **1. The authors should instead fix delta to something small such as $\delta = 1e^{-5}$ for all experiments and rerun all the experiments with the corresponding to that $\delta$.**
>
> Setting $\delta = 1 /| \mathcal{D}|$ is a standard choice and simplifies comparisons with other papers. That said, for experiments where $|\mathcal{D}|$ varies, we agree that a better choice would be set $\delta$ to a small constant value (e.g. $\delta = 1e^{-5}$). It would be difficult to rerun all our experiments at this constant value as we utilized in excess of 50,000 A100 GPU hours to get the results in the paper and we no longer have the required compute capacity. However, we can add the table shown below to the paper that would convert the $\epsilon$ values for $\delta = 1 / |\mathcal{D}|$ to $\delta = 1e^{-5}$. This would allow for a more consistent comparison.
>
> | dataset | orig_epsilon | S=1 | 5 | 10 | 25 | 50 | 100 | 250 | 500 |
> |---|---|---|---|---|---|---|---| ---| ---|
> | CIFAR-10 | 1 | 3.30-3.32 | 2.21-2.33 | 1.94-2.20 | 1.69-1.71 | 1.55-1.56 | 1.43-1.46 | 1.30-1.34 | 1.22-1.24 |
> | CIFAR-10 | 2 | 5.41-5.42 | 3.95-4.49 | 3.56-3.60 | 3.18-3.22 | 2.95-2.97 | 2.76-2.92 | 2.55-2.66 | 2.41-2.50 |
> | CIFAR-10 | 4 | 9.14-9.16 | 7.17-7.62 | 6.57-6.78 | 6.01-6.11 | 5.61-5.73 | 5.31-5.68 | 4.97-5.31 | 4.73-4.87 |
> | CIFAR-10 | 8 | 15.80-15.82 | 13.11-13.84 | 12.19-13.41 | 11.35-11.71 | 10.78-11.57 | 10.25-10.83 | 9.68-10.50 | 9.28-9.51 |
> | CIFAR-100 | 1 | 1.94-1.97 | 1.54-1.65 | 1.43-1.44 | 1.30-1.31 | 1.22-1.23 | 1.16-1.17 | 1.08-1.09 | 1.03-1.04 |
> | CIFAR-100 | 2 | 3.56-3.62 | 2.95-2.98 | 2.75-2.77 | 2.55-2.55 | 2.41-2.42 | 2.29-2.30 | 2.16-2.16 | 2.07-2.08 |
> | CIFAR-100 | 4 | 6.60-6.81 | 5.63-5.65 | 5.32-5.42 | 4.96-5.02 | 4.73-4.94 | 4.53-4.55 | 4.29-4.30 | 4.14-4.15 |
> | CIFAR-100 | 8 | 12.26-12.72 | 10.76-10.85 | 10.25-10.48 | 9.67-9.85 | 9.28-9.36 | 8.94-9.03 | 8.53-8.55 | 8.25-8.29 |
> | SVHN | 1 | 3.30-3.32 | 2.19-2.23 | 1.94-1.96 | 1.69-1.71 | 1.55-1.56 | 1.43-1.44 | 1.30-1.31 | 1.23-1.24 |
> | SVHN | 2 | 5.41-5.43 | 3.95-3.97 | 3.57-3.62 | 3.18-3.20 | 2.94-2.97 | 2.75-2.91 | 2.54-2.56 | 2.41-2.43 |
> | SVHN | 4 | 9.14-9.16 | 7.14-7.44 | 6.60-7.52 | 6.00-6.50 | 5.63-5.93 | 5.32-5.68 | 4.97-5.01 | 4.73-4.77 |
> | SVHN | 8 | 15.80-15.82 | 13.03-13.71 | 12.29-12.72 | 11.35-11.76 | 10.76-11.02 | 10.25-10.33 | 9.67-9.82 | 9.29-9.40 |
>
>
>
> While setting $\delta = 1 / |\mathcal{D}|$ is not an ideal choice for small $|\mathcal{D}|$, the recomputed epsilons show the differences are not enormous and where they are accuracies are in the range of chance.
>
> We will add the fact that we used $\delta = 1 / \mathcal{D}$ everywhere as a potential limitation to the paper.
>
> **2. The authors should provide more comparison to other PEFT methods.**
>
> The same issue was raised by reviewer RPAK.  The response to that reviewer is repeated below:
>
> We agree that the paper would be improved by evaluating different adapters. However, we varied the amount of training data, privacy level, learnable parameter set, downstream dataset, and network architecture and in each experiment we tuned the hyperparameters (gradient clipping norm, epochs, batch size, and learning rate) with 20 iterations of a hyperparameter tuner (Optuna) and averaged the results of 3 runs. This consumed more than 50,000 GPU hours of computation time. Based on our experience, the specific type of adapters used does not have a large effect on the overall trends that we observed (in comparison to the items that we did vary) and making a fair comparison on a reasonable set of adapters would have exceeded our computational resources.
>
>  In addition, the recent paper “Strong Baselines for Parameter Efficient Few-Shot Fine-tuning" https://arxiv.org/pdf/2304.01917.pdf finds that “Fine-tuning just the LayerNorm parameters during few-shot adaptation is an extremely strong baseline across ViTs pre-trained with both self-supervised and supervised objective.”. In our paper, we implement FiLM via LayerNorm tuning. The aforementioned paper also indicates that the performance difference between the best adapters is relatively small.

---

> > ### Author Response · Authors · 2023-09-14
> >
> > **3. The paper needs to clearly state its contributions because it's very unclear what is being claimed by this work and what is shared with prior work.**
> >
> > In the following we attempt to clarify our contributions. At a high level, the primary differences between our work and prior works that you cite are the following:
> >
> > (i) Our work focuses exclusively on the few-shot domain. The other papers assume a full training set for $\mathcal{D}$.
> >
> > (ii) For the benefit of practitioners, our work quantifies results (some of which are known). Indeed, it is well known that when training under DP requires more data to achieve good results, however, we quantify precisely how much more data is required under various conditions. It is also well known that non-private models are vulnerable to membership inference attacks, but we quantify how vulnerable they are under a range of conditions (shot, privacy level, amount of data, etc.).
> >
> > (iii) Our work comprehensively evaluates few-shot DP models that employ a parameter efficient adapter, which has not been previously undertaken.
> >
> > (iv) We evaluate few-shot DP models on high TD (i.e. out of distribution) datasets (in particular many of the VTAB datasets). To date, previous DP work has concentrated on low TD (i.e. in distribution) datasets.
> >
> > (v) We extend the analysis to the few-shot DP federated learning setting where (to our knowledge) there are no studies on fine-tuning large pretrained models and certainly none that involve parameter efficient adapters. We establish state-of-the-art on the FLAIR federated learning benchmark by using a parameter efficient adapter which improves accuracy and considerably lowers the per round communication cost.
> >
> > **Additional responses to various comments:**
> >
> > **"Additional data is required under DP" This point has already been made in well-known work ([1])**
> >
> > Indeed, we state this in the Introduction section where we list our contributions: “It is known that classification accuracy under DP decreases as the level of privacy increases and the amount of data decreases...”. To clarify, our contribution is to quantify the amount of additional data required to match non-private accuracy. The heading will be revised to: “How much additional data is required under few-shot DP?”
> >
> > **"Transfer learning under DP is fundamentally different from non-private" This point has already been made in many prior works, see [2]**
> >
> > While reference [2] does illuminate some differences between private and non-private learning, they do not discuss learning dynamics (i.e. train and test curves) and do not consider the few-shot or high TD cases. To clarify, we will reword the first section of the summary to say: Transfer learning dynamics under DP are fundamentally different from non-private.
> >
> > **"The point at which accuracy begins to improve varies with TD" This is known even non-privately, see the well-cited work done with the Stanford WILDS dataset [3]**
> >
> > While this reference provides excellent insight info transfer learning on OOD datasets, it does not deal with DP or few-shot settings.  "The point at which accuracy begins to improve varies with TD" comment was aimed specifically at the DP case where the amount of data $S$ is sufficient to overcome the regularization imposed by DP.
> >
> > **"As ϵ increases, the train accuracy grows as the amount of DP regularization is reduced, ultimately entering the interpolating regime" There are two issues with this statement. First, without knowing the specific hyperparameters searched for, it's not necessarily the case that the amount of noise (what I assume the authors mean by DP regularization) is reduced, it could be that the number of iterations is increased.**
> >
> > We have clarified this by changing the phrase to “... as the amount of regularization pressure from DP is reduced, ...”. Our aim is to refer to the regularization effect implied by the DP definition, which sets a bound on the difference of train and test accuracies. This bound applies to any DP algorithm, regardless of the hyperparameters.
> >
> > **Second, I don't see any evidence on the plots that the models are in the interpolating regime.**
> >
> > Interpolating mode is when an overparameterized model achieves 100% training accuracy, yet test accuracy still improves. This can be seen in Figure 3 for $\epsilon = \infty$.
> >
> > **"For SVHN with ϵ = ∞, Head leaves the interpolating regime for S > 100, as there is not enough capacity to adapt to a high TD dataset. " It's not at all clear that this is an issue of capacity, because there is prior work in non-private (and even private) fine-tuning for linear probing (what the community would typically call the 'head' setting) that shows good performance on SVHN.**
> >
> > We think you are right about this. On reflection, we agree it is not necessarily a capacity issue, but perhaps an anomalous point. Maybe due to the hyper-parameter tuner not doing the correct thing at that data point?

---

> > > ### Author Response · Authors · 2023-09-14
> > >
> > > **I didn't see an explicit definition for CS in the text. The term appears at the top of page 5. My understanding is that it's the number of classes times the number of shots.**
> > >
> > > Your understanding is correct. We will clarify this in the revision.

---

> > ### Comment · Reviewer_drm9 · 2023-09-14
> >
> > Thank you for the detailed response, I have read it in detail and I feel generally positive. One question: I'm not able to get these same numbers you're getting for the $\delta$ table, the numbers I am getting for recomputed $\varepsilon$ are far higher. Could you point me to the code you used to recompute $\varepsilon$ based on decreasing $\delta$?

---

> > > ### Author Response · Authors · 2023-09-14
> > >
> > > Thanks for your reply. We are happy to hear that you feel generally positive about our response.
> > >
> > > **Could you point me to the code you used to recompute  based on decreasing $\delta$?**
> > >
> > > We will provide some our code below, which uses https://github.com/pytorch/opacus version '1.4.0'.
> > > Please do not hesitate to ask if our computation requires further clarifications.
> > >
> > > ```
> > > from opacus.accountants.utils import create_accountant
> > >
> > >
> > > def compute_epsilon_decreased_delta(noise_multiplier: float, sample_rate: float, total_steps: int, orig_delta: float, new_delta: float):
> > >     rdp_accountant = create_accountant("rdp")
> > >     rdp_accountant.history.append((noise_multiplier, sample_rate, total_steps))
> > >     orig_epsilon = rdp_accountant.get_epsilon(orig_delta)
> > >     new_epsilon = rdp_accountant.get_epsilon(new_delta)
> > >     return orig_epsilon, new_epsilon
> > > ```
> > >
> > > You can call the above script with any privacy parameters, but we provide an except from our results below:
> > >
> > > ```
> > > # CIFAR-10 with $S=1$
> > > noise_multiplier=15.625, sample_rate=1.0, total_steps=130, orig_delta=0.1, new_delta=1e-05
> > > results in: orig_eps=0.9935854149063458 new_eps=3.3040318728252274
> > > noise_multiplier=10.13671875, sample_rate=1.0, total_steps=130, orig_delta=0.1, new_delta=1e-05
> > > results in: orig_eps=1.9948319272667987 new_eps=5.415650957392214
> > > noise_multiplier=6.494140625, sample_rate=1.0, total_steps=130, orig_delta=0.1, new_delta=1e-05
> > > results in: orig_eps=3.9987686560077975 new_eps=9.158419255124453
> > > noise_multiplier=4.169921875, sample_rate=1.0, total_steps=130, orig_delta=0.1, new_delta=1e-05
> > > results in: orig_eps=7.99893036421346 new_eps=15.818443316767048
> > >
> > > # CIFAR-100 with $S=50
> > > noise_multiplier=7.734375, sample_rate=0.1111111111111111, total_steps=423, orig_delta=0.0002, new_delta=1e-05
> > > results in: orig_eps=0.9985350723626613 new_eps=1.2348540711471907
> > > noise_multiplier=12.890625, sample_rate=0.25, total_steps=796, orig_delta=0.0002, new_delta=1e-05
> > > results in: orig_eps=1.9902214626187198 new_eps=2.4079030591560486
> > > noise_multiplier=9.765625, sample_rate=0.5, total_steps=378, orig_delta=0.0002, new_delta=1e-05
> > > results in: orig_eps=3.9989524112997814 new_eps=4.738694278833799
> > > noise_multiplier=5.546875, sample_rate=0.5, total_steps=378, orig_delta=0.0002, new_delta=1e-05
> > > results in: orig_eps=7.990316814815099 new_eps=9.27596086409404
> > >
> > > # SVHN with $S=500$
> > > noise_multiplier=11.484375, sample_rate=0.14285714285714285, total_steps=567, orig_delta=0.0002, new_delta=1e-05
> > > results in: orig_eps=0.9943314031496198 new_eps=1.228526979380389
> > > noise_multiplier=4.27734375, sample_rate=0.1111111111111111, total_steps=423, orig_delta=0.0002, new_delta=1e-05
> > > results in: orig_eps=1.9922824242735215 new_eps=2.418325564608541
> > > noise_multiplier=2.4462890625, sample_rate=0.1111111111111111, total_steps=423, orig_delta=0.0002, new_delta=1e-05
> > > results in: orig_eps=3.9917870957815778 new_eps=4.761334975024106
> > > noise_multiplier=1.495361328125, sample_rate=0.1111111111111111, total_steps=423, orig_delta=0.0002, new_delta=1e-05
> > > results in: orig_eps=7.9976866304695315 new_eps=9.396769301727694
> > > ```

---

> > > > ### Comment · Reviewer_drm9 · 2023-09-14
> > > >
> > > > Thank you! Given the updated $\varepsilon$ numbers can you provide an updated quantitative statement (in the next revision of the paper) that answers this question "How much additional data is required under few-shot DP?" In particular I would be interested in seeing how you would revise the second paragraph in Section 4.1 given the feedback from myself and other reviewers.
> > > >
> > > > I think most of my concerns have been addressed by the author's response. The paper has good experimental rigor and as long as the authors can present the insights from these experiments in a manner that is useful for the reader (that is, by drawing some conclusions that have not already been reached by prior work) then this paper can make a good addition to the body of work on empirical DP fine-tuning. I will wait to see the revised version from the authors but their proposed plan for incorporating reviewer feedback sounds good to me.

---

> > > > ### Comment · Reviewer_drm9 · 2023-09-15
> > > > **Decreasing Delta Computation**
> > > >
> > > > Ok, the accounting numbers you sent are too loose (due to suboptimality of RDP) can you try using this code instead?
> > > >
> > > > ```
> > > > import scipy
> > > > import numpy as np
> > > > from scipy import stats
> > > > from scipy.optimize import fsolve
> > > >
> > > > def delta_exact(eps, T, sigma):
> > > >     mu = np.sqrt(T)/sigma
> > > >     return stats.norm.cdf(-eps/mu+mu/2)-np.exp(eps)*stats.norm.cdf(-eps/mu-mu/2)
> > > >
> > > > def eps_exact(delta, T, sigma):
> > > >     root = fsolve(lambda eps: delta_exact(eps, T, sigma)-delta, 0, xtol=1e-12)
> > > >     eps = root[0]
> > > >     return eps
> > > > ```
> > > >
> > > > These are the results I get:
> > > >
> > > > ```
> > > > eps_exact(delta=0.1, T=130, sigma=15.625) = 0.6453202894590613
> > > > eps_exact(delta=1e-5, T=130, sigma=15.625) = 3.0502298494025775
> > > > ```
> > > >
> > > > This code is for the non-subsampled Gaussian, so this will work for $S=1$, which is what's most relevant since the $\varepsilon$ doesn't increase that much for $S>1$ since the original dataset size is not that small. The reason why the exact numbers are important is because the end results need to still give us an idea of the behavior of few-shot DP under strong privacy guarantees. I know that you've already used a lot of GPU hours but it shouldn't be that much more to just run some experiments that will produce some guarantees for $\varepsilon=1$ for the new values of $\delta$.

---

> > > > > ### Author Response · Authors · 2023-09-19
> > > > > **Regarding the fixed delta**
> > > > >
> > > > > Thanks for your suggestions. We uploaded a revision that includes three changes:
> > > > > 1) Section A.3.7 with Tables 23-24 that provides the $\epsilon$ ranges for $\delta = 1e^{-5}$ based on the RDP accountant and another table that uses the PRV accountant (Gopi et al., 2021).
> > > > > 2) A reference to the new tables in Section 4.1 that states again that $\delta$ changes based on $\mathcal{|D|}$
> > > > > 3) A reminder in first bullet point of the Discussion and Recommendation (Section 6) that reminds the reader that $\delta = 1/\mathcal{|D|}$
> > > > >
> > > > > **Given the updated  $\epsilon$ numbers can you provide an updated quantitative statement (in the next revision of the paper) that answers this question "How much additional data is required under few-shot DP?"**
> > > > >
> > > > > We understand that an updated quantitative statement about much more additional data is required using a fixed $\delta$ would be ideal but we believe that it is difficult to recompute the multipliers with the updated $\epsilon$ ranges. We believe that we stress to the reader in multiple places of the paper that $\delta$ is not constant and that the insights are useful to the readers. We would like to emphasise again that the largest impact of the change in $\delta$ is in the low-shot regime where the accuracy is near random choice.
> > > > >
> > > > > **Ok, the accounting numbers you sent are too loose (due to suboptimality of RDP) can you try using this code instead?**:
> > > > >
> > > > > The recomputed epsilons at $S=1$ for CIFAR-10 and SVHN (these are the only experiments that don't use subsampling) are below. We decided to not include this table in the paper because the PRV accountant $\epsilon$ ranges (see Table 24) are similarly tight and can support subsampling.
> > > > >
> > > > > |       dataset | original epsilon     | recomputed $\epsilon$ at $S = 1$           |
> > > > > |:-----------:|:-------:|:------------:|
> > > > > | CIFAR-10 | 1 | 3.04-3.06   |
> > > > > | CIFAR-10 | 2 | 5.01-5.03   |
> > > > > | CIFAR-10 | 4 | 8.51-8.52   |
> > > > > | CIFAR-10 | 8 | 14.79-14.80 |
> > > > > | SVHN | 1     | 3.04-3.06   |
> > > > > | SVHN | 2     | 5.01-5.03   |
> > > > > | SVHN | 4     | 8.51-8.53   |
> > > > > | SVHN | 8     | 14.79-14.80 |

---

### Review · Reviewer_wjZ8 · 2023-09-07

**Summary Of Contributions:**

This paper performs a large-scale study of few-shot classification under differential privacy (DP) constraints. The authors explore how changing the privacy level, training routine, and various other parameters affects the performance of the downstream model. The authors synthesize these experiments into several high-level takeaways that seem to hold robustly across different experimental setups. As a function of these findings, the authors put forward a set of recommendations for building differentially private few-shot machine learning systems.

**Audience:**

Yes

**Claims And Evidence:**

Yes

**Requested Changes:**

I think the paper is deserving of acceptance, but would recommend the changes below to strengthen it:

- Given that TD is a quantitative metric, it would be nice to see a plot with TD on the x axis and K-shot accuracy (for some fixed value of K) on the y axis, and then have different lines for different privacy levels. This could reinforce the authors' point about the "gap" between private and non-private models increasing with transfer difficulty.

- I suspect that the correlation between the two quantities (TD and few-shot accuracy under privacy constraints) will not be perfect, i.e., TD is not quite what dictates how hard a dataset is to learn under DP. It would be nice if the authors could discuss this a bit in their paper, and maybe provide some hypotheses about what the underlying factor of variation is.

- It would be interesting to see whether the choice of efficient adapter changes the results significantly---i.e., is it just about the number of parameters, or is there something about the structure of the FiLM adapter that makes it better for DP?

- The authors should increase the font size on the labels of some of the figures (e.g., Figure 3) as they are a bit hard to read.

**Strengths And Weaknesses:**

I think this paper is a great fit for TMLR. The paper does not necessarily provide any groundbreaking insights or "surprising" results, but the experiments are extensive and thorough, proper scientific practice is followed (error bars, ablation studies, and raw tables are all provided), and I think having a list of best practices is a valuable resource for the community.

Questions:

- Is there an intuitive reason why one should expect the roughly linear trend (or i guess logarithimic, since the x axis is on a log scale) shown in Figure 2?

- From what I can tell from the figures, it seems as though even though FiLM does better than Head under DP, a large part of this could be attributed to FiLM having higher accuracy to start with (i.e., in the non-private regime). Is this an accurate interpretation? E.g., in Figure 5, it seems like the "drop" for Head models is smaller than the drop for FiLM, but FiLM starts higher up.

---

> ### Author Response · Authors · 2023-09-14
>
> Thank you for taking the time to review our paper and providing insightful feedback. We will upload a new revision of the paper to address some of your points very soon. The responses to your review are below:
>
> **Is there an intuitive reason why one should expect the roughly linear trend (or i guess logarithimic, since the x axis is on a log scale) shown in Figure 2?**
>
> There are theoretical results that show the sample complexity of learning problems under centralised DP to be $O(1/\epsilon)$. (Dropping all other factors here.) Our results seem to follow this trend quite closely. (See e.g., Ch. 11.1 of Dwork, C., & Roth, A. (2014). The algorithmic foundations of differential privacy. Foundations and Trends® in Theoretical Computer Science, 9(3–4), 211-407.)
>
> **From what I can tell from the figures, it seems as though even though FiLM does better than Head under DP, a large part of this could be attributed to FiLM having higher accuracy to start with (i.e., in the non-private regime). Is this an accurate interpretation? E.g., in Figure 5, it seems like the "drop" for Head models is smaller than the drop for FiLM, but FiLM starts higher up.**
>
> Yes, we believe that your interpretation is correct. FiLM has more capacity than Head and is able to adapt the body of network to the downstream data. As a result, we expect it to better in both the private and non-private settings.
>
> However, we would like to note that the observation that was works well in non-private would work in private does not always hold. See for example Figure 6 (parameterization for VTAB datasets), where for non-private settings, the All learnable parameters setting outperforms FiLM which outperforms Head. In contrast, for DP settings, All performs worst, FiLM and Head perform similarly, though FiLM is better in the majority of cases.
>
> **Given that TD is a quantitative metric, it would be nice to see a plot with TD on the x axis and K-shot accuracy (for some fixed value of K) on the y axis, and then have different lines for different privacy levels. This could reinforce the authors' point about the "gap" between private and non-private models increasing with transfer difficulty.**
>
> Thanks for the suggestion, we added Figure 3 and some text to refer to it. The Figure has the following caption: “Classification accuracy as a function of transfer difficulty (TD) and $\epsilon$ for CIFAR-10 (TD= 1.0), EuroSAT (TD=1.7), CIFAR-100 (TD=7.8) and SVHN (TD=52.9) at S = 100. EuroSAT has been chosen because the result for S = 100 can be easily taken from the VTAB results (Tables 8 to 13) due to $C = 10$. Backbone is VIT-B and the best performing configuration out of All, FiLM and Head is used for each ε, with $\delta = 1/|\mathcal{D}|$. The accuracy is reported over three seeds with the line showing the median and the band reporting the lowest and highest accuracy. Analysis: Accuracy gap between non-private and private training increases as TD increases.”
>
> **I suspect that the correlation between the two quantities (TD and few-shot accuracy under privacy constraints) will not be perfect, i.e., TD is not quite what dictates how hard a dataset is to learn under DP. It would be nice if the authors could discuss this a bit in their paper, and maybe provide some hypotheses about what the underlying factor of variation is.**
>
> We agree with you that TD is not a perfect measure – there are many factors at play here and capturing them in a single easy-to-compute metric is hard – and it is a valid point that it is computed using the non-private performances (also see responses to reviewers RPAK Question 3). However, when looking at the new Figure 3 (the one that you suggested) the gap between private and non-private performance for a fixed amount of $S$ increases with increasing TD. So, it seems that TD has value when predicting the performance under DP in this limited setting.
>
>
> **It would be interesting to see whether the choice of efficient adapter changes the results significantly---i.e., is it just about the number of parameters, or is there something about the structure of the FiLM adapter that makes it better for DP?**
>
> Please see our comments in response to the two other reviewers regarding adapters. There is nothing special about FiLM with respect to DP. We chose FiLM because it is (i) effective and competitive with other adapters (see “Strong Baselines for Parameter Efficient Few-Shot Fine-tuning" https://arxiv.org/pdf/2304.01917.pdf) in the non-private setting; and (ii) simple to implement (freeze all the parameters in the network except the LayerNorm affine weights and biases); and (iii) very parameter efficient. We don’t expect our results to differ much if we evaluated with a wider variety of adapters. In general, adapters with more parameters than FiLM can perform better when $S$ is high (e.g. the entire dataset), but adapters with fewer parameters do better in the few-shot setting (low $S$) as they tend not to overfit.

---

> > ### Author Response · Authors · 2023-09-14
> >
> > **The authors should increase the font size on the labels of some of the figures (e.g., Figure 3) as they are a bit hard to read.**
> >
> > Thanks for the suggestion. We increased the font size for Figure 3 (will be Figure 4 in the revision).

---

### Author Response · Authors · 2023-09-15
**We posted a first revision of the paper**

We posted a first revision of the paper where we incorporate most changes requested by the reviewers. The changes are highlighted in red and changed or added figures are framed in red.

We haven't yet fully incorporated the feedback raised by reviewer drm9 regarding the decreasing of $\delta$. We will aim to respond and suggest a revision soon.

---

> ### Author Response · Authors · 2023-09-19
> **Second revision of the paper**
>
> We just wanted to point out that we uploaded a second revision that adds the tables for $\epsilon$ given fixed $\delta$ that reviewer drm9 requested. (See detailed description of the revision in the thread that is below the review of reviewer drm9).

---

> > ### Author Response · Authors · 2023-09-24
> > **Third revision of the paper (regarding the MIA results)**
> >
> > We uploaded a new revision of the submission that solves an issue regarding the setup of the LiRA membership inference attack (MIA) experiments. The MIA shadow models were trained using the same noise seed and made the attack more powerful than originally designed by Carlini et al. (2022). The new results show that non-DP models are still highly vulnerable but the models trained under DP are more protected through DP. We updated the results and claims in the paper accordingly but the **new results don't change the main message of the paper regarding the vulnerability of the models**: We wrote in Section 6 before discovering the issue that *"The vulnerability of non- private few-shot models increases as S decreases. DP significantly mitigates the effectiveness of MIAs".* **Only the actual vulnerability scores change.**
> >
> > We would like to emphasize that our work is to the best of our knowledge the first that quantify the vulnerability against MIA of few-shot models and we believe that the insights could motivate practitioners to introduce DP training for their few-shot models. The concrete changes in the revision are the following:
> >
> > - Figure 8 updated with new results (and  Figure 22, Figure 23 and Table 14 in the appendix)
> > - Section 4.4: Update of percentage of examples that can be identified and slight rewriting of some conclusions from that
> >  - Section 6: Update of percentage of examples that can be identified and slight rewriting of some conclusions from that
> >
> > We are happy to answer any further questions from the reviewers arising from solving this issue.

---

### Decision · Action_Editors · 2023-11-06

**Recommendation:** Accept with minor revision

**Comment:**

The key contribution this paper aims to make is to offer an empirical insight into few-shot learning under differentially private constraints. A large-scale study was conducted to search for empirical insights into the choice of # of shots (per class), privacy level, downstream dataset, model architecture, and the subset of learnable parameters.

Both Reviewer wjz8 and RPAK were positive about having an empirical study of differentially private learning in various contexts; thus, I put little weight on their recommendation. Reviewer drm9 was a solicit reviewer and provided a valuable point regarding the results: the empirical study should justify their hyper-parameter choices well and the meaningful connections to existing practices.

My impression of the paper’s empirical results is not super-surprising and significant. However, according to TMLR's acceptance criteria (https://www.jmlr.org/tmlr/acceptance-criteria.html):

> Crucially, it should not be used as a reason to reject work that isn't considered “significant” or “impactful” because it isn't achieving a new state-of-the-art on some benchmark. Nor should it form the basis for rejecting work on a method considered not “novel enough”, as novelty of the studied method is not a necessary criteria for acceptance. We explicitly avoid these terms (“significant”, “impactful”, “novel”), and focus instead on the notion of “interest”. If the authors make it clear that there is something to be learned by some researchers in their area from their work, then the criteria of interest is considered satisfied

I am leaning toward accepting this paper with minor revision.

**To authors:** Reviewer drm9 has a valid point. Before camera-ready, please tone down the claims with respect to the settings of $(\epsilon, \delta)$ in the experiments. For example, "We also show that learning parameter-efficient FiLM adapters under DP is competitive with and often superior to learning just the final classifier layer or learning all of the network parameters" in the abstract.

**Audience:**

I think the audience of this paper could be limited, e.g., to those interested in the "empirical" interaction between differential-privacy and few-shot learning. Many existing works studied empirical connections between differentially privacy and (supervised) machine learning, and I hesitate to say this manuscript offers insights different from those. Another line of work in this domain is to connect theory and practice, but this work does not offer many takeaways from the theory side.

**Claims And Evidence:**

The initial draft of this paper contained empirical results from a large-scale study of few-shot classification under differential privacy (DP) constraints, but the paper lacked the conclusions, implications, and limitations of the results. After revisions, the conclusions and implications of this empirical study were included, and the limitations were also properly discussed at the end of the manuscript.

---

> ### Author Response · Authors · 2023-11-10
>
> Dear AE,
>
> thanks for the suggestions! We just uploaded a new revision of the paper that toned down some claims. See these highlighted in blue.
>
> Additionally, we modified two sentences in the abstract:
>
> 1)
> - before: We show that to achieve DP accuracy on par with non-private models, the shots per class must be increased as the privacy level increases by as much as 20 − 35× at ϵ = 1.
> - after:  We show that to achieve DP accuracy on par with non-private models, the shots per class must be increased as the privacy level increases.
>
> 2)
> - before: We also show that learning parameter-efficient FiLM adapters under DP is competitive with and often superior to learning just the final classifier layer or learning all of the network parameters.
>
> - after: We also show that learning parameter-efficient FiLM adapters under DP is competitive with learning just the final classifier layer or learning all of the network parameters.
>
> We hope these revisions are sufficient and if you agree we would prepare the camera ready version as soon as possible. If not, we would be happy to revise additional claims that you request.
>
> Best,
> the authors

---

> > ### Author Response · Authors · 2023-12-04
> > **Camera-ready posted**
> >
> > Dear AE,
> >
> > thanks for the reply.
> >
> > We just posted the camera-ready version of the paper.
> >
> > Best,
> > the Authors